# PAC Reinforcement Learning for Predictive State Representations

## Abstract

In this paper we study online Reinforcement Learning (RL) in partially observable dynamical systems. We focus on the Predictive State Representations (PSRs) model, which is an expressive model that captures other well-known models such as Partially Observable Markov Decision Processes (POMDP). PSR represents the states using a set of predictions of future observations and is defined entirely using observable quantities. We develop a novel model-based algorithm for PSRs that can learn a near optimal policy in sample complexity scaling polynomially with respect to all the relevant parameters of the systems. Our algorithm naturally works with function approximation to extend to systems with potentially large state and observation spaces. We show that given a realizable model class, the sample complexity of learning the near optimal policy only scales polynomially with respect to the statistical complexity of the model class, without any explicit polynomial dependence on the size of the state and observation spaces. Notably, our work is the first work that shows polynomial sample complexities to compete with the globally optimal policy in PSRs. Finally, we demonstrate how our general theorem can be directly used to derive sample complexity bounds for special models including $m$-step weakly-revealing and $m$-step decodable tabular POMDPs, POMDPs with low-rank latent transition, and POMDPs with linear emission and latent transition.

## 1 Introduction

Efficient exploration strategies in reinforcement learning have been well investigated on many models from tabular models [25, 2] to models with general function approximation [10, 27, 30, 16, 42]. These works have focused on fully observable Markov decision processes (MDPs); however, their algorithms do not result in statistically efficient algorithms in partially observable Markov decision processes (POMDPs). Since the markovian properties of dynamics are often questionable in practice, POMDPs are known to be useful models that capture environments in real life. While strategic exploration in POMDPs was less investigated due to its difficulty, it has been actively studied in recent few years [20, 3, 29]. In our work, we consider Predictive state representation (PSR) [36, 41, 24] that is a more general model of controlled dynamical systems than POMDPs.

PSRs are specified by the probability of a sequence of future observations/actions (referred to as a test) conditioned on the past history. Unlike the POMDP model, PSR directly predicts the future given the past *without* modeling the latent state/dynamics. PSRs can model every POMDP, but potentially result in much more compact representations; there are dynamical systems that have finite PSR ranks, but that cannot be modeled by any POMDPs with finite latent states [36, 24].

PSRs are not only general but also amenable to learning and scalable. First, PSRs can be efficiently learned from exploratory data using a spectral learning algorithm [6] motivated by method-of-moments [23]. This learning algorithm allows us to perform fast closed-form sequential filtering, unlike EM-type algorithms that would be the most natural algorithm derived from POMDP perspectives. Secondly, while original PSRs are defined in the tabular setting, PSRs also support rich functional forms through kernel mean embedding [4]. Variants of PSRs equipped with neural networks have been proposed as well [43, 9, 46, 49].

In spite of the abovementioned advances in research on PSRs made in the recent two decades, strategic exploration without exploratory data has been barely investigated. To make PSRs more practical, it is of significant importance to understand how to perform efficient strategic exploration.

| | $m$-step weakly-revealing POMDPs | $m$-step decodable POMDPs | PSRs |
|---|:---:|:---:|:---:|
| Efroni et al. [13] | | + | |
| Liu et al. [37] | + | | |
| Jiang et al. [26] | | | ○ |
| Uehara et al. [45] | ○ | + | ○ |
| Our Work | + | + | + |

Table 1: Comparison of our work with existing works. $+$ means that algorithms can learn the near *globally* optimal policy with polynomial sample complexities. Our work is the *only* work that has a desirable guarantee on three models. In $m$-step weakly-revealing POMDPs, ○ in Uehara et al. [45] means the sample complexity is quasi-polynomial but not polynomial. In $m$-step decodable POMDPs, all of the works have certain caveats. More specifically, in Efroni et al. [13], Uehara et al. [45], it is unclear whether they can avoid $\text{poly}(|\mathcal{O}|^m)$. On the other hand, our result can surprisingly avoid $\text{poly}(|\mathcal{O}|^m)$ while we need a regularity assumption. For more details, refer to Section 5. In PSRs, ○ in Jiang et al. [26] means the guarantee is limited to reactive PSRs where the optimal value function depends on current observations. Similarly, ○ in Uehara et al. [45] means the algorithm can compete with short-memory policies but not near globally optimal policies.

To the best of the author's knowledge, Jiang et al. [26], Uehara et al. [45] tackle this challenge; however, they fail to show results with polynomial sample complexity to compete with the globally optimal policy. We aim to obtain algorithms that can compete with the globally optimal policy with polynomial sample complexity. Another desideratum for algorithms is to permit for general function approximation. This desideratum is important to enjoy the scalable property of PSRs. In summary, the key question we wish to address in this work is:

Can we design provably efficient RL algorithms for learning PSR with function approximation?

**Contributions.** Our main contributions are summarized below. This is summarized in Table 1.

1. We develop the first PAC learning algorithm for PSRs that can compete with the globally optimal policy and identify the PSR rank $d_{\text{PSR}}$ as the key structural quantity of PSR systems. Starting with a realizable model class, our algorithm learns a near-optimal policy with sample complexity scaling polynomially in $d_{\text{PSR}}$ and the statistical complexity (log bracket number of the model class), without any explicit polynomial dependence on the size of state and observation space. Thus, our approach can be applied to large-scale partially observable systems.

2. We demonstrate how our general result can be seamlessly applied to existing POMDP models with function approximation. These models include tabular $m$-step weakly-revealing POMDPs [37] and tabular $m$-step decodable POMDPs [13]. Especially, our work is the first work that ensures PAC guarantees with polynomial sample complexities for $m$-step weakly-revealing POMDPs and $m$-step decodable POMDPs *simultaneously*. We further show sample complexity results when these two types of POMDPs have additional two types of structures to permit for large state/observation space: with low-rank latent transition and with linear latent transition and observation distributions, which all have $d_{\text{PSR}}$ much smaller than $|\mathcal{S}|$.

**Notations.** In this work we use $[n]$ to denote the set $\{1, 2, \cdots, n\}$ and $[n]^+$ to denote the set $\{0, 1, 2, \cdots, n\}$ for any positive integer $n$. For any set $\mathcal{C}$, we use $|\mathcal{C}|$ to denote its cardinality and $[x_c]_{c \in \mathcal{C}}$ to denote the vector whose entry is $x_c$ for all $c \in \mathcal{C}$. We also use $\Delta_{\mathcal{C}}$ to represent the set of all probability distributions over $\mathcal{C}$. For any vector $x$, we use $\|x\|_1$, $\|x\|_2$ and $\|x\|_\infty$ to denote its $\ell_1$, $\ell_2$ and $\ell_\infty$ norm. For any matrix $M$, we use $(M)_{i,j}$ to denote the $(i, j)$-th entry of $M$ and $M^\dagger$ to denote the pseudo inverse of $M$. We also use $\|M\|_{\infty,\infty}$ to denote $\max_{i,j} |(M)_{i,j}|$ and $\|M\|_{1 \mapsto 1}$ to denote its $\ell_1$ norm $\sup_{\|x\|_1 = 1} \|Mx\|_1$. In addition, we use $\sigma_{\min}(M)$ to denote the minimum nonzero singular value of $M$ and $\sigma_n(M)$ to denote the $n$-th largest singular value of $M$.

**Related works.** Our work is mostly related to the literature on provable online RL algorithms for PSRs without offline exploratory data. Although there is a growing body of literature that discusses efficient online learning for POMDPs under various structures [3, 20, 34, 35, 40, 7], there are few works [26, 45] that study strategic exploration in PSRs and none of them obtain polynomial sample complexity results for learning globally optimal policies. See Appendix B for details.

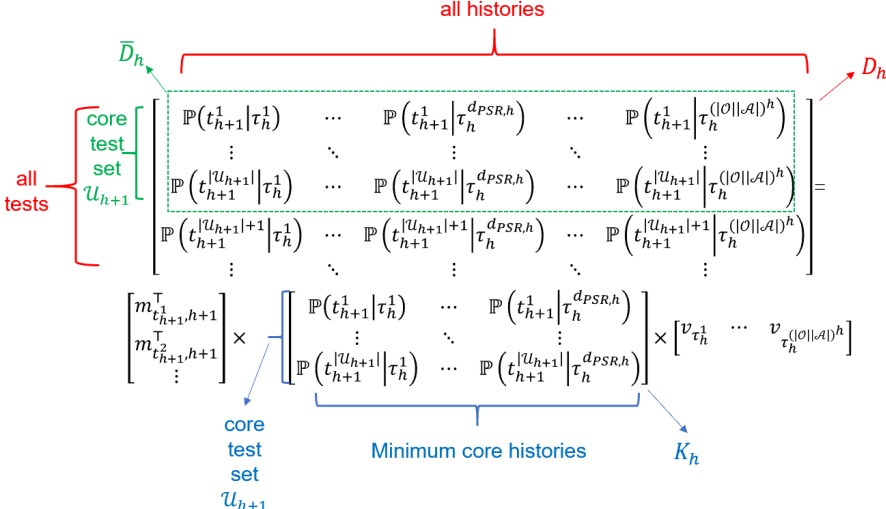

Figure 1: Illustration of the key concepts in PSRs using the system dynamics matrix $D_h$ indexed by all tests and all histories. Denote $d_{\text{PSR},h}$ as the rank of $D_h$. A core test set $\mathcal{U}_{h+1}$ is a subset of tests such that the submatrix $\overline{D}_h$ whose rows are indexed by tests in $\mathcal{U}_{h+1}$ has rank $d_{\text{PSR},h}$. Thus any row in $D_h$ can be written as a linear combination of the rows in $\overline{D}_h$. A core test set $\mathcal{U}_{h+1}$ whose size is exactly equal to $d_{\text{PSR},h}$ is called a minimum core test set. The minimum core history set is a size–$d_{\text{PSR},h}$ subset of histories such that the submatrix $K_h$ of $\overline{D}_h$ whose columns are indexed by the history in the minimum core history set has rank $d_{\text{PSR},h}$. Any column in $D_h$ can be written as a linear combination of columns indexed by histories in the minimum core history set.

## 2 PRELIMINARIES

In this section, we introduce the definition and key properties of PSRs. After that, we state our learning objective for PSRs with function approximation.

### 2.1 PREDICTIVE STATE REPRESENTATIONS

We consider an episodic sequential decision making process $\mathcal{P} = \{\mathcal{O}, \mathcal{A}, \mathbb{P}, \{r_h\}_{h=1}^H, H\}$, where $\mathcal{O}$ is the observation space, $\mathcal{A}$ is the action space, $\mathbb{P}$ is the system dynamics, $r_h$ is the reward function at $h$-th step and $H$ is the length of each episode. We suppose the reward $r_h$ at $h$-th step is a deterministic function of $(o_h, a_h)$ conditioned on the history $\tau_h$ where $\tau_h = (o_1, a_1, \cdots, o_h, a_h)$.

We assume the initial observation $o_1$ of each episode follows a fixed distribution $\mu_1 \in \Delta_{\mathcal{O}}$. At step $h \in [H]$, the agent observes the observation $o_h$ and takes action $a_h$ based on the whole history $(\tau_{h-1}, o_h)$. After that, the agent receives its reward $r_h(o_h, a_h)$ and the environment generates $o_{h+1} \sim \mathbb{P}(\cdot|\tau_h)$. After the agent takes $a_H$, we suppose the environment will only generate dummy observations $o_{\text{dummy}}$ no matter what actions the agent takes afterward.

**Policy and value.** A policy $\pi = \{\pi_h : (\mathcal{O} \times \mathcal{A})^{h-1} \times \mathcal{O} \to \Delta_{\mathcal{A}}\}_{h=1}^H$ specifies the action selection probability at each step conditioned on the history $(\tau_{h-1}, o_h)$. Given any policy $\pi$, its value $V^\pi$ is the expected cumulative reward as defined below: $V^\pi := \mathbb{E}_\pi[\sum_{h=1}^H r_h]$, where the expectation is w.r.t. to the distribution of the trajectory induced by executing $\pi$ in the environment. We use $\mathbb{P}^\pi(\tau)$ to represent the probability of trajectory $\tau$ when executing policy $\pi$ in the environment.

#### 2.1.1 KEY CONCEPTS IN PSRS

**Tests and Linear PSRs.** A test is a sequence of future observations and actions. For some test $t_h = (o_{h:h+W-1}, a_{h:h+W-2})$ with length $W \in \mathbb{N}^+$, we define the probability of test $t_h$ being successful conditioned on reachable history $\tau_{h-1}$ as $\mathbb{P}(t_h|\tau_{h-1}) := \mathbb{P}(o_{h:h+W-1}|\tau_{h-1}; \text{do}(a_{h:h+W-2}))$, i.e., the probability of observing $o_{h:h+W-1}$ by actively executing actions $a_{h:h+W-2}$ conditioned on history $\tau_{h-1}$.[1] When the history $\tau_{h-1}$ is unreachable, i.e.,

---

[1]The do operator means that $\mathbb{P}(o_{h:h+W-1}|\tau_{h-1}; \text{do}(a_{h:h+W-2})) = \prod_{t=h}^{h+W-1} \mathbb{P}(o_t|\tau_{h-1}, o_{h:t-1}, a_{h:t-1})$. Here, we remark conditional probability of $o_{h:h+W-1}$ given $\tau_{h-1}$ is not specified not only by dynamics, but also by the policy. Given a policy $\pi$, conditional probability of $o_{h:h+W-1}$ given $\tau_{h-1}$ under a policy $\pi_t$ is $\mathbb{P}(o_{h:h+W-1}|\tau_{h-1}; a_{h:h+W-2} \sim \pi) \propto \prod_{t=h}^{h+W-1} \mathbb{P}(o_t|\tau_{t-1})\pi_t(a_t|\tau_{t-2}, o_t)$. The $\text{do}(a_{h:h+W-2})$ operator can be understood as a policy that deterministically picks actions $a_t$ for $h \le t \le h + W - 2$, i.e., $\pi_t(A_t = \cdot|\tau_{t-1}, o_t) = \delta_{a_t}$.

$\mathbb{P}^\pi(\tau_{h-1}) = 0$ for all policy $\pi$, we define the conditional probability $\mathbb{P}(t_h|\tau_{h-1})$ to be 0. Now, we define the one-step system dynamics matrix $D_h$ whose rows are indexed by tests and columns are indexed by histories, and the entry corresponding to the test-history pair $(t_{h+1}, \tau_h)$ is equal to $\mathbb{P}(t_{h+1}|\tau_h)$ (see Fig 1 for an illustration). Denote $d_{\mathrm{PSR},h} = \mathrm{rank}(D_h)$. Then *Linear PSRs* are defined to be systems with low-rank one-step system dynamic matrices:

**Definition 1.** *A partially observable system is called a Linear PSR with rank $d_{\mathrm{PSR}}$ if $\max_h \mathrm{rank}(D_h) = d_{\mathrm{PSR}}$.*

**Core tests.** For time step $h$, consider a set of tests $\mathcal{U}_{h+1} \subset \cup_{C\in\mathbb{N}^+}\mathcal{O}^C \times \mathcal{A}^{C-1}$. If the submatrix $\overline{D}_h$ (see the matrix inside the green box in Fig. 1) of $D_h$ whose rows are indexed by the tests in $\mathcal{U}_{h+1}$ and columns are indexed by all histories, has rank equal to $d_{\mathrm{PSR},h}$, then we call such set $\mathcal{U}_{h+1}$ as a *core test set*. The key property of such a core test set is that from linear algebra, for any row in $D_h$, we can express it as a linear combination of the rows of $\overline{D}_h$. This is formalized as follows.

**Lemma 1** (Core test sets in linear PSRs). *For any $h \in [H-1]^+$, a set $\mathcal{U}_{h+1} \subset \cup_{C\in\mathbb{N}^+}\mathcal{O}^C \times \mathcal{A}^{C-1}$ is a core test set at $(h+1)$-th step if and only if we have for any $W \in \mathbb{N}^+$, any possible future (i.e., test) $t_{h+1} = (o_{h+1:h+W}, a_{h+1:h+W-1}) \in \mathcal{O}^W \times \mathcal{A}^{W-1}$ and any history $\tau_h$, there exists $m_{t_{h+1},h+1} \in \mathbb{R}^{|\mathcal{U}_{h+1}|}$ such that*

$$\mathbb{P}(t_{h+1}|\tau_h) = \langle m_{t_{h+1},h+1}, [\mathbb{P}(u|\tau_h)]_{u\in\mathcal{U}_{h+1}}\rangle. \tag{1}$$

*The vector $[\mathbb{P}(u|\tau_h)]_{u\in\mathcal{U}_{h+1}}$ is referred to as the predictive state at $(h+1)$-th step.*

Throughout this work, we use $q_{\tau_h}$ to denote $[\mathbb{P}(u|\tau_h)]_{u\in\mathcal{U}_{h+1}}$ and $q_0$ to represent the initial predictive states $[\mathbb{P}(u)]_{u\in\mathcal{U}_1}$. In particular, we are interested in the set of all action sequences in $\mathcal{U}_h$ and denote it by $\mathcal{U}_{A,h}$. A core test set with the smallest number of tests is called a minimum core test set, which we denote by $\mathcal{Q}_h$. Note that by the definition of the rank, we know that $|\mathcal{Q}_{h+1}| = d_{\mathrm{PSR},h}$. To simplify writing, we further define $|\mathcal{U}| := \max_{h\in[H]}|\mathcal{U}_h|, |\mathcal{U}_A| := \max_{h\in[H]}|\mathcal{U}_{A,h}|$. In this paper we assume a core test $\mathcal{U}_h$ (we will see that this is a natural assumption for models such as POMDPs) is given while $\mathcal{Q}_h$ is *unknown*. This setting is standard in literature on PSRs [6].

**Minimum core histories.** Similar to the minimum core test set, we can define the minimum core history set as well. Consider the matrix $\overline{D}_h$ in Figure 1. Recall that the columns of $\overline{D}_h$ are indexed by all possible $h$-length histories and each column is $q_{\tau_h}$. Since $\overline{D}_h$ has rank $d_{\mathrm{PSR},h}$, there must exist $d_{\mathrm{PSR},h}$ histories $\tau_h^1, \cdots, \tau_h^{d_{\mathrm{PSR},h}}$, such that any column in $\overline{D}_h$ is a linear combination of the columns in $\overline{D}_h$ that correspond to histories $\tau_h^1, \cdots, \tau_h^{d_{\mathrm{PSR},h}}$. In other words, for any $h$-length history $\tau_h$, there exists a vector $v_{\tau_h} \in \mathbb{R}^{d_{\mathrm{PSR},h}}$ which satisfies

$$q_{\tau_h} = K_h v_{\tau_h}, \tag{2}$$

where $K_h \in \mathbb{R}^{|\mathcal{U}_{h+1}|\times d_{\mathrm{PSR},h}}$ is a full-rank matrix whose $i$-th column is $q_{\tau_h^i}$. We call $\{\tau_h^1, \cdots, \tau_h^{d_{\mathrm{PSR},h}}\}$ as the minimum core histories at step $h$ and $K_h$ as the core matrix – see Figure 1 for an illustration of $K_h$. Particularly, when $h = 0$, we have $K_0 = q_0$. Note (2) shows that all $h$-length histories can be captured by the core histories in the sense that the predictive states given any history can be expressed as a linear combination of the predictive states corresponding to the minimum core histories. The minimum core histories and the core matrix may not be unique given the core test set. Here we particularly define $K_h$ to be the core matrix with the smallest $\|K_h^\dagger\|_{1\mapsto 1}$ to facilitate our subsequent analysis.

**PSRs vs POMDPs.** PSRs have much stronger expressivity than POMDPs. All POMDPs can be expressed as PSRs with the minimum core test set size as most $|\mathcal{S}|$ while PSRs are not necessarily compact POMDPs [36]. In Appendix D we construct a sequential decision making process where if we want to formulate it into a POMDP, the number of states we need will be exponentially larger than the core test set size in the PSR formulation. The key intuition behind the construction is simple: the non-negative rank of a matrix could be exponentially larger than its rank. In the literature [41], there are also some other concrete instances like probability clock which POMDPs cannot model with finite latent states while PSRs can model with finite rank.

In the following, we explain that POMDPs are PSRs. Consider an episodic POMDP $(\mathcal{S}, \mathcal{O}, \mathcal{A}, \{\mathbb{T}_h\}_{h=1}^H, \{\mathbb{O}_h\}_{h=1}^H, \{r_h\}_{h=1}^H, H, \mu_1)$ where $\mathcal{S}$ is the state space, $\mathcal{O}$ is the observation space, $\mathcal{A}$ is the action space, $\mathbb{O}_h$ is the emission matrix at $h$-th step, $r_h$ is the reward function at $h$-th step and $\mu_1$

is the initial state distribution, $\mathbb{T}_h$ is the transition matrix at step $h$ where $(\mathbb{T}_{h,a})_{s',s} = \mathbb{P}_h(s'|s,a)$. Then the following lemma shows that any POMDP is a PSR and its minimum core test set size is no larger than $|\mathcal{S}|$, whose proof is deferred to Appendix E:

**Lemma 2.** *All POMDPs satisfy the definition of PSRs (Definition 1) such that $d_{\mathrm{PSR}} \leq |\mathcal{S}|$.*

After showing any POMDP is a linear PSR with rank at most $|\mathcal{S}|$, now we demonstrate that under what conditions we could find a core test set. We focus on 1-step weakly-revealing POMDPs [29, 37] here, i.e., the rank of $\mathbb{O}_h$ is $|\mathcal{S}|$ for all $h$, then we can show that $\mathcal{O}$ is a core test set.

**Lemma 3.** *When $\mathrm{rank}(\mathbb{O}_h) = |\mathcal{S}|$ for all $h \in [H]$, the POMDP is a PSR with the core test set $\mathcal{U}_h = \mathcal{O}$ for all $h \in [H]$.*

We defer the proof and other examples including $m$-step weakly-revealing POMDPs [37], latent MDPs [34], $m$-step decodable POMDPs [13] and low-rank POMDPs to Appendix E and Section 5.

### 2.1.2 System dynamics of PSRs

Predictive states can evolve just like the beliefs in POMDPs, which indicates that we can characterize the system dynamics of the PSRs via predictive states efficiently. In particular, for any $o \in \mathcal{O}, a \in \mathcal{A}, h \in [H]$, let $M_{o,a,h} \in \mathbb{R}^{|\mathcal{U}_{h+1}| \times |\mathcal{U}_h|}$ denote the matrix whose rows are $m_{(o,a,u),h}^{\top}$ (defined in Lemma 1) for $u \in \mathcal{U}_{h+1}$ (note that $o, a, u$ can be understood as a test that starts with $o, a$, followed by $u$). Then the probability of an arbitrary trajectory can be expressed as the product of $M_{o,a,h}, m_{o_H,H}, q_0$, as shown in the following lemma:

**Lemma 4.** *For any trajectory $\tau_H$ and policy $\pi$, we have*

$$\mathbb{P}^\pi(\tau_H) = m_{o_H,H}^{\top} \cdot \prod_{h=1}^{H-1} M_{o_h,a_h,h} \cdot q_0 \cdot \pi(\tau_H), \tag{3}$$

*where $\pi(\tau_H) := \prod_{h=1}^H \pi(a_h|\tau_{h-1}, o_h)$ is the probability of the actions chosen in the trajectory. More generally, for any $h \in [H]$ and trajectory $\tau_h$, letting $b_{\tau_h} := \{\prod_{l=1}^h M_{o_l,a_l,l}\} q_0$, we have*

$$[\mathbb{P}(u|\tau_h)\mathbb{P}^\pi(\tau_h)]_{u \in \mathcal{U}_{h+1}} = b_{\tau_h} \times \pi(\tau_h), \tag{4}$$

The proof is deferred to Appendix F. Lemma 4 shows that the parameters

$$\{M_{o,a,h}, m_{o,H}, q_0\}_{o \in \mathcal{O}, a \in \mathcal{A}, h \in [H-1]}$$

are sufficient to characterize a PSR. Here we call $M_{o,a,h}$ the predictive operator matrix. Recall that in POMDPs, the same decomposition holds since we can represent $\{M_{o,a,h}, m_{o,H}, q_0\}$ using $\{\mathbb{T}_h, \mathbb{O}_h\}$ as we see in the proof of Lemma 3. However, as emphasized in Singh et al. [41], the main reason PSRs are more expressive is that $\{M_{o,a,h}, m_{o,H}, q_0\}$ are *not* constrained to be non-negative.

### 2.1.3 Learning Objective

In this paper, we study online learning in PSRs and want to find the optimal policy. Suppose the output policy is $\widehat{\pi}$, then our goal is to find an $\epsilon$-optimal policy with polynomial number of samples such that: $V^* - V^{\widehat{\pi}} \leq \epsilon$, where $V^* := V^{\pi^*} = \sup_\pi V^\pi$ and $\pi^*$ is the optimal policy.

## 2.2 Function Approximation

To deal with the potentially large observation and action space, we consider learning with function approximation in this paper. We assume a function class $\mathcal{F}$ to approximate the true model and let $\mathbb{P}_f^\pi(\tau_H)$ denote the probability of any trajectory $\tau_H$ under any policy $\pi$ and model $f$. Here we assume that the models in $\mathcal{F}$ are all valid PSRs with core test set $\{\mathcal{U}_h\}_{h \in [H]}$, which implies that for each $f \in \mathcal{F}$, we can calculate its corresponding predictive operator matrices, initial predictive states and core matrices, denoted by $M_{o,a,h;f}, q_{0;f}, K_{h;f}$ respectively. We define $V_f^\pi$ to be the value of policy $\pi$ under model $f$. We also use $f^*$ to represent the true model for consistency.

Generally, we put models on $\{M_{o,a,h;f}, q_{0:f}\}$ since this is the most natural parametrization in PSRs. When we have more prior knowledge about models like models are POMDPs, we can also put models on $\{\mathbb{T}_h, \mathbb{O}_h, \mu_1\}$.

To measure the size of $\mathcal{F}$, we use $|\mathcal{F}|$ to denote its cardinality when $\mathcal{F}$ is finite. For infinite function classes, we introduce the $\epsilon$-bracket number to measure its size, which is defined as follows:

**Definition 2** ($\epsilon$-bracket and $\epsilon$-bracket number). *A size-$N$ $\epsilon$-bracket is a bracket $\{g_1^i, g_2^i\}_{i=1}^N$ where $g_1^i(g_2^i)$ is a function mapping any policy $\pi$ and trajectory $\tau$ to $\mathbb{R}$ such that for all $i \in [N]$, $\|g_1^i(\pi, \cdot) - g_2^i(\pi, \cdot)\|_1 \leq \epsilon$ for any policy $\pi$, and for any $f \in \mathcal{F}$, there must exist an $i \in [N]$ such that $g_1^i(\pi, \tau_H) \leq \mathbb{P}_f^\pi(\tau_H) \leq g_2^i(\pi, \tau_H)$ for all $\tau_H, \pi$. The $\epsilon$-bracket number of $\mathcal{F}$, denoted by $\mathcal{N}_{\mathcal{F}}(\epsilon)$, is the minimum size of such an $\epsilon$-bracket.*

Although $\mathbb{P}_f^\pi$ is an $(|\mathcal{O}||\mathcal{A}|)^H$-dimensional vector, its log $\epsilon$-bracket number will not scale exponentially with $H$ because $\mathbb{P}_f^\pi$ is Lipschitz continuous with respect to $\{M_{o,a,h;f}, q_{0;f}\}$, whose dimension only scales polynomially with $H$. In Appendix G we show that the bracket number of $\mathcal{F}$ can be upper bounded by the covering number of $\{M_{o,a,h;f}, q_{0;f}\}$ in linear PSRs, and we provide exact upper bounds for tabular PSRs and various POMDPs.

## 3 ALGORITHM: CRANE

The statistical hardness of learning POMDPs due to the partial observability is well-known in the literature [32], which also exists in PSR learning since PSRs are a more general model. In addition, existing algorithms [29, 13, 37] for learning sub-classes of POMDPs require the existence of latent states since they directly put models on $\mathbb{T}$ and $\mathbb{O}$. Thus, their methods are not applicable to PSRs. That said, the existence of predictive states indeed implies the low-rank linear structure of PSRs. The trajectory probability decomposition (3) further suggests that we are able to capture a PSR completely as long as we can learn the predictive operator matrices $\{M_{o,a,h}\}_{o \in \mathcal{O}, a \in \mathcal{A}, h \in [H-1]}$ and the initial predictive state $q_0$ efficiently. Therefore, inspired from the success of maximum log-likelihood estimation (MLE) in learning weakly-revealing POMDPs [37], we propose a new MLE-based PSR learning algorithm to learn these parameters as follows.

**CRANE.** Intuitively, our algorithm is an iterative MLE algorithm with optimism, where in each iteration we use MLE to estimate the model parameters based on the previously collected trajectories and choose an optimistic policy to execute. We call it OptimistiC PSR leArniNg with MLE (CRANE). CRANE mainly consists of three steps, whose details are shown in Algorithm 1:

- **Optimism:** Since we consider the online learning problem, the unknown model dynamics force us to deal with the exploration-exploitation tradeoff. Here we utilize the *Optimism in the Face of Uncertainty* principle and choose an optimistic estimation $f^k$ of the model parameters from the constructed confidence set $\mathcal{B}^k$. Our policy $\pi^k$ is the optimal policy under $f^k$, ensuring that $V_{f^k}^{\pi^k} \geq V^*$ with high probability.

- **Trajectory collection:** For each step $h \in [H-1]^+$ and each action sequence $u_{a,h+1}$ in $\mathcal{U}_{A,h+1}$, we collect a trajectory $\tau_H^{k,u_{a,h+1},h}$ by executing the policy $\pi^{k,u_{a,h+1},h} = \pi_{1:h-1}^k \circ \text{Unif}(\mathcal{A}) \circ u_{a,h+1}$ (and uniform policy afterwards if the episode has not ended). This helps us obtain the required information for estimating each predictive operator matrix $M_{o,a,h}$ and initial predictive state $q_0$.

- **Parameter estimation with MLE:** Finally we need to update the confidence set with newly collected trajectories. We achieve this by implementing MLE on all the collected trajectories with slackness $\beta$, as shown below:

$$\mathcal{B}^{k+1} \leftarrow \left\{ f \in \mathcal{F} : \sum_{(\pi, \tau_H) \in \mathcal{D}} \log \mathbb{P}_f^\pi(\tau_H) \geq \max_{f' \in \mathcal{F}} \sum_{(\pi, \tau_H) \in \mathcal{D}} \log \mathbb{P}_{f'}^\pi(\tau_H) - \beta \right\}. \quad (5)$$

For example, the likelihood $\mathbb{P}_f^\pi(\tau_H)$ is specified by (3) if we model $\{M_{o,a,h}, q_0\}$ as $f$. In POMDPs, if we model $\{\mathbb{T}_h, \mathbb{O}_h, \mu_1\}$ as $f$, the likelihood is specified by marginalizing over latent states. The slackness $\beta$ is chosen appropriately such that the true parameters $f^* \in \mathcal{B}^{k+1}$ with high probability, which in turn guarantees optimism in the first step.

**Comparison with Liu et al. [37].** The main difference is that our algorithm can allow more general models. For example, in PSRs, we can generally take $\{M_{o,a,h}, q_0\}$ that depends on only observable quantities as a model $f$. On the other hand, Liu et al. [37] attempts to put models on $\{\mathbb{T}_h, \mathbb{O}_h\}$ that involve latent states. The practical benefit of modeling $\{M_{o,a,h}, q_0\}$ is we don't need to specify the latent space while Liu et al. [37] needs to do. Since we often do not have good prior knowledge about latent states, our algorithm would be more practical in this scenario. Due to the generality of our algorithm, we can capture more models such as $m$-step decodable POMDPs and low-rank POMDPs as we will see in the following sections.

---

**Algorithm 1 `CRANE`**

---

**Input**: confidence parameter $\beta$.
Initialize $\mathcal{B}^1 \leftarrow \mathcal{F}, \mathcal{D} = \emptyset$.
**for** $k = 1, \cdots, T$ **do**
    **Optimistic Planning:** $(f^k, \pi^k) \leftarrow \arg\max_{f \in \mathcal{B}^k, \pi} V_f^\pi$.
    **Collect samples:**
    **for** $h \in [H-1]^+, u_{a,h+1} \in \mathcal{U}_{A,h+1}$ **do**
        Execute $\pi^{k,u_{a,h+1},h} = \pi_{1:h-1}^k \circ \text{Unif}(\mathcal{A}) \circ u_{a,h+1}$ and collect a trajectory $\tau_H^{k,u_{a,h+1},h}$.
        Update the dataset $\mathcal{D} \leftarrow \mathcal{D} \cup (\pi^{k,u_{a,h+1},h}, \tau_H^{k,u_{a,h+1},h})$.
    **end for**
    **Update confidence set:** Compute $\mathcal{B}^{k+1}$ via (5).
**end for**

---

## 4 MAIN RESULT

Next, we present the regret analysis for `CRANE`. We will utilize the fact that the core matrix $K_h$ is full-rank. However, matrix rank is vulnerable to estimation errors since small perturbations might change the rank drastically. Here we assume the $\ell_1$ norm of $K_h^\dagger$ is upper bounded, which is a more robust assumption than $K_h$ being full rank. Note a similar assumption is often imposed in the PSR literature [27, Appendix B.4].

**Assumption 1** ($\alpha$-regularity of PSRs.). *Assume that there exists $\alpha > 0$ such that for any $h \in [H-1]^+$, we have $\|K_h^\dagger\|_{1 \mapsto 1} \leq 1/\alpha$.*

**Remark 1.** *$\|K_h^\dagger\|_{1 \mapsto 1}$ can be upper bounded by $\sqrt{d_{\text{PSR},h}}/\sigma_{\min}(K_h)$. In the literature of POMDPs, many works [29, 13, 37] assume a similar condition called $\alpha$-weakly revealing condition. That is, the minimal singular value of the observation matrix or the multi-step observation matrix is lower bounded by $\alpha$. Assumption 1 can be regarded as a generalization of such weakly revealing condition in PSRs by viewing core histories as the "states" and tests $u \in \mathcal{U}_h$ as the "observations" in PSRs.*

In addition, to simplify analysis, we assume all one-step observations $o \in \mathcal{O}$ belong to $\mathcal{U}_H$. This does not harm the generality of our model since augmenting the core test set is always feasible and adding all one-step observations will at most increase $|\mathcal{U}_A|$ by one.

**Assumption 2.** *For all $o \in \mathcal{O}$, we assume that $o \in \mathcal{U}_H$.*

This assumption immediately implies that $m_{o,H} = e_{o,H}$, i.e., it is a one-hot vector which indexes the observation $o$ in $\mathcal{U}_H$. To see that, note that $o \in \mathcal{U}_H$ implies that the predictive state $q_{\tau_{H-1}}$ contains the probability $\mathbb{P}(o|\tau_{H-1})$. Thus, when $m_{o,H} = e_{o,H}$, we have $m_{o,H}^\top q_{\tau_{H-1}} = \mathbb{P}(o|\tau_{H-1})$. Therefore when Assumption 2 holds, we can assume that for all models induced by $\mathcal{F}$, we have $m_{o,H;f} = e_{o,H}$ for all $o$ without loss of generality.

Furthermore, we impose constraints on the function class $\mathcal{F}$ as follows:

**Assumption 3.** *Assume the function class $\mathcal{F}$ satisfies the following conditions: (1) **Realizability:** We have $f^* \in \mathcal{F}$, (2) **Regularity:** For all $f \in \mathcal{F}$ and $h \in [H-1]^+$, we have $\|K_{h;f}^\dagger\|_{1 \mapsto 1} \leq 1/\alpha$, (3) **Validity:** For all $f \in \mathcal{F}$, the model dynamics induced by $f$ is a valid PSR with core test set $\{\mathcal{U}_h\}_{h \in [H]}$, i.e., the trajectory probability $\mathbb{P}_f^\pi$ should be a valid distribution for any policy $\pi$ and satisfies the definition of PSRs.*

The last two constraints (2), (3) in Assumption 3 can be easily satisfied by eliminating those functions which do not satisfy the regularity or validity.

Notice that the system dynamics in (3) only utilizes the inner product of $m_{(o,a,u),h;f}$ and $q_{\tau_{h-1};f}$, and $q_{\tau_{h-1};f}$ lives in the column space of $K_{h-1;f}$ (i.e., (2)), which implies there is redundancy in the choice of $m_{(o,a,u),h;f}$ given the model $\mathbb{P}_f^\pi$. Next we show that among these possible $m_{(o,a,u),h;f}$, we can always find one that lies in the column space of $K_{h-1;f}$. More specifically, if we replace any $m_{(o,a,u),h;f}$ with its projection on the space spanned by $\{q_{\tau_{h-1};f}\}_{\tau_{h-1}}$ (which is exactly the column space of $K_{h-1;f}$), the resulting model dynamics will remain the same. Formally, we can show given any $\mathbb{P}_f^\pi$, there exists a set of $\{m_{(o,a,u),h;f}\}_{o \in \mathcal{O}, a \in \mathcal{A}, u \in \mathcal{U}_{h+1}, h \in [H-1]}$ such that $m_{(o,a,u),h;f}$ belongs to the column space of $K_{h-1;f}$ (the proof is deferred to Appendix H). Therefore, in the following, we let $M_{o,a,h;f}$ consist of such $m_{(o,a,u),h;f}$ without loss of generality.

| Model | Core test set $\mathcal{U}_h$ | $d_{\text{PSR}}$ | $\log \mathcal{N}_{\mathcal{F}}(\epsilon)$ |
|---|---|---|---|
| tabular POMDPs | | $\leq |\mathcal{S}|$ | $\text{poly}(|\mathcal{O}|, |\mathcal{A}|, |\mathcal{S}|, H, \log(1/\epsilon))$ |
| low-rank POMDPs | $(\mathcal{O} \times \mathcal{A})^{m-1} \times \mathcal{O}$ | $\leq d_{\text{trans}}$ | Refer to Appendix A.2 |
| linear POMDPs | | $\leq d_{\text{lin}}$ | $\text{poly}(d_{\text{lin}}, H, \log(|\mathcal{O}||\mathcal{A}|/\epsilon))$ |

Table 2: Core test sets, minimum core test size and bracket number for POMDP models. Here all the POMDPs we consider are $m$-step weakly-revealing or $m$-step decodable. The exact function classes $\mathcal{F}$ we use are elaborated in the following discussion.

With the above assumptions, we have the following theorem, which characterizes the sample complexity of CRANE to learn a near-optimal policy, whose proof is deferred to Appendix I:

**Theorem 1** (Sample complexity). *Under Assumption 1,2,3, for any $\delta \in (0, 1], \epsilon > 0$, if we choose*

$$T = 1/\epsilon^2 \times \text{poly}(d_{\text{PSR}}, |\mathcal{U}_A|, 1/\alpha, \log \mathcal{N}_{\mathcal{F}}(\epsilon_{\text{b}}), H, |\mathcal{A}|, \log |\mathcal{O}|, \log(1/\delta)),$$

*and set $\beta = c \log(\mathcal{N}_{\mathcal{F}}(\epsilon_{\text{b}}) T H |\mathcal{U}_A|/\delta)$, then with probability at least $1 - \delta$ we have*

$$V^{\widehat{\pi}} \geq V^* - \epsilon,$$

*where $\widehat{\pi}$ is the uniform mixture of the output policies, i.e., $\widehat{\pi} = \text{Unif}(\{\pi^k\}_{k=1}^T)$.*

Theorem 1 indicates that the complexity of CRANE only depends polynomially on the PSR rank $d_{\text{PSR}}$, the size of $\mathcal{U}_{A,h}$, the $\ell_1$-norm of the pseudoinverse of the core matrix $\frac{1}{\alpha}$, the log bracket number of function classes $\log \mathcal{N}_{\mathcal{F}}(\epsilon_{\text{b}})$, $H$ and $|\mathcal{A}|$. CRANE avoids direct dependency on $\text{poly}(|\mathcal{O}|)$ and our sample complexity remains the same even if the observation parts in core test set $\mathcal{U}_h$ is large. Via the relationship between POMDPs and PSRs, Theorem 1 can be applied to $m$-step weakly-revealing tabular POMDPs (including undercomplete POMDPs [29] and overcomplete POMDPs [37]), $m$-step weakly-revealing low-rank POMDPs [47], $m$-step weakly-revealing linear POMDPs and $m$-step decodable POMDPs [13], which we will elaborate on in Section 5 and Appendix A. Our sample complexity in Theorem 1 depends on the upper bound of $\|K_h^\dagger\|_{1 \mapsto 1}$, i.e., $1/\alpha$, which is not avoidable in worst-case. We state the lower bound formally in Appendix K.

**Proof techniques of Theorem 1.** The existing analysis for POMDPs does not apply to PSRs since we do not assume latent states in PSRs, let alone the emission matrix and transition matrix. In our proof, we utilize the linear nature of PSRs and leverage the core matrix $K_h$ to bound the error propagation induced by the product of predictive operator matrices, i.e., $\prod_{h=1}^{H-1} M_{o_h, a_h, h}$ in (3). This key step enables us to bound the difference of model dynamics $\mathbb{P}_f^\pi$ and $\mathbb{P}^\pi$ (i.e., $\mathbb{P}_{f^*}^\pi$) by the estimation error of $M_{o_h, a_h, h}$ and $q_0$, and thus obtain a polynomial bound on the total suboptimality.

**Comparison with existing works on PSRs.** As far as we know, there are only two works that tackle provably efficient RL for PSRs. Jiang et al. [26] shows a polynomial sample complexity result in reactive POMDPs where optimal value functions only depend on current observations. Later, Uehara et al. [45] shows a favorable sample complexity result without this assumption. However, their result is an agnostic-type result that depends on $(|\mathcal{O}||\mathcal{A}|)^M$ when competing with $M$-memory policies. Thus, to compete with the globally optimal policy, their results do *not* imply a polynomial sample complexity bound.

## 5 EXAMPLES

In this section, we illustrate the sample complexity of CRANE to learn $m$-step weakly-revealing/decodable tabular POMDPs and low-rank POMDPs. We defer more details (including the concrete function classes we utilize to satisfy Assumption 3 and comparison with existing works) and other examples including tabular PSRs and $m$-step weakly-revealing/decodable linear POMDPs to Appendix A. Note that we can identify the minimum core test size and the bracketing number of related models, which is summarized in Table 2 and the proof is deferred to Appendix E and G.

### 5.1 $m$-STEP WEAKLY-REVEALING TABULAR POMDPS

We first focus on $m$-step weakly-revealing tabular POMDPs [37] defined as follows.

**Definition 3** ($m$-step weakly-revealing Tabular POMDPs). *Define the $m$-step emission matrix $\mathbb{O}_{h,m} \in \mathbb{R}^{|\mathcal{A}|^{m-1}|\mathcal{O}|^m \times |\mathcal{S}|}$ for any $h \in [H - m + 1]$ as follows:*

$$(\mathbb{O}_{h,m})_{(\boldsymbol{a}, \boldsymbol{o}), s} := \mathbb{P}(o_{h:h+m-1} = \boldsymbol{o} | s_h = s, a_{h:h+m-2} = \boldsymbol{a}), \forall (\boldsymbol{a}, \boldsymbol{o}) \in \mathcal{A}^{m-1} \times \mathcal{O}^m, s \in \mathcal{S}.$$

*When $\text{rank}(\mathbb{O}_{h,m}) = |\mathcal{S}|$, POMDPs are referred as $m$-step weakly-revealing POMDPs.*

This assumption implies that the observations leak at least some information about the states so that we can learn the POMDPs efficiently. Substituting the results in Table 2 into Theorem 1, we can obtain the sample complexity for learning $m$-step weakly-revealing tabular POMDPs as follows.

**Corollary 1** (Sample complexity for $m$-step weakly-revealing tabular POMDPs). *Suppose the POMDP is $m$-step weakly-revealing and we execute CRANE with $\beta = c\log(\mathcal{N}_\mathcal{F}(\epsilon_b)TH|\mathcal{U}_A|/\delta)$ up to the step $H - m$ where $\mathcal{U}_h$ and $\mathcal{N}_\mathcal{F}(\epsilon_b)$ are specified in Table 2. Then under Assumption 1, for any $\delta \in (0, 1], \epsilon > 0$, if we choose*

$$T = 1/\epsilon^2 \times \mathrm{poly}(d_\mathrm{PSR}, |\mathcal{A}|^m, 1/\alpha, |\mathcal{O}|, |\mathcal{S}|, H, \log(1/\delta)),$$

*then with probability at least $1 - \delta$ we have $V^{\widehat{\pi}} \geq V^* - \epsilon$.*

### 5.2 $m$-STEP WEAKLY-REVEALING LOW-RANK POMDPS

Next, we consider $m$-step weakly-revealing low-rank POMDPs. We first define low-rank POMDPs to be a special subclass of POMDPs.

**Definition 4** (Low-rank POMDPs). *Suppose the transition kernel $\mathbb{T}_h$ has the following low-rank form for all $h \in [H]$: $\mathbb{T}_h(s'|s, a) = (\psi_h(s'))^\top \phi_h(s, a)$, where $\psi_h : \mathcal{S} \mapsto \mathbb{R}^{d_\mathrm{trans}}$ and $\phi_h : \mathcal{S} \times \mathcal{A} \mapsto \mathbb{R}^{d_\mathrm{trans}}$ are unknown feature vectors. Then, we call these POMDPs as low-rank POMDPs.*

The low-rank structure leads to a smaller minimum core test set size than general POMDPs since we can show $d_\mathrm{PSR} \leq d_\mathrm{trans}$ as in Appendix E. Weakly-revealing low-rank POMDPs are defined as weakly-revealing POMDPs that have this low-rank structure. For the function class $\mathcal{F}$, we let it model the feature vectors, emission matrix and initial state distribution, i.e., $\{\Phi_f, \Psi_f, \mathbb{O}_f, \mu_f\}_{f \in \mathcal{F}}$ supposing $\mathcal{F}$ is finite (the infinite case is deferred to Appendix A). Then substitute the results in Table 2 into Theorem 1 and the sample complexity for learning low-rank POMDPs is as follows:

**Corollary 2** (Sample complexity for $m$-step weakly-revealing low-rank POMDPs). *Suppose low-rank POMDPs are $m$-step weakly-revealing, and we execute CRANE with $\beta = c\log(|\mathcal{F}|TH|\mathcal{U}_A|/\delta)$ up to the step $H - m$ where $\mathcal{U}_h$ is specified in Table 2. Then under Assumption 1 and 4, for any $\delta \in (0, 1], \epsilon > 0$, if we choose*

$$T = 1/\epsilon^2 \times \mathrm{poly}(d_\mathrm{trans}, |\mathcal{A}|^m, 1/\alpha, \log|\mathcal{F}|, H, \log|\mathcal{O}|, \log(1/\delta)),$$

*then with probability at least $1 - \delta$ we have $V^{\widehat{\pi}} \geq V^* - \epsilon$.*

Notice that the sample complexity only depends on $d_\mathrm{trans}$ rather than $|\mathcal{S}|$ for low-rank POMDPs.

### 5.3 $m$-STEP DECODABLE TABULAR/LOW-RANK POMDPS

Next, we instantiate our result on $m$-step decodable POMDPs [13] defined as follows.

**Definition 5** ($m$-step decodable POMDPs). *There exist unknown decoders $\{\phi_{\mathrm{dec},h}\}_{m \leq h \leq H}$ such that for every reachable trajctory $\tau_H$, we have $s_h = \phi_{\mathrm{dec},h}(z_h)$ for all $m \leq h \leq H$ where $z_h = ((o, a)_{h-m+1:h-1}, o_h)$.*

This definition says that we can decode the current state with $m$-step history. Surprisingly, Table 2 shows that $m$-step decodable POMDPs can be formulated as PSRs just like weakly-revealing POMDPs, which leads to the following corollary:

**Corollary 3** (Sample complexity for $m$-step decodable POMDPs)**.**

- *In $m$-step decodable tabular POMDPs, the same statement in Corollary 1 holds.*

- *In $m$-step decodable low-rank POMDPs, the same statement in Corollary 4 holds.*

**Remark 2** (Tabular PSRs and linear POMDPs). *We can instantiate our result on tabular PSRs. The sample complexity is polynomial in all parameters: $d_\mathrm{PSR}, |\mathcal{O}|, |\mathcal{A}|, |\mathcal{U}|, H, 1/\epsilon, 1/\alpha, \log(1/\delta)$. Here, we leverage the observation of covering numbers after Definition 2. We also consider linear POMDPs where latent transitions and emissions follow linear structures. While similar models are considered [7, 47], our result is still new in that our model is more general. The details are deferred to Section A.*

## 6 CONCLUSION

We consider PAC learning in PSRs that represent states as a vector of prediction about future events conditioned on histories. We propose CRANE and show polynomial sample complexities when we compete with the globally optimal policy. Our work is the first work attaining this goal. Since PSRs are more general than POMDPs, we instantiate our result to several concrete POMDPs such as $m$-step weakly-revealing POMDPs, $m$-step decodable POMDPs, POMDPs with latent low-lank transition. Notably, our work is the first work that simultaneously achieves polynomial sample complexities in $m$-step weakly-revealing POMDPs and $m$-step decodable POMDPs.

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

# A  Examples: More Details and Models

In this section, we supplement the details in Section 5 and illustrate the sample complexity of CRANE to learn tabular PSRs and several other POMDPs in comparison with existing algorithms. We consider two types of POMDPs: $m$-step weakly-revealing POMDPs and $m$-step decodable POMDPs. Assumptions like weakly-revealing property and decodability allow us to identify core test sets.

## A.1  $m$-Step Weakly-Revealing Tabular POMDPs

We first give more details about the discussions of $m$-step weakly-revealing tabular POMDPs.

**Core test sets and function classes.** In this case we can choose $\mathcal{U}_h$ to be the set of all $m$-step futures $(\mathcal{O} \times \mathcal{A})^{m-1} \times \mathcal{O}$. For the function class $\mathcal{F}$, we first let it model the parameters $\{\mathbb{T}_{h;f}, \mathbb{O}_{h;f}, \mu_{1;f}\}_{h \in [H]}$ directly, lift weakly-revealing POMDPs to PSR formulation, and then pre-process it to satisfy Assumption 3. The corresponding $\epsilon$-bracket number $\mathcal{N}_{\mathcal{F}}(\epsilon)$ is shown in Table 2. Besides, since now the core test set is $(\mathcal{O} \times \mathcal{A})^{m-1} \times \mathcal{O}$, we let $m_{(o_{H-m+1:H}, a_{H-m+1:H-1}), H-m+1; f} = e_{(o_{H-m+1:H}, a_{H-m+1:H-1}), H-m+1}$, i.e., it is a one-hot vector which indexes the future $(o_{H-m+1:H}, a_{H-m+1:H-1})$ in $\mathcal{U}_{H-m+1}$. Then from Lemma 4 we know for any trajecotry $\tau_H$,

$$\mathbb{P}_f^{\pi}(\tau_H) = e_{(o_{H-m+1:H}, a_{H-m+1:H-1}), H-m+1} \cdot \prod_{l=1}^{H-m} M_{o_l, a_l, l} q_0 \times \pi(\tau_H).$$

Note that here $\mathcal{U}_H$ does not contain the observation space. Nevertheless, we can replace (17) with

$$V_{f^k}^{\pi^k} - V^{\pi^k} \le H \sum_{\tau_{H-m}} \left\| \prod_{h=1}^{H-m} M_{o_h, a_h, h}^k \cdot q_0^k - \prod_{h=1}^{H-m} M_{o_h, a_h, h} \cdot q_0 \right\|_1 \times \pi^k(\tau_{H-m}),$$

and follow the same proof to show that Theorem 1 still holds even though Assumption 2 is not satisfied.

**Remarks about Corollary 1.** By executing CRANE to the step $H - m$ we mean that when collecting trajectories, we only execute $\pi^{k, u_{a,h+1}, h}$ and collect $\tau_H^{k, u_{a,h+1}, h}$ for $h \in [H - m]^+, u_{a,h+1} \in \mathcal{U}_{A,h+1}$. Since we have $d_{\mathrm{PSR}} \le |\mathcal{S}|$, we can obtain that the sample compleixty will not be larger than $\mathrm{poly}(|\mathcal{S}|, H, |\mathcal{A}|^m, 1/\alpha, 1/\epsilon, |\mathcal{O}|, \log(1/\delta))$ from Corollary 1. This indicates that CRANE is able to achieve polynomial sample complexity for $m$-step weakly-revealing tabular POMDPs.

**Comparison with [37].** In $m$-step weakly revealing tabular POMDPs, CRANE is similar to the algorithm OMLE proposed in [37] and their analysis leads to a sample complexity similar to Corollary 1. However, their algorithm has a pre-processing step on the emission matrix $\mathbb{O}_{h,m}$ while we have a step to formulate POMDPs into PSRs for pre-processing, thus the algorithm is still different. Further, they assume an upper bound on $\|\mathbb{O}_{h,m}^{\dagger}\|_{1\mapsto 1}$ while we assume $\|K_h^{\dagger}\|_{1\mapsto 1} \le 1/\alpha$ in Assumption 1. For tabular POMDPs, our assumption is slightly stronger since we have $K_{h-1} = \mathbb{O}_{h,m} \boldsymbol{S}_h$ where $(\boldsymbol{S})_{s, \tau_{h-1}^l} = \mathbb{P}(s | \tau_{h-1}^l)$ and thus $\sigma_{\min}(K_{h-1}) \le d_{\mathrm{PSR}} \sigma_{\min}(\mathbb{O}_{h,m})$. That said, the analysis and algorithm in [37] is specially tailored to $m$-step weakly reavling POMDPs and relies on the existence of latent states. In contrast, our algorithm and analysis can be applied to any PSR models including $m$-step decodable POMDPs.

**Comparison with [34].** [34] deals with latent MDPs but they require either proper initialization or other assumptions including Sufficient Tests, Sufficient Histories, strong separation of the MDPs and reachability of the states. In contrast, we show in Appendix E that LMDP with Sufficient Tests can be formulated into a $(l + 1)$-step weakly-revealing POMDP, therefore CRANE is capable of tackling LMDP with sample complexity $1/\epsilon^2 \times \mathrm{poly}(M, |\mathcal{S}|, |\mathcal{A}|^l, 1/\alpha, H, \log(1/\delta))$ under only Sufficient Tests and Assumption 1. In addition, the sample complexity in [34, Theorem 3.5] will scale with the initialization error while CRANE circumvents such dependence completely.

### A.2  $m$-STEP WEAKLY-REVEALING LOW-RANK POMDPS

Next, we supplement the details about $m$-step weakly-revealing low-rank POMDPs.

**Core test sets and function classes.** For weakly-revealing low-rank POMDPs, we can still choose $\mathcal{U}_h$ to be the set of all $m$-step futures $(\mathcal{O} \times \mathcal{A})^{m-1} \times \mathcal{O}$ due to the weakly-revealing property. For the function class $\mathcal{F}$, we let it model the feature vectors, emission matrix and initial state distribution, i.e., $\{\Phi_f, \Psi_f, \mathbb{O}_f, \mu_f\}_{f \in \mathcal{F}}$ where $\Phi_f : \mathcal{S} \times \mathcal{A} \times [H] \mapsto \mathbb{R}^{d_{\text{trans}}}$, $\Psi_f : \mathcal{S} \times [H] \mapsto \mathbb{R}^{d_{\text{trans}}}$, $\mathbb{O}_f : \mathcal{S} \times \mathcal{O} \times [H] \mapsto [0,1]$, $\mu_f : \mathcal{S} \mapsto [0,1]$ such that

$$\phi_{h;f}(s,a) = \Phi_f(s,a,h), \psi_{h;f}(s) = \Psi_f(s,h), \mathbb{O}_{h;f}(o|s) = \mathbb{O}_f(s,o,h), \mu_{1;f}(s) = \mu_f(s).$$

Here $\mathcal{F}$ can be infinite. Denote the $\ell_\infty$-norm covering number of $\Phi_f, \Psi_f, \mathbb{O}_f, \mu_f$ by $\mathcal{Y}_\Phi(\epsilon), \mathcal{Y}_\Psi(\epsilon), \mathcal{Y}_\mathbb{O}(\epsilon), \mathcal{Y}_\mu(\epsilon)$. Then we have

$$\log \mathcal{N}_\mathcal{F}(\epsilon) \leq \log \mathcal{Y}_\Phi(\epsilon_{\text{LR}}/d_{\text{trans}}) + \log \mathcal{Y}_\Psi(\epsilon_{\text{LR}}/d_{\text{trans}}) + \log \mathcal{Y}_\mathbb{O}(\epsilon_{\text{LR}}) + \log \mathcal{Y}_\mu(\epsilon_{\text{LR}}),$$

where $\epsilon_{\text{LR}} := \mathcal{O}(\epsilon/(|\mathcal{O}|^{H+2}|\mathcal{A}|^H))$. The proof is deferred to Appendix G. To make Assumption 3 hold, we only need to assume the feature vector classes satisfies realizablity:

**Assumption 4.** *Suppose that there exsits $f^* \in \mathcal{F}$ such that for all $s \in \mathcal{S}, a \in \mathcal{A}, h \in [H]$ we have $\phi_h(s,a) = \Phi_{f^*}(s,a,h), \psi_h(s) = \Psi_{f^*}(s,h)$.*

Then we can lift low-rank POMDPs to PSR formulation, and then pre-process it to satisfy Assumption 3.

**Remarks about Corollary 2.** Corollary 2 only considers finite function class $\mathcal{F}$. With the above discussion, we can extend it to infinite function classes as follows:

**Corollary 4** (Sample complexity for $m$-step weakly-revealing low-rank POMDPs). *Suppose low-rank POMDPs are $m$-step weakly-revealing, and we execute CRANE with $\beta = c \log(\mathcal{N}_\mathcal{F}(\epsilon_{\text{b}})TH|\mathcal{U}_A|/\delta)$ up to the step $H - m$ where $\mathcal{U}_h$ and $\mathcal{N}_\mathcal{F}(\epsilon_{\text{b}})$ are specified in Table 2. Then under Assumption 1 and 4, for any $\delta \in (0,1], \epsilon > 0$, if we choose*

$$T = 1/\epsilon^2 \times \text{poly}(d_{\text{trans}}, |\mathcal{A}|^m, 1/\alpha, \log \mathcal{Y}_\Phi(\epsilon_{\text{LR}}/d_{\text{trans}}), \log \mathcal{Y}_\Psi(\epsilon_{\text{LR}}/d_{\text{trans}}),$$
$$\log \mathcal{Y}_\mathbb{O}(\epsilon_{\text{LR}}), \log \mathcal{Y}_\mu(\epsilon_{\text{LR}}), H, \log |\mathcal{O}|, \log(1/\delta)),$$

*then with probability at least $1 - \delta$ we have $V^{\widehat{\pi}} \geq V^* - \epsilon$.*

**Comparison with [47].** In Corollary 4, we do not specify the function class and keep the bracket number to facilitate the comparison with [47]. [47] also tackles the online learning problem of $m$-step weakly-revealing low-rank POMDPs and our sample complexity only has an additional $\log |\mathcal{O}|$ factor compared to theirs. However, the model they have considered is less general than ours in the sense that they require the feature vectors $\phi_h(s,a)$ to be a $d_{\text{trans}}$-dimensional probability distribution to guarantee the existence of some bottleneck variables. Besides, their analysis depends on some possibly complicated assumptions to recover the bottleneck variable like "*past sufficiency*". In contrast, CRANE only requires $\|K_h^\dagger\|_{1 \mapsto 1}$ to be upper bounded and does not assume the existence of bottleneck variables.

**Comparsion with [45].** They show favorable sample complexity results in weakly-revealing low-rank POMDPs. However, their sample complexity results are quasi-polynomial. On the other hand, our results are polynomial while we have an additional $\log |\mathcal{O}|$ factor.

### A.3  TABULAR PSRS

Notice that in Theorem 1 the log bracket number $\log \mathcal{N}_\mathcal{F}(\epsilon_{\text{b}})$ is abstract. Here we consider tabular PSRs as a speical case to provide an intuition how large the bracket number will be in general. In tabular PSRs we directly use $\{M_{o,a,h}, q_0\}_{o \in \mathcal{O}, a \in \mathcal{A}, h \in [H-1]}$ as the parameters of $\mathcal{F}$ and assume for all $f \in \mathcal{F}$ we have

$$\max_{o \in \mathcal{O}, a \in \mathcal{A}, h \in [H-1], u \in \mathcal{U}_{h+1}} \|m_{(o,a,u),h;f}\|_\infty \leq 1, \|q_{0;f}\|_\infty \leq 1$$

Then following the arguments in Appendix G, we have

$$\log \mathcal{N}_{\mathcal{F}}(\epsilon) \leq \mathcal{O}(|\mathcal{U}|^2 |\mathcal{O}||\mathcal{A}|H^2 \log(H|\mathcal{O}||\mathcal{A}||\mathcal{U}_A||\mathcal{U}|/(\alpha\epsilon))). \qquad (6)$$

By substituting the above results into Theorem 1, we have the following corollary which characterizes the sample complexity to learn tabular PSRs:

**Corollary 5** (Sample complexity for tabular PSRs). *Execute* CRANE *with* $\beta = c\log(\mathcal{N}_{\mathcal{F}}(\epsilon_{\mathrm{b}})TH|\mathcal{U}_A|/\delta)$ *where* $\mathcal{N}_{\mathcal{F}}(\epsilon_{\mathrm{b}})$ *are specified in* (6). *Then under Assumption 1,2 and 3, for any $\delta \in (0,1], \epsilon > 0$, if we choose*

$$T = 1/\epsilon^2 \times \mathrm{poly}(d_{\mathrm{PSR}}, |\mathcal{U}_A|, 1/\alpha, |\mathcal{U}|, |\mathcal{O}|, |\mathcal{A}|, H, \log(1/\delta)),$$

*then with probability at least $1 - \delta$ we have*

$$V^{\widehat{\pi}} \geq V^* - \epsilon.$$

From Corollary 5 we can see that CRANE is capable of learning tabular PSRs efficiently, with sample complexity polynomial in all relevant parameters. Here, though we have a $\mathrm{poly}(|\mathcal{U}|)$ dependency in learning PSRs (since our model parameters $M_{a,o}$ have degree of freedom scaling in $\mathrm{poly}(|\mathcal{U}|)$), we will show that we would not incur $\mathrm{poly}(|\mathcal{U}|)$ in the log $\epsilon$-bracket number when PSRs are POMDPs. This is because we can directly model the latent transition and omission distribution when we know it is a POMDP.

### A.4 $m$-STEP WEAKLY-REVEALING LINEAR POMDPS

In low-rank POMDPs the bracket number is still somehow abstract because we do not specify the function class $\{\Phi_f, \Psi_f, \mathbb{O}_f, \mu_f\}$. Next we consider linear POMDPs and illustrate a more concrete result. Here we assume that linear POMDPs possess a linear structure in both the transition kernel and emission matrix. More formally, we can generalize the linear MDPs in [48] and define linear POMDPs as follows:

**Definition 6** (Linear POMDPs). *A POMDP is linear with respect to the given feature vectors* $\{\phi_h(s,a) \in \mathbb{R}^{d_1}, \psi_h(s) \in \mathbb{R}^{d_2}, \overline{\phi}_h(s) \in \mathbb{R}^{d_3}, \overline{\psi}_h(o) \in \mathbb{R}^{d_4}, \widehat{\phi}(s) \in \mathbb{R}^{d_5}\}_{s\in\mathcal{S},a\in\mathcal{A},o\in\mathcal{O},h\in[H]}$ *where* $\|\phi_h(s,a)\|_\infty \leq 1, \|\psi_h(s)\|_\infty \leq 1, \|\overline{\phi}_h(s,a)\|_\infty \leq 1, \|\overline{\phi}_h(o)\|_\infty \leq 1, \|\widehat{\phi}(s)\|_\infty \leq 1$ *for all* $s \in \mathcal{S}, a \in \mathcal{A}, o \in \mathcal{O}, h \in [H]$ *if there exists a set of matrices* $\{B^*_{h,1} \in \mathbb{R}^{d_2\times d_1}, B^*_{h,2} \in \mathbb{R}^{d_4\times d_3}\}_{h\in[H]}$ *where* $\|B^*_{h,1}\|_{\infty,\infty} \leq 1, \|B^*_{h,2}\|_{\infty,\infty} \leq 1$ *for all $h \in [H]$ and $\theta^* \in \mathbb{R}^{d_5}$ where $\|\theta^*\|_\infty \leq 1$ such that for any $s, s' \in \mathcal{S}, a \in \mathcal{A}, o \in \mathcal{A}, h \in [H]$ we have*

$$\mathbb{T}_h(s'|s,a) = (\psi_h(s'))^\top B^*_{h,1}\phi_h(s,a), \mathbb{O}(o|s) = (\overline{\psi}_h(o))^\top B^*_{h,2}\overline{\phi}_h(s), \mu_1(s) = (\theta^*)^\top \widehat{\phi}(s).$$

Denote $d_{\mathrm{lin}} = \max\{d_1, d_2, d_3, d_4, d_5\}$. Notice that since $\mathbb{T}_h(s'|s,a) = (\psi_h(s'))^\top B^*_{h,1}\phi_h(s,a)$, linear POMDPs are also low-rank POMDPs with dimension $\min\{d_1, d_2\}$ and thus for linear POMDPs we have

$$d_{\mathrm{PSR}} \leq d_{\mathrm{lin}}.$$

We further define the function class to be $\{f = (B_{h,1} \in \mathbb{R}^{d_2\times d_1}, B_{h,2} \in \mathbb{R}^{d_4\times d_3}, \theta \in \mathbb{R}^{d_5}) : \|B_{h,1}\|_{\infty,\infty} \leq 1, \|B_{h,2}\|_{\infty,\infty} \leq 1, \|\theta\|_\infty \leq 1, \forall h \in [H]\}$ such that for any $o \in \mathcal{O}, a \in \mathcal{A}, h \in [H - m]$

$$\mathbb{T}_{h;f}(s'|s,a) = (\psi_h(s'))^\top B_{h,1}\phi_h(s,a), \mathbb{O}_{h;f}(o|s) = (\overline{\psi}_h(o))^\top B_{h,2}\overline{\phi}_h(s), \mu_{1;f}(s) = \theta^\top \widehat{\phi}(s).$$

This enables us to bound $\mathcal{N}_{\mathcal{F}}(\epsilon)$ as in Table 2. Note that this function class satisfies realizability and we can pre-process it to make Assumption 3 hold.

Finally, we assume $m$-step weakly-revealing property. In this case, we can still choose the same $\mathcal{U}_h$ as in tabular POMDPs. Using the above models and formulating POMDPS into PSRS for the pre-processing step to satisfy Assumption 3, we can run CRANE. The sample complexity for learning linear POMDPs will scale with $d_{\mathrm{lin}}$ rather than $\mathrm{poly}(|\mathcal{O}|, |\mathcal{S}|)$ as follows.

**Corollary 6** (Sample complexity of $m$-step weakly-revealing linear POMDPs). *Suppose the linear POMDP is $m$-step weakly-revealing and we execute* CRANE *with $\beta = c \log(\mathcal{N}_{\mathcal{F}}(\epsilon_{\mathrm{b}}) T H |\mathcal{U}_A|/\delta)$ up to the step $H - m$ where $\mathcal{U}_h$ and $\mathcal{N}_{\mathcal{F}}(\epsilon_{\mathrm{b}})$ are specified in Table 2. Then under Assumption 1, for any $\delta \in (0,1], \epsilon > 0$, if we choose*

$$T = 1/\epsilon^2 \times \mathrm{poly}(d_{\mathrm{lin}}, |\mathcal{A}|^m, 1/\alpha, H, \log |\mathcal{O}|, \log(1/\delta)),$$

*then with probability at least $1 - \delta$ we have*

$$V^{\widehat{\pi}} \geq V^* - \epsilon.$$

**Comparison with [7].** From Corollary 6, we can see that the linear structure helps us circumvent the polynomial scaling with $|\mathcal{O}|$ and $|\mathcal{S}|$. [7] also discusses linear POMDPs and achieves a similar polynomial sample complexity. However, they only consider undercomplete setting (i.e., $|\mathcal{O}| \geq |\mathcal{S}|$) and assume $\{(\psi_h(s))_i\}_{s \in \mathcal{S}}$ is a distribution on $\mathcal{S}$ for any $i \in [d_2]$. In addition, they not only assume the transition and emission are linear, but also impose a linear structure on the state distribution conditioned on future observations such as Cai et al. [7, Assumption 2.2]. Therefore our model is more general and requires fewer assumptions.

### A.5 $m$-STEP DECODABLE TABULAR/LOW-RANK/LINEAR POMDPS

Next, we supplement the discussion about $m$-step decodable POMDPs in Section 5.

**Core test sets and function classes.** Like $m$-step weakly-revealing POMDPs, $m$-step decodable POMDPs can be formulated as PSRs where core tests are $m$-step futures and the PSR rank is $|\mathcal{S}|$ in the tabular case and $d_{\mathrm{trans}}$ in low-rank POMDPs. Intuitively, this is proved by the observation that $m$-step futures can decode the latent state $m$-step ahead, i.e., $s_{m+h}$ by treating "histories" in the definition as "futures". In Appendix E we have a more detailed discussion. We also utilize the same function classes as $m$-step weakly-revealing POMDPs for $m$-step decodable POMDPs.

**Remarks about Corollary 3.** Note that the discussion about $m$-step weakly-revealing linear POMDPs also holds for $m$-step decodable POMDPs, therefore we can extend Corollary 3 to the following corollary:

**Corollary 7** (Sample comlexity for $m$-step decodable POMDPs).

* *In $m$-step decodable tabular POMDPs, the same statement in Corollary 1 holds.*

* *In $m$-step decodable low-rank POMDPs, the same statement in Corollary 4 holds.*

* *In $m$-step decodable linear POMDPs, the same statement in Corollary 6 holds.*

**Comparison with [13].** [13] works on $m$-step decodable tabular POMDPs and show sample complexity polynomial in $|\mathcal{S}|, H, |\mathcal{A}|^m, 1/\epsilon, \log(1/\delta)$ and log covering number of a value function class. They also provide a result on $m$-step decodable low-rank POMDPs where the sample complexity scales with $d_{\mathrm{trans}}$ rather than $|\mathcal{S}|$. However, there are some differences between their results and Corollary 7. First, the log covering number of the value function class in their results will typically scale with $\mathrm{poly}(|\mathcal{O}|^m)$. Our results, on the other hand, only scale with $\mathrm{poly}(|\mathcal{O}|)$ since the log bracket number of our function classes only scales with $\mathrm{poly}(|\mathcal{O}|)$. Secondly, the analysis in [13] does not require the regularity-type assumption (Assumption 1). This is because their algorithm is tailored to $m$-step decodable POMDPs. The lower bound in Theorem 3 has shown that the scaling with the regularity parameter $1/\alpha$ is inevitable in PSRs, highlighting the necessity of such regularity in general.

## B  RELATED WORKS

We discuss related works to our paper in this section.

**PSRs and its learning algorithm**  PSRs represent states as a vector of predictions about future events [36, 41, 38, 21, 44, 19]. Importantly, compared to well-known models of dynamical systems like HMMs that postulate latent state variables that are never observed, we do not need to refer to latent state variables and every definition relies on observable quantities. While PSRs were originally introduced in the tabular setting, PSRs can be extended to the non-tabular setting using conditional mean embeddings [4]. Using data obtained by exploratory open-loop policies such as uniform policies, Boots et al. [6; 4], Zhang et al. [49] proposed a learning algorithm for dynamics by leveraging spectral learning [33, 23, 28]. Later, Hefny et al. [22] pointed out an insightful connection between spectral learning and supervised learning (more specifically, instrumental variable regression when histories are instrumental). Based on this viewpoint, Hefny et al. [22] proposed a two-stage regression learning algorithm. Compared to these settings, our setting is significantly challenging. This is because their goal is learning system dynamics with exploratory offline data while we want to learn the optimal policy when we don't have access to such exploratory data.

**Provably efficient RL for POMDPs and PSRs.**  Seminal works [31, 14] obtained $A^H$-type sample complexity bounds for POMDPs. We can avoid exponential dependence with more structural assumptions. Recently, there is a growing body of literature that discusses provably efficient RL in the online setting under various structures.

In the tabular setting, one of the most standard structural assumptions is an observability (i.e., weakly-revealing) assumption, which implies that observations retain information about latent states. Under observability and various additional assumptions, in Azizzadenesheli et al. [3], Guo et al. [20], Kwon et al. [34], favorable polynomial sample complexities are obtained by leveraging the spectral learning technique [23]. Later, Jin et al. [29], Liu et al. [37] improve these results and obtain polynomial sample complexity results under only observability assumptions. Golowich et al. [17; 18] develop algorithms with quasi-polynomial sample and computational complexity under observability properties.

In the non-tabular POMDP setting, several positive results are obtained. One of the most investigated models is linear quadratic gaussian (LQG), which is a partial observable version of LQRs. Lale et al. [35], Simchowitz et al. [40] proposed sub-linear regret algorithms. Polynomial sample complexities are obtained on other various POMDP models such as M-step decodable POMDPs [13] where we can decode the latent state by $m$-step back histories (when $m = 1$, it is Block MDP), weakly-revealing linear-mixture type POMDPs [7] where emission and transition are modeled by linear mixture models, weakly-revealing low-rank POMDPs [45] where latent transition have low-rank structures. Our proposed algorithm can capture all of the abovementioned models except for LQG.

There are few works that discuss strategic exploration in PSRs. None of them obtain polynomial sample complexity results for learning approximate globally optimal policies [26, 45]. For details, refer to Section 4.

## C  NOTATIONS

We sum up the notations in PSRs in Table 3.

## D  EXPRESSIVITY OF PSRS

In this section, we will construct a sequential decision making process to illustrate the superior expressivity of PSRs with respect to POMDPs. In short, we will show that if we formulate the process into a POMDP, the number of latent states we need can be exponentially larger the core test set size in PSRs. The construction leverages existing results in perfect matching polytope and largely follows the arguments in [1].

First, let $n$ be even and $K_n$ be the complete graph on $n$ vertices. Consider a vector $x \in \mathbb{R}^{\binom{n}{2}}$ that associates a weight to each edge and we denote its entry by $x_{u,v}$ where $u \neq v \in [n]$ are the vertices. Let $\mathbf{1}_{\mathcal{M}} \in \mathbb{R}^{\binom{n}{2}}$ denote the edge-indicator vector for a subset of edges $\mathcal{M}$. Then [12] shows that the convex hull of all edge-indicator vectors corresponding to a perfect match, which we also call the

Table 3: Notations of PSRs. We also refer readers to Figure 1 for an illustration of the notations such as $\mathcal{U}_h, \overline{D}_h, D_h$, and $K_h$.

| Notation | Definition |
|---|---|
| $V^\pi$ | $\mathbb{E}_\pi \left[ \sum_{h=1}^H r_h \right]$ |
| $\mathbb{P}(t_h\|\tau_{h-1})$ | $\mathbb{P}(o_{h:h+W-1}\|\tau_{h-1}; \mathrm{do}(a_{h:h+W-2}))$ |
| $\mathcal{U}_h$ | core test set at step $h$ |
| $\mathcal{U}_{A,h}$ | the set of all action sequences in $\mathcal{U}_h$ |
| $q_{\tau_h}$ | predictive states $[\mathbb{P}(u\|\tau_h)]_{u \in \mathcal{U}_{h+1}}$ |
| $m_{t_h,h}$ | $\mathbb{P}(t_h\|\tau_{h-1}) = \langle m_{t_h,h}, q_{\tau_{h-1}} \rangle$ |
| $d_{\mathrm{PSR},h}$ | minimum core test set size at step $h$ |
| $b_{\tau_h}$ | unnormalized predictive state $\{\prod_{l=1}^h M_{o_l,a_l,l}\}q_0$ |
| $D_h$ | system dynamics |
| $\overline{D}_h$ | predictive state dynamics whose columns are $q_{\tau_h}$ |
| $K_h$ | core matrix at step $h$ |
| $M_{o,a,h}$ | predictive operator matrix |

perfect matching polytope, can be expressed with a number of constraints as follows:

$$\mathcal{P}_n := \mathrm{conv}\left\{ \mathbf{1}_{\mathcal{M}} \in \mathbb{R}^{\binom{n}{2}} | \mathcal{M} \text{ is a perfect matching in } K_n \right\}$$

$$= \left\{ x \in \mathbb{R}^{\binom{n}{2}} : x \geq 0; \forall v : \sum_u x_{u,v} = 1; \forall U \subset [n] \text{ and } |U| \text{ is odd} : \sum_{v \notin U} \sum_{u \in U} x_{u,v} \geq 1 \right\}.$$

There are $V := n!/(2^{n/2}(n/2)!)$ vertices in $\mathcal{P}_n$ and the number of constraints is $C := 2^{\Omega(n)}$. We denote the vertices by $\{v_1, \cdots, v_V\}$ and the constraints by $c_1, \cdots, c_C$. We further add another dimension to $v_i (i \in [V])$ to account for the offsets in the constraints and obtain vectors $v_i' \in \mathbb{R}^{\binom{n}{2}+1}(i \in [V])$. Then we have $\langle c_i, v_j' \rangle \geq 0$ for all $i \in [C], j \in [V]$. Now we can define the slack matrix for the polytope $\mathcal{P}_n$ to be $Z \in \mathbb{R}_+^{C \times V}$ where $Z_{i,j} = \langle c_i, v_j' \rangle$.

Notice that the rank of $Z$ is $\mathcal{O}(n^2)$. However, since $\mathcal{P}_n$ has extension complexity $2^{\Omega(n)}$ [39] and the extension complexity of a polytipe is the non-negative rank of its slack matrix [15], we know the non-negative rank of $Z$ is at least $2^{\Omega(n)}$.

Now we can construct our sequential decision making process. Suppose for the step $1 \leq h \leq H-1$, the process behaves according to a POMDP with state space $\mathcal{S}'$, action space $\mathcal{A}$, observation space $\mathcal{O}$, initial state distribution $\mu_1$, emission matrix $\mathbb{O}_h$ and transition kernel $\mathbb{T}_h$. At step $h = H$ though, the one-step system dynamics $D_{H-1}$ is given by associating each pair $(o_{H-1}, a_{H-1})$ with a constraint $c_i$ and each future test $t \in \mathcal{O}$ (which is one-step observation now) with a vertex $v_j'$ such that for any history $\tau_{H-1}$ that ends with $(o_{H-1}, a_{H-1})$ we have

$$\mathbb{P}(t|\tau_{H-1}) = \frac{\langle c_i, v_j' \rangle}{\sum_{k=1}^V \langle c_i, v_j' \rangle}.$$

Now we fix a history $\tau_{H-2}$ with length $H-2$ and consider the matrix $\hat{D}_{H-1} \in \mathbb{R}^{|\mathcal{O}| \times (|\mathcal{O}||\mathcal{A}|)}$ where the rows are indexed by the test $t \in \mathcal{O}$, the columns are indexed by the history $(\tau_{H-2}, o, a)$ for all $o \in \mathcal{O}, a \in \mathcal{A}$ and $(\hat{D}_{H-1})_{t,(\tau_{H-2},o,a)} = \mathbb{P}(t|\tau_{H-2}, o, a)$. Since the non-negative rank is preserved under positive diagonal rescaling [8], we know the non-negative rank of $\hat{D}_{H-1}^\top$ is at least $2^{\Omega(n)}$. Then for the above sequential process, if we formulate it into a POMDP with state space $\mathcal{S}$, then we have

$$\mathbb{P}(t|\tau_{H-2}, o, a) = \sum_{s_H \in \mathcal{S}} \mathbb{P}(t|s_H)\mathbb{P}(s_H|\tau_{H-2}, o, a).$$

Notice that for a row-stochastic matrix $\hat{D}_{H-1}^\top$, the non-negative rank is equal to the smallest number of factors we can use to write $\hat{D}_{H-1}^\top = RS$ where both $R, S$ are row-stochastic [8]. This implies

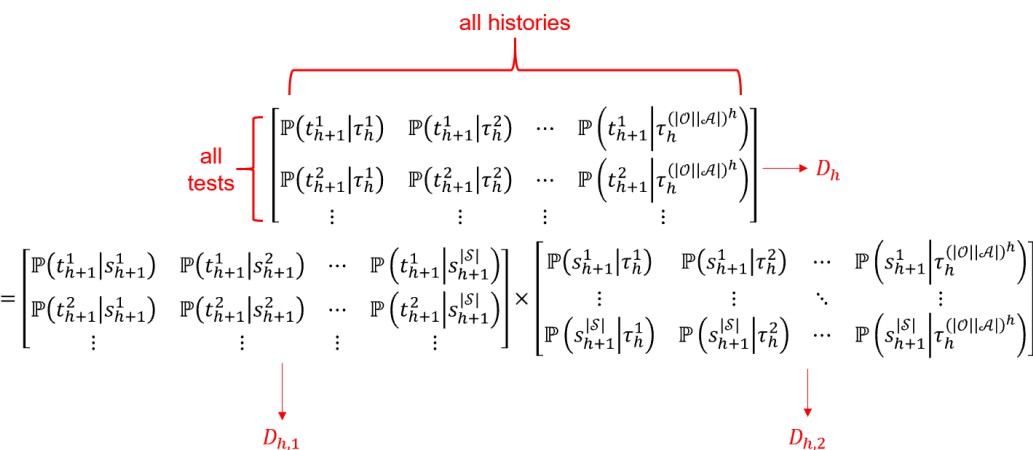

Figure 2: For any POMDP, the system dynamics matrix $D_h$ can always be factorized using the latent states. This factorization implies that the rank of $D_h$ is no larger than the number of latent states, which implies that POMDP is a linear PSR with rank at most equal to the number of latent states. Note that here $D_{h,1}$ and $D_{h,2}$ both contains non-negative entires. In contrast, from Figure 1, the low-rank factorization of $D_h$ in PSR can have negative entries (i.e., $m$ and $v$ can have negative entries).

that we must have $|\mathcal{S}|$ not smaller than the non-negative rank of $\hat{D}_{H-1}^{\top}$, therefore we have

$$|\mathcal{S}| \geq 2^{\Omega(n)}.$$

On the other hand, if we formulate the above process into a PSR, at step $h = H$, since the rank of $D_{H-1}$ is equal to the rank of $Z$, we know the rank of $D_{H-1}$ is not larger than $\mathcal{O}(n^2)$, which implies that there exists a core test set $\mathcal{U}_H$ whose size is not larger than $\mathcal{O}(n^2)$. When $1 \leq h \leq H-1$, notice that for any test $t = \{o_{h:H}, a_{h:H-1}\}$ and history $\tau_{h-1}$ we have

$$\mathbb{P}(t|\tau_{h-1}) = \sum_{s_h \in \mathcal{S}'} \mathbb{P}(s_h|\tau_{h-1})(\mathbb{P}(t_{h:H-1}|s_h)\mathbb{P}(o_H|o_{H-1}, a_{H-1})),$$

where $t_{h:H-1} = \{o_{h:H-1}, a_{h:H-2}\}$. Notice that $\mathbb{P}(t_{h:H-1}|s_h)\mathbb{P}(o_H|o_{H-1}, a_{H-1})$ only depends on $t$ and $\mathbb{P}(s_h|\tau_{h-1})$ only depends on $\tau_{h-1}$. This implies that there exists a core test set $\mathcal{U}_h$ whose size is not larger than $|\mathcal{S}'|$ for all $1 \leq h \leq H-1$. Therefore, the core test set size of the PSR can be smaller than $\max\{\mathcal{O}(n^2), |\mathcal{S}'|\}$. This shows that PSRs can express this sequential decision making process exponentially more efficient than POMDPs.

## E   EXAMPLES OF PSRS

In this section we present the proofs of Lemma 2 and Lemma 3, and then formulate $m$-step weakly-revealing POMDPs, $m$-step decodable POMDPs and low rank POMDPs into PSRs and analyze their core test set and minimum core test set size.

### E.1   PROOFS OF LEMMA 2 AND LEMMA 3

We first prove Lemma 2. Consider the one-step system dynamics $D_h$ shown in Figure 2 whose rows are indexed by all possible future tests $t_{h+1}$ and columns are indexed by all histories $\tau_h$ at $h$-th step. Each entry of $D_h$ is the successful probability of the test, i.e., $(D_h)_{t_{h+1}, \tau_h} = \mathbb{P}(t_{h+1}|\tau_h)$. Since we know $\mathbb{P}(t_{h+1}|\tau_h) = \sum_{s_{h+1} \in \mathcal{S}} \mathbb{P}(t_{h+1}|s_{h+1})\mathbb{P}(s_{h+1}|\tau_h)$ (where we also define $\mathbb{P}(s_{h+1}|\tau_h) = 0$ for unreachable $\tau_h$), we can decompose $D_h$ into the product of $D_{h,1}$ and $D_{h,2}$ as in Figure 2, where $(D_{h,1})_{t_{h+1}, s_{h+1}} = \mathbb{P}(t_{h+1}|s_{h+1})$ and $(D_{h,2})_{s_{h+1}, \tau_h} = \mathbb{P}(s_{h+1}|\tau_h)$. This implies that the rank of $D_h$ is not larger than $|\mathcal{S}|$, which proves that it is a linear PSR with rank no larger than $|\mathcal{S}|$.

Now we prove Lemma 3. Consider any $h \in [H]$, let $q_{\tau_{h-1}} = [\mathbb{P}(o|\tau_{h-1})]_{o \in \mathcal{O}}$. Then the belief state of the POMDP $s_{\tau_{h-1}} = [\mathbb{P}(s_h|\tau_{h-1})]_{s_h \in \mathcal{S}}$ can be expressed as:

$$s_{\tau_{h-1}} = \mathbb{O}_h^{\dagger} q_{\tau_{h-1}}.$$

Here, we use $\mathbb{O}_h^\dagger \mathbb{O}_h$ is an $|\mathcal{S}| \times |\mathcal{S}|$ identity matrix, which is verified by the assumption. Then for any test $t = (o_{h:h+W}, a_{h:h+W-1})$, we know $\mathbb{P}(t|\tau_{h-1}) = m'_{t,h} s_{\tau_{h-1}}$ where

$$m'_{t,h} = \mathbb{O}_{h+W}(o_{h+W}|\cdot)^\top \prod_{l=h}^{h+W-1} \mathbb{T}_{l,a_l} \text{diag}(\mathbb{O}_l(o_l|\cdot)).$$

where $\mathbb{O}_h(o|\cdot) \in \mathbb{R}^{|\mathcal{S}|}$ is a vector whose $s$-th entry is $\mathbb{O}_h(o|s)$ and $\mathbb{T}_{l,a_l}$ is a $|\mathcal{S}| \times |\mathcal{S}|$ matrix with entry $(\mathbb{T}_{l,a_l})_{s',s} = \mathbb{T}_l(s'|s, a_l)$.

Therefore we have $\mathbb{P}(t|\tau_{h-1}) = \langle m_{t,h}, q_{\tau_{h-1}} \rangle$ where $m_{t,h} = (m'_{t,h}\mathbb{O}_h^\dagger)^\top$. Thus we have shown that the probability of any test $t$ is a linear combination of the probabilities of the tests $o \in \mathcal{O}$ (the linear combination weights $m_{t,h}$ only depends on test but is independent of history). This indicates that $\mathcal{O}$ is a core test set for 1-step weakly-revealing POMDPs.

### E.2 $m$-STEP WEAKLY-REVEALING POMDPS

Recall the definition of the $m$-step emission matrix $\mathbb{O}_{h,m} \in \mathbb{R}^{|\mathcal{A}|^{m-1}|\mathcal{O}|^m \times |\mathcal{S}|}$ for any $h \in [H - m + 1]$ is as follows:

$$(\mathbb{O}_{h,m})_{(\boldsymbol{a},\boldsymbol{o}),s} := \mathbb{P}(o_{h:h+m-1} = \boldsymbol{o}|s_h = s, a_{h:h+m-2} = \boldsymbol{a}), \forall (\boldsymbol{a}, \boldsymbol{o}) \in \mathcal{A}^{m-1} \times \mathcal{O}^m, s \in \mathcal{S}.$$

Then $m$-step weakly revealing condition [37] means that the rank of $\mathbb{O}_{h,m}$ is $|\mathcal{S}|$ for all $h \in [H - m + 1]$. From Lemma 2, we know that $d_{\text{PSR}} \le |\mathcal{S}|$. In addition, the following lemma suggests that a general core test set for $m$-step weakly-revealing POMDPs is the set of all $m$-step futures:

**Lemma 5.** *When $\mathbb{O}_{h,m}$ is full rank for all $h \in [H - m + 1]$, the POMDP is a PSR with the core test set $\mathcal{U}_h = \mathcal{O} \times (\mathcal{A} \times \mathcal{O})^{m-1}$ for all $h \in [H - m + 1]$.*

*Proof.* The proof is similar to 1-step weakly-revealing POMDPs. Consider any $h \in [H - m + 1]$, let $q_{\tau_{h-1}} = [\mathbb{P}(u|\tau_{h-1})]_{u \in \mathcal{O} \times (\mathcal{A} \times \mathcal{O})^{m-1}}$. Then the belief state $s_{\tau_{h-1}} = [\mathbb{P}(s_h|\tau_{h-1})]_{s_h \in \mathcal{S}}$ can be expressed as:

$$s_{\tau_{h-1}} = \mathbb{O}_{h,m}^\dagger q_{\tau_{h-1}}.$$

Then we know for any test $t = (o_{h:h+W}, a_{h:h+W-1})$, we know $\mathbb{P}(t|\tau_{h-1}) = m'_{t,h} s_{\tau_{h-1}}$ where

$$m'_{t,h} = \mathbb{O}_{h+W}(o_{h+W}|\cdot)^\top \prod_{l=h}^{h+W-1} \mathbb{T}_{l,a_l} \text{diag}(\mathbb{O}_l(o_l|\cdot)).$$

Recall that here $\mathbb{O}_h(o|\cdot) \in \mathbb{R}^{|\mathcal{S}|}$ is a vector whose $s$-th entry is $\mathbb{O}_h(o|s)$ and $\mathbb{T}_{l,a_l}$ is a $|\mathcal{S}| \times |\mathcal{S}|$ matrix with entry $(\mathbb{T}_{l,a_l})_{s',s} = \mathbb{T}_l(s'|s, a_l)$.

Therefore we have $\mathbb{P}(t|\tau_{h-1}) = \langle m_{t,h}, q_{\tau_{h-1}} \rangle$ where $m_{t,h} = (m'_{t,h}\mathbb{O}_{h,m}^\dagger)^\top$. This indicates that $\mathcal{U}_h = \mathcal{O} \times (\mathcal{A} \times \mathcal{O})^{m-1}$ is a core test set for all $h \in [H - m + 1]$. $\qquad \square$

Notice that here we only show the core test set of $m$-step weakly-revealing POMDPs up to step $H - m + 1$. However, this is sufficient to charaterize the whole POMDP. From Lemma 4 we know that for any trajectory $\tau_H$, $\mathbb{P}^\pi(\tau_H)$ is one of the entries in $\prod_{l=1}^{H-m} M_{o_l,a_l,l} q_0 \times \pi(\tau_H)$. Therefore, with parameters $\{M_{o,a,h}, q_0\}_{o \in \mathcal{O}, a \in \mathcal{A}, h \in [H-m]}$ (which only depends on $\mathcal{U}_h$ where $h \in [H-m+1]$) we can recover the POMDPs easily.

### E.3 LATENT MDPS

Next we consider the Latent MDP (LMDP) model in [34]. Supoose there are $M$ MDPs and each MDP $m$ is characterized by $(\mathcal{S}, \mathcal{A}, \mathbb{T}_{m,h}, R_{m,h}, H, \mu_m)$ where $\mathcal{S}$ is the common state space, $\mathcal{A}$ is the common action space, $\mathbb{T}_{m,h}$ is the transition probability at step $h$ of MDP $m$, $R_{m,h} : \mathcal{S} \times \mathcal{A} \times \{0, 1\} \mapsto [0, 1]$ is a probability measure for rewards at step $h$ of MDP $m$ that maps a state-action pair and a binary reward to a probability, $H$ is the horizon and $\mu_m$ is the initial state distribution of MDP $m$. At the start of every episode, one MDP $m \in [M]$ is randomly chosen with some probability $w_m$.

[34, Theorem 3.1] shows that with no further assumptions, learning an instance of the above LMDP requires at least $\Omega((|\mathcal{S}||\mathcal{A}|)^M)$ episodes at worst. A number of assumptions are considered to circumvent this lower bound and one of them is called *Sufficient Tests*. More specifically, for each step $h \in [H - l + 1]$, consider all possible length-$l$ action-reward-state sequences $a_h, r_h, s_{h+1}, \cdots, a_{h+l-1}, r_{h+l-1}, s_{h+l}$ and denote the set of all such sequences by $\mathcal{T}_h$. Then suppose that the successful probability of $\mathcal{T}_h$ under different MDPs given any $s_h \in \mathcal{S}$ has rank $M$:

**Assumption 5** (Sufficient Tests, [34, Condition 1]). *For any $h \in [H - l + 1], s_h \in \mathcal{S}$ and any $t = (a_h^t, r_h^t, s_{h+1}^t, \cdots s_{h+l}^t) \in \mathcal{T}_h$, we define*

$$\mathbb{P}_m(t|s_h) := \mathbb{P}_m(r_h^t, s_{h+1}^t, \cdots s_{h+l}^t | s_h, \mathrm{do}(a_h^t, \cdots, a_{h+l-1}^t)),$$

*where $\mathbb{P}_m$ denotes the probability under MDP $m$. Let $L_{s_h} = [[\mathbb{P}_1(t|s_h)]_{t\in\mathcal{T}_h}, \cdots, [\mathbb{P}_M(t|s_h)]_{t\in\mathcal{T}_h}]$, then $\sigma_M(L_{s_h}) \geq \alpha$ for all $s_h \in \mathcal{S}$ with some $\alpha > 0$.*

The following lemma indicates that LMDPs with Assumption 5 can be formulated into an $(l + 1)$-step weakly-revealing POMDP and thus a PSR with the core test set being all $(l + 1)$-step futures:

**Lemma 6.** *Under Assumption 5, the LMDP can be formulated into an $(l+1)$-step weakly-revealing POMDP.*

*Proof.* First notice that the LMDP can be formulated into a POMDP with state space $\overline{\mathcal{S}} = \mathcal{S} \times \{0, 1\} \times [M]$ and observation space $\mathcal{O} = \mathcal{S} \times \{0, 1\}$. At each step $h$, the latent state $\overline{s}_h \in \overline{\mathcal{S}}$ is $(s_h, r_{h-1}, I)$ where $s_h$ is the current observed state, $r_{h-1}$ is the reward of last step and $I$ is the index of the underlying MDP. On the other hand, the observation $o_h$ is $(s_h, r_{h-1})$. Then for any $h \in [H - l + 1]$, any latent state $\overline{s}_h = (s_h, r_{h-1}, I)$ and $(l+1)$-step test $\overline{t} = (o_h^t, a_h^t, \cdots, o_{h+l}^t)$, we have

$$\mathbb{P}(\overline{t}|\overline{s}_h) = \mathbf{1}(o_h^t = (s_h, r_{h-1})) \cdot \mathbb{P}_I(t|s_h),$$

where $t = (a_h^t, o_{h+1}^t, \cdots, o_{h+l}^t)$. Therefore, the $(l + 1)$-step emission matrix can be written as follows:

$$\mathbb{O}_{h,l+1} = \begin{bmatrix} L_{s_h^1} & 0 & 0 & 0 & \cdots & 0 \\ 0 & L_{s_h^1} & 0 & 0 & \cdots & 0 \\ 0 & 0 & L_{s_h^2} & 0 & \cdots & 0 \\ 0 & 0 & 0 & L_{s_h^2} & \cdots & 0 \\ \vdots & \vdots & \vdots & \vdots & \ddots & \vdots \\ 0 & 0 & 0 & 0 & \cdots & L_{s_h^{|\mathcal{S}|}} \end{bmatrix}.$$

Since the rank of $L_{s_h}$ is $M$ for any $s_h \in \mathcal{S}$, we know the rank of $\mathbb{O}_{h,l+1}$ is $2M|\mathcal{S}|$ for all $h \in [H - l + 1]$. This implies that the POMDP satisfies the $(l + 1)$-step weakly-revealing condition. $\square$

### E.4 $m$-STEP DECODABLE POMDPS

Recall the definition of $m$-step decodable POMDPs [13] is that there exist unknown decoders $\{\phi_{\mathrm{dec},h}\}_{m\leq h\leq H}$ such that for every reachable trajctory $\tau_H$, we have $s_h = \phi_{\mathrm{dec},h}(z_h)$ for all $m \leq h \leq H$ where $z_h = ((o, a)_{h-m+1:h-1}, o_h)$. From Lemma 2, we know that $d_{\mathrm{PSR}} \leq |\mathcal{S}|$. Further, the following lemma suggests that a general core test set for $m$-step decodable POMDPs is the set of all $m$-step futures:

**Lemma 7.** *A $m$-step decodable POMDP is a PSR with the core test set $\mathcal{U}_h = \mathcal{O} \times (\mathcal{A} \times \mathcal{O})^{m-1}$ for all $h \in [H - m + 1]$.*

*Proof.* Consider any $h \in [H - m + 1]$, let $q_{\tau_{h-1}} = [\mathbb{P}(u|\tau_{h-1})]_{u\in\mathcal{O}\times(\mathcal{A}\times\mathcal{O})^{m-1}}$. Then for any test $t = (o_{h:h+W}, a_{h:h+W-1})$, when $W \leq m - 1$, we have for any length-$(m - 1 - W)$ action sequence $\boldsymbol{a}$,

$$\mathbb{P}(t|\tau_{h-1}) = \sum_{u\in\mathcal{U}_{t,\boldsymbol{a}}} \mathbb{P}(u|\tau_{h-1}),$$

where $\mathcal{U}_{t,\boldsymbol{a}}$ denotes the set of all length-$m$ tests whose action sequence is $(a_{h:h+W-1}, \boldsymbol{a})$ and the first $W+1$ observations are $o_{h:h+W}$. This implies $\mathbb{P}(t|\tau_{h-1}) = m_{t,h}^\top q_{\tau_{h-1}}$ where $m_{t,h}$ sets the entries corresponding to the tests in $\mathcal{U}_{t,\boldsymbol{a}}$ as 1 and the others as 0.

When $W > m - 1$, we denote $t_{h:h+m-1}$ to be $(o_{h:h+m-1}, a_{h:h+m-1})$ and $t_{h+m}$ to be $(o_{h+m:h+W}, a_{h+m:h+W-1})$. Then we have

$$\mathbb{P}(t|\tau_{h-1}) = \mathbb{P}(t_{h:h+m-1}|\tau_{h-1})\mathbb{P}(o_{h+m:h+W}|(\tau_{h-1}, t_{h:h+m-1}); \mathrm{do}(a_{h+m:h+W-1}))$$
$$= \mathbb{P}(t_{h:h+m-1}|\tau_{h-1})\mathbb{P}(o_{h+m:h+W}|\phi_{\mathrm{dec},h+m-1}(z_{h+m-1}), a_{h+m-1}; \mathrm{do}(a_{h+m:h+W-1})).$$

Notice that $\mathbb{P}(o_{h+m:h+W}|\phi_{\mathrm{dec},h+m-1}(z_{h+m-1}), a_{h+m-1}; \mathrm{do}(a_{h+m:h+W-1}))$ only depends on $t$, therefore we have $\mathbb{P}(t|\tau_{h-1}) = m_{t,h}^\top q_{\tau_{h-1}}$ where $m_{t,h}$ sets the entry corresponding to $(o_{h:h+m-1}, a_{h:h+m-2})$ as $\mathbb{P}(o_{h+m:h+W}|\phi_{\mathrm{dec},h+m-1}(z_{h+m-1}), a_{h+m-1}; \mathrm{do}(a_{h+m:h+W-1}))$ and the others as 0. This concludes our proof. □

Similar to the discussion for $m$-step weakly-revealing POMDPs, it is suffcient to show the core test set of $m$-step decodable POMDPs up to step $H - m + 1$.

### E.5   LOW-RANK POMDPS

Next we consider low-rank POMDPs. Recall that for low-rank POMDPs, the transition kernel $\mathbb{T}_h$ has the following low-rank form for all $h \in [H]$:

$$\mathbb{T}_h(s'|s, a) = (\psi_h(s'))^\top \phi_h(s, a),$$

where $\psi_h : \mathcal{S} \mapsto \mathbb{R}^{d_{\mathrm{trans}}}$ and $\phi_h : \mathcal{S} \times \mathcal{A} \mapsto \mathbb{R}^{d_{\mathrm{trans}}}$. The next lemma indicates that for low-rank POMDPs, the minimum core test set size will be not larger than $d_{\mathrm{trans}}$, which can be potentially much smaller than $|\mathcal{S}|$:

**Lemma 8.** *For any low-rank POMDP, its minimum core test set size will be not larger than $d_{\mathrm{trans}}$.*

*Proof.* First notice that we have for any test $t$ and history $\tau_h$:

$$\mathbb{P}(t|\tau_h) = \langle [\mathbb{P}(t|s_{h+1})]_{s_{h+1}\in\mathcal{S}}, \boldsymbol{s}_{\tau_h} \rangle.$$

Besides, notice that from the low rank structure (4), we have for any $s_{h+1} \in \mathcal{S}$,

$$\mathbb{P}(s_{h+1}|\tau_h) = \psi_h(s_{h+1})^\top \sum_{s_h\in\mathcal{S}} \phi_h(s_h, a_h)\mathbb{P}(s_h|\tau_h).$$

This implies that

$$\mathbb{P}(t|\tau_h) = \left( \sum_{s_{h+1}\in\mathcal{S}} \mathbb{P}(t|s_{h+1})\psi_h(s_{h+1}) \right)^\top \cdot \left( \sum_{s_h\in\mathcal{S}} \phi_h(s_h, a_h)\mathbb{P}(s_h|\tau_h) \right).$$

This implies that the rank of the one-step system dynamics $D_h$ is not larger than $d_{\mathrm{trans}}$ for all $h \in [H]$. Therefore we have $d_{\mathrm{PSR}} \leq d_{\mathrm{trans}}$. □

## F   PROOF OF LEMMA 4

We first prove (4). Notice that we have

$$\mathbb{P}^\pi(\tau_h)q_{\tau_h} = [\mathbb{P}(u|\tau_h)\mathbb{P}^\pi(\tau_h)]_{u\in\mathcal{U}_{h+1}} = (\mathbb{P}(u|\tau_h))_{u\in\mathcal{U}_{h+1}}\mathbb{P}(o_h|\tau_{h-1})\pi(a_h|\tau_{h-1}, o_h)\mathbb{P}^\pi(\tau_{h-1})$$
$$= (\mathbb{P}(o_h, o(u)|\tau_{h-1}; \mathrm{do}(a_h, a(u))))_{u\in\mathcal{U}_{h+1}} \cdot \pi(a_h|\tau_{h-1}, o_h)\mathbb{P}^\pi(\tau_{h-1})$$
$$= M_{o_h,a_h,h}(q_{\tau_{h-1}}\mathbb{P}^\pi(\tau_{h-1}))\pi(a_h|\tau_{h-1}, o_h)$$
$$= \cdots = \prod_{l=1}^h M_{o_l,a_l,l}q_0 \times \pi(\tau_h) = b_{\tau_h} \times \pi(\tau_h),$$

where the third step comes from the definition (1). In particular, for any trajectory $\tau_H$, we have

$$\mathbb{P}^\pi(\tau_H) = \pi(a_H|\tau_{H-1}, o_H)(m_{o_H,H}^\top q_{\tau_{H-1}})\mathbb{P}^\pi(\tau_{H-1}) = m_{o_H,H}^\top \cdot \prod_{h=1}^{H-1} M_{o_h,a_h,h} \cdot q_0 \cdot \pi(\tau_H).$$

# G  $\epsilon$-BRACKET NUMBER OF $\mathcal{F}$

In this section we introduce some basic properties of the $\epsilon$-bracket number $\mathcal{N}_{\mathcal{F}}(\epsilon)$. We first consider PSRs and then take POMDPs as special examples.

## G.1  PSRs

For PSRs, let us define the covering number for the parameters $\{M_{o,a,h;f}, q_{0;f}\}$ as follows:

**Definition 7** ($\epsilon$-covering number). *The $\epsilon$-covering number of $\mathcal{F}$, denoted by $\mathcal{Z}_{\mathcal{F}}(\epsilon)$, is the minimum integer $n$ such that there exists a function class $\mathcal{F}'$ with $|\mathcal{F}'| = n$ and for any $f \in \mathcal{F}$ there exists $f' \in \mathcal{F}'$ such that $\max_{o\in\mathcal{O}, a\in\mathcal{A}, h\in[H-1], u\in\mathcal{U}_{h+1}} \|m_{(o,a,u),h;f} - m_{(o,a,u),h;f'}\|_\infty \leq \epsilon$ and $\|q_{0;f} - q_{0;f'}\|_\infty \leq \epsilon$.*

Here $\mathcal{F}'$ does not need to be valid PSR model classes and $m_{(o,a,u),h;f'}$ does not need to belong to the column space of $K_{h;f'}$. That said, we still use $\mathbb{P}^\pi_{f'}(\tau_H)$ to denote the product $m^\top_{o_H,H} \cdot \prod_{h=1}^{H-1} M_{o_h,a_h,h;f'} \cdot q_{0;f'} \cdot \pi(\tau_H)$ where $m_{o_H,H} = e_{o_H,H}$, although this might no longer be a valid distribution. Then the following lemma shows that the bracket number can be upper bounded by the covering number, whose proof is deferred to Appendix G.3.

**Lemma 9.** *Given $\mathcal{F}$ and any $\epsilon > 0$, suppose Assumption 1,2 and 3 hold, then we have*
$$\mathcal{N}_{\mathcal{F}}(\epsilon) \leq \mathcal{Z}_{\mathcal{F}}(\alpha\epsilon/(8|\mathcal{O}|^{H+1}|\mathcal{A}|^H H|\mathcal{U}_A|^2|\mathcal{U}|)).$$

Since the log covering number $\log \mathcal{Z}_{\mathcal{F}}(\epsilon)$ typically scales with $\log(1/\epsilon)$, Lemma 9 shows that $\log \mathcal{N}_{\mathcal{F}}(\epsilon)$ also scales with polynomial $H$ in general.

**Tabular PSRs.** Let us consider the tabular cases for example where we directly use $\{M_{o,a,h}, q_0\}_{o\in\mathcal{O}, a\in\mathcal{A}, h\in[H-1]}$ as the parameters of $\mathcal{F}$ and assume for all $f \in \mathcal{F}$ we have
$$\max_{o\in\mathcal{O}, a\in\mathcal{A}, h\in[H-1], u\in\mathcal{U}_{h+1}} \|m_{(o,a,u),h;f}\|_\infty \leq 1, \|q_{0;f}\|_\infty \leq 1$$
without loss of generality. Then we know
$$\log \mathcal{Z}_{\mathcal{F}}(\alpha\epsilon/(8|\mathcal{O}|^{H+1}|\mathcal{A}|^H H|\mathcal{U}_A^2||\mathcal{U}|)) \leq \mathcal{O}(|\mathcal{U}|^2|\mathcal{O}||\mathcal{A}|H^2 \log(H|\mathcal{O}||\mathcal{A}||\mathcal{U}_A||\mathcal{U}|/(\alpha\epsilon))),$$
which implies that
$$\log \mathcal{N}_{\mathcal{F}}(\epsilon) \leq \mathcal{O}(|\mathcal{U}|^2|\mathcal{O}||\mathcal{A}|H^2 \log(H|\mathcal{O}||\mathcal{A}||\mathcal{U}_A||\mathcal{U}|/(\alpha\epsilon))).$$

## G.2  POMDPs

For POMDPs, we can obtain a more efficient function class by modeling the emission matrix $\mathbb{O}_h$, transition kernel $\mathbb{T}_h$ and initial state distribution $\mu_1$ instead of $\{M_{o,a,h;f}, q_{0;f}\}$. Let us define the covering number for the parameters $\{\mathbb{T}_{h;f}, \mathbb{O}_{h;f}, \mu_{1;f}\}_{h\in[H]}$ as follows:

**Definition 8.** *The $\epsilon$-covering number of $\{\mathbb{T}_{h;f}, \mathbb{O}_{h;f}, \mu_{1;f}\}_{h\in[H], f\in\mathcal{F}}$, denoted by $\mathcal{V}_{\mathcal{F}}(\epsilon)$, is the minimum integer $n$ such that there exists a function class $\mathcal{F}'$ with $|\mathcal{F}'| = n$ and for any $f \in \mathcal{F}$ there exists $f' \in \mathcal{F}'$ such that $\max_{h\in[H-1], a\in\mathcal{A}} \|\mathbb{T}_{h,a;f} - \mathbb{T}_{h,a;f'}\|_{\infty,\infty} \leq \epsilon, \max_{h\in[H]} \|\mathbb{O}_{h;f} - \mathbb{O}_{h;f'}\|_{\infty,\infty} \leq \epsilon$ and $\|\mu_{1;f} - \mu_{1;f'}\|_\infty \leq \epsilon$.*

Then we have the following lemma:

**Lemma 10.** *For any $f \in \mathcal{F}$ and $0 < \epsilon_1 \leq 1$, suppose $f'$ satisfies $\max_{h\in[H-1], a\in\mathcal{A}} \|\mathbb{T}_{h,a;f} - \mathbb{T}_{h,a;f'}\|_{\infty,\infty} \leq \epsilon_{\mathrm{op}}, \max_{h\in[H]} \|\mathbb{O}_{h;f} - \mathbb{O}_{h;f'}\|_{\infty,\infty} \leq \epsilon_{\mathrm{op}}$ and $\|\mu_{1;f} - \mu_{1;f'}\|_\infty \leq \epsilon_{\mathrm{op}}$, where*
$$\epsilon_{\mathrm{op}} = \epsilon_1/(14|\mathcal{O}|^2).$$
*Then for any policy $\pi$, we have*
$$\sum_{\tau_H} |\mathbb{P}^\pi_{f'}(\tau_H) - \mathbb{P}^\pi_f(\tau_H)| \leq \epsilon_1.$$

The proof is omitted here since it follows similar arguments in the proof of Lemma 11. Therefore, following the arguments in the proof of Lemma 9, we know
$$\mathcal{N}_{\mathcal{F}}(\epsilon) \leq \mathcal{V}_{\mathcal{F}}(\epsilon/(28|\mathcal{O}|^{H+2}|\mathcal{A}|^H)).$$

**Tabular POMDPs.** For tabular POMDPs where $\{\mathbb{T}_{h;f}, \mathbb{O}_{h;f}, \mu_{1;f}\}_{h \in [H]}$ is modeled directly, it can be observed that $\log \mathcal{V}_{\mathcal{F}}(\epsilon) = \mathcal{O}(H|\mathcal{O}||\mathcal{S}|^2|\mathcal{A}| \log(1/\epsilon))$. Therefore we have

$$\log \mathcal{N}_{\mathcal{F}}(\epsilon) \leq \text{poly}(|\mathcal{O}|, |\mathcal{A}|, |\mathcal{S}|, H, \log(1/\epsilon)).$$

**Low-rank POMDPs.** For low-rank POMDPs, when utilizing the function class introduced in Section 5, we can obtain that $\log \mathcal{V}_{\mathcal{F}}(\epsilon) \leq \log \mathcal{Y}_{\Phi}(\epsilon/(3d_{\text{trans}})) + \log \mathcal{Y}_{\Psi}(\epsilon/(3d_{\text{trans}})) + \log \mathcal{Y}_{\mathbb{O}}(\epsilon/3) + \log \mathcal{Y}_{\mu}(\epsilon)$, which implies that

$$\log \mathcal{N}_{\mathcal{F}}(\epsilon) \leq \log \mathcal{Y}_{\Phi}(\epsilon_{\text{LR}}/d_{\text{trans}}) + \log \mathcal{Y}_{\Psi}(\epsilon_{\text{LR}}/d_{\text{trans}}) + \log \mathcal{Y}_{\mathbb{O}}(\epsilon_{\text{LR}}) + \log \mathcal{Y}_{\mu}(\epsilon_{\text{LR}}),$$

where $\epsilon_{\text{LR}} := \mathcal{O}\big(\epsilon/(|\mathcal{O}|^{H+2}|\mathcal{A}|^H)\big)$.

**Linear POMDPs.** For linear POMDPs, when utilizing the function class introduced in Section 5, it can be calculated that $\log \mathcal{V}_{\mathcal{F}}(\epsilon) = \mathcal{O}(H \prod_{i=1}^{5} d_i \log(\prod_{i=1}^{5} d_i/\epsilon))$, which implies that

$$\log \mathcal{N}_{\mathcal{F}}(\epsilon) \leq \mathcal{O}\big(H^2 \prod_{i=1}^{5} d_i \log(|\mathcal{O}||\mathcal{A}|/\epsilon)\big).$$

### G.3 PROOF OF LEMMA 9

First let us prove that $\mathbb{P}_f^{\pi}(\cdot)$ is Lipschitz continuous with respect to $\{M_{o,a,h;f}, q_{0;f}\}$ for any policy $\pi$, as shown in the following lemma:

**Lemma 11.** *For any $f \in \mathcal{F}$ and $0 < \epsilon_1 \leq |\mathcal{U}_A|$, suppose $f'$ satisfies*

$$\max_{o \in \mathcal{O}, a \in \mathcal{A}, h \in [H-1], u \in \mathcal{U}_{h+1}} \|m_{(o,a,u),h;f} - m_{(o,a,u),h;f'}\|_{\infty} \leq \epsilon_{\text{op}}, \|q_{0;f} - q_{0;f'}\|_{\infty} \leq \epsilon_{\text{op}},$$

*where*

$$\epsilon_{\text{op}} = \alpha \epsilon_1/(4H|\mathcal{U}_A|^2|\mathcal{U}||\mathcal{O}|).$$

*Then for any policy $\pi$, we have*

$$\sum_{\tau_H} |\mathbb{P}_{f'}^{\pi}(\tau_H) - \mathbb{P}_f^{\pi}(\tau_H)| \leq \epsilon_1.$$

Now consider the minimum $\epsilon_{\text{op}}$-covering net of $\mathcal{F}$, denoted by $\mathcal{F}'$. Then by the definition of minimum covering net, we know for any $f \in \mathcal{F}'$, there exists $f' \in \mathcal{F}$ such that

$$\max_{o \in \mathcal{O}, a \in \mathcal{A}, h \in [H-1], u \in \mathcal{U}_{h+1}} \|m_{(o,a,u),h;f} - m_{(o,a,u),h;f'}\|_{\infty} \leq \epsilon_{\text{op}}, \|q_{0;f} - q_{0;f'}\|_{\infty} \leq \epsilon_{\text{op}}.$$

Using Lemma 11, we know for any policy $\pi$ and trajectory $\tau_H$,

$$\mathbb{P}_{f'}^{\pi}(\tau_H) - \epsilon_1 \leq \mathbb{P}_f^{\pi}(\tau_H) \leq \mathbb{P}_{f'}^{\pi}(\tau_H) + \epsilon_1.$$

Therefore, let us define $g_1^{f'}(\pi, \cdot) = \mathbb{P}_{f'}^{\pi}(\cdot) - \epsilon_1$ and $g_2^{f'}(\pi, \cdot) = \mathbb{P}_{f'}^{\pi}(\cdot) + \epsilon_1$, then the set $\{[g_1^{f'}, g_2^{f'}] : f' \in \mathcal{F}'\}$ is a $2\epsilon_1(|\mathcal{O}||\mathcal{A}|)^H$-bracket of $\mathcal{F}$ where we use the fact that there are at most $(|\mathcal{O}||\mathcal{A}|)^H$ many trajectories. Let $2\epsilon_1(|\mathcal{O}||\mathcal{A}|)^H = \epsilon$ and then we have

$$\mathcal{N}_{\mathcal{F}}(\epsilon) \leq \mathcal{Z}_{\mathcal{F}}(\alpha \epsilon/(8|\mathcal{O}|^{H+1}|\mathcal{A}|^H H|\mathcal{U}_A|^2|\mathcal{U}|)).$$

### G.4 PROOF OF LEMMA 11

We use Lemma 16 to prove this lemma via induction. First notice that we have for any $o \in \mathcal{O}, a \in \mathcal{A}, h \in [H-1], u \in \mathcal{U}_{h+1}$,

$$\|m_{(o,a,u),h;f'} - m_{(o,a,u),h;f}\|_{\infty} \leq \epsilon_{\text{op}}, \tag{7}$$

$$\|q_{0;f'} - q_{0;f}\|_{\infty} \leq \epsilon_{\text{op}}. \tag{8}$$

In the following discussion, we use $q_0, m_{(o,a,u),h}, M_{o,a,h}, b_{\tau_h}$ to denote $q_{0;f}, m_{(o,a,u),h;f},$ $M_{o,a,h;f}, b_{\tau_h;f}$ and $q_0', m_{(o,a,u),h}', M_{o,a,h}', b_{\tau_h}'$ to denote $q_{0;f'}, m_{(o,a,u),h;f'}, M_{o,a,h;f'}, b_{\tau_h;f'}$ to simplify writing. Next we use induction to prove the lemma.

For the base case, we have $b_{\tau_0} = q_0, b_{\tau_0}' = q_0'$. Therefore from (8) we have

$$\|b_{\tau_0} - b_{\tau_0}'\|_1 \le |\mathcal{U}|\epsilon_{\mathrm{op}} \le \epsilon_1.$$

Now suppose for any $h' \le h$ where $h \in [H-2]^+$ and policy $\pi$, we have $\sum_{\tau_{h'}} \|b_{\tau_{h'}} - b_{\tau_{h'}}'\|_1 \times \pi(\tau_{h'}) \le \epsilon_1$. Notice that here $f'$ might not satisfy Assumption 3, but from the proof of Lemma 14 we can see that Lemma 14 still holds since $f \in \mathcal{F}$. Therefore we have for any policy $\pi$,

$$\sum_{\tau_{h+1}} \|b_{\tau_{h+1}} - b_{\tau_{h+1}}'\|_1 \times \pi(\tau_{h+1})$$

$$\le \frac{|\mathcal{U}_A|}{\alpha}\left(\sum_{l=1}^{h+1}\sum_{\tau_l}\|[M_{o_l,a_l,l} - M_{o_l,a_l,l}']b_{\tau_{l-1}}'\|_1 \times \pi(\tau_l) + \|q_0 - q_0'\|_1\right). \tag{9}$$

From (7), we know for any $l \in [h+1]$,

$$\sum_{\tau_l} \|[M_{o_l,a_l,l} - M_{o_l,a_l,l}']b_{\tau_{l-1}}'\|_1 \times \pi(\tau_l)$$

$$\le \epsilon_{\mathrm{op}}|\mathcal{U}|\sum_{\tau_l}\|b_{\tau_{l-1}}'\|_1 \times \pi(\tau_l)$$

$$= \epsilon_{\mathrm{op}}|\mathcal{U}||\mathcal{O}|\sum_{\tau_{l-1}}\|b_{\tau_{l-1}}'\|_1 \times \pi(\tau_{l-1})$$

$$\le \epsilon_{\mathrm{op}}|\mathcal{U}||\mathcal{O}|\sum_{\tau_{l-1}}(\|b_{\tau_{l-1}}\|_1 \times \pi(\tau_{l-1}) + \|b_{\tau_{l-1}} - b_{\tau_{l-1}}'\|_1 \times \pi(\tau_{l-1}))$$

$$\le \epsilon_{\mathrm{op}}|\mathcal{U}||\mathcal{O}|\sum_{\tau_{l-1}}(|\mathcal{U}_A| + \|b_{\tau_{l-1}} - b_{\tau_{l-1}}'\|_1 \times \pi(\tau_{l-1}))$$

$$\le \epsilon_{\mathrm{op}}|\mathcal{U}||\mathcal{O}|(\epsilon_1 + |\mathcal{U}_A|). \tag{10}$$

Here the first step comes from Cauchy-Schwartz inequality and (7). The fourth step comes from the fact that $(b_{\tau_{l-1}}\pi(\tau_{l-1}))_u = \mathbb{P}_f(u|\tau_{l-1})\mathbb{P}_f^\pi(\tau_{l-1})$ and thus $\sum_{\tau_{l-1}}\|b_{\tau_{l-1}}\|_1 \times \pi(\tau_{l-1}) = \sum_{\tau_{l-1}}\|q_{\tau_{l-1};f}\|_1 \cdot \mathbb{P}_f^\pi(\tau_{l-1}) \le |\mathcal{U}_A|\sum_{\tau_{l-1}}\mathbb{P}_f^\pi(\tau_{l-1}) = |\mathcal{U}_A|$. The last step comes from the induction hypothesis.

Substituting (8) and (10) into (9), we have

$$\sum_{\tau_{h+1}} \|b_{\tau_{h+1}} - b_{\tau_{h+1}}'\|_1 \times \pi(\tau_{h+1}) \le \epsilon_1.$$

Therefore, we have for all $h \in [H-1]$ and policy $\pi$,

$$\sum_{\tau_h} \|b_{\tau_h} - b_{\tau_h}'\|_1 \times \pi(\tau_h) \le \epsilon_1. \tag{11}$$

Notice that from Lemma 4 and Assumption 2 (where we let $m_{o_H,H}^k = m_{o_H,H} = e_{o_H,H}$), we have for any policy $\pi$,

$$\sum_{\tau_H} |P_{f'}^\pi(\tau_H) - P_f^\pi(\tau_H)| \le \sum_{\tau_{H-1}} \|b_{\tau_{H-1}} - b_{\tau_{H-1}}'\|_1 \times \pi(\tau_{H-1}). \tag{12}$$

Combining (11) and (12), we have for all policy $\pi$

$$\sum_{\tau_H} |P_{f'}^\pi(\tau_H) - P_f^\pi(\tau_H)| \le \epsilon_1.$$

This concludes our proof.

## H  REDUNDANCY OF $m_{(o,a,u),h;f}$

In Section 4 we mention that there is redundancy in the choice of $m_{(o,a,u),h;f}$ given the model $\mathbb{P}^\pi_f$ and we can replace any $m_{(o,a,u),h;f}$ with its projection on the space spanned by $\{q_{\tau_{h-1};f}\}_{\tau_{h-1}}$. The following lemma characterizes this formally:

**Lemma 12.** *Suppose Assumption 2 holds. Given any parameter $\{M_{o,a,h;f}, q_{0;f}\}$, suppose for another set of parameters $\{M_{o,a,h;f'}, q_{0;f}\}$ we have for all $o \in \mathcal{O}, a \in \mathcal{A}, h \in [H-1], u \in \mathcal{U}_{h+1}$*

$$m_{(o,a,u),h;f'} = \text{Proj}_{\text{Col}(K_{h-1;f})}(m_{(o,a,u),h;f}).$$

*Then for any trajectory $\tau_H$ and policy $\pi$, we have $\mathbb{P}^\pi_f(\tau_H) = \mathbb{P}^\pi_{f'}(\tau_H)$. This means that $\{m_{(o,a,u),h;f'}\}$ is also a valid set of predictive parameters for the model $\mathbb{P}_f$.*

*Proof.* We first show that $b_{\tau_h;f} = b_{\tau_h;f'}$ for any $h \in [H-1]^+$ and trajectory $\tau_h$. We prove this via induction. For the base case where $h = 0$, $b_{\tau_h;f} = b_{\tau_h;f'} = q_{0;f}$. Next for any $h \in [H-2]^+$, we suppose $b_{\tau_{h'};f} = b_{\tau_{h'};f'}$ for any $h' \in [h]^+$ and trajectory $\tau_{h'}$. Then for any trajectory $\tau_{h+1}$, let $\pi_{\tau_h}$ denote the policy that always takes the action sequence in $\tau_h$. From (4) in Lemma 4, we have

$$\begin{aligned}
b_{\tau_{h+1};f} &= M_{o_{h+1},a_{h+1},h+1;f} b_{\tau_h;f} = M_{o_{h+1},a_{h+1},h+1;f} b_{\tau_h;f} \pi_{\tau_h}(\tau_h) \\
&= M_{o_{h+1},a_{h+1},h+1;f} q_{\tau_h;f} \mathbb{P}^{\pi_{\tau_h}}(\tau_h) \\
&= (m^\top_{(o_{h+1},a_{h+1},u),h+1;f} q_{\tau_h;f})_{u\in\mathcal{U}_{h+2}} \mathbb{P}^{\pi_{\tau_h}}(\tau_h).
\end{aligned} \tag{13}$$

Similarly, since $b_{\tau_h;f'} = b_{\tau_h,f}$, we have

$$b_{\tau_{h+1};f'} = (m^\top_{(o_{h+1},a_{h+1},u),h+1;f'} q_{\tau_h;f})_{u\in\mathcal{U}_{h+2}} \mathbb{P}^{\pi_{\tau_h}}(\tau_h). \tag{14}$$

From (2), we know $q_{\tau_h;f}$ belongs to the column space of $K_{h;f}$. This implies that for any $u \in \mathcal{U}_{h+2}$

$$m^\top_{(o_{h+1},a_{h+1},u),h+1;f} q_{\tau_h;f} = m^\top_{(o_{h+1},a_{h+1},u),h+1;f'} q_{\tau_h;f}. \tag{15}$$

Combining (13),(14) and (15), we have

$$b_{\tau_{h+1};f} = b_{\tau_{h+1};f'}.$$

Therefore, for any $h \in [H-1]^+$ and trajectory $\tau_h$, we have

$$b_{\tau_h;f} = b_{\tau_h;f'}.$$

This suggests that for any policy $\pi$ and trajctory $\tau_{H-1}$, we have

$$(\mathbb{P}_f(u|\tau_{H-1})\mathbb{P}^\pi_f(\tau_{H-1}))_{u\in\mathcal{U}_H} = (\mathbb{P}_{f'}(u|\tau_{H-1})\mathbb{P}^\pi_{f'}(\tau_{H-1}))_{u\in\mathcal{U}_H}.$$

Therefore with Assumption 2 we have for any policy $\pi$ and trajectory $\tau_H$, we have

$$\mathbb{P}^\pi_f(\tau_H) = \mathbb{P}^\pi_{f'}(\tau_H).$$

$\square$

## I  PROOF OF THEOREM 1

In this section we present a proof sketch for Theorem 1. Note that to prove Theorem 1, we only need to show that CRANE can achieve sublinear total suboptimality, which is stated in the following theorem:

**Theorem 2.** *Under Assumption 1,2,3, there exists an absolute constant $c$ such that for any $\delta \in (0,1]$, $T \in \mathbb{N}$, if we choose $\beta = c\log(\mathcal{N}_\mathcal{F}(\epsilon_b)TH|\mathcal{U}_A|/\delta)$ in CRANE where $\epsilon_b = 1/(TH|\mathcal{U}_A|)$, then with probability at least $1 - \delta$, we have:*

$$\sum_{k=1}^T (V^* - V^{\pi^k}) \leq \mathcal{O}(d^2_{\text{PSR}} H^{\frac{7}{2}} |\mathcal{U}_A|^4 |\mathcal{A}|^2 T^{\frac{1}{2}} \alpha^{-3} \cdot \log(TH\mathcal{N}_\mathcal{F}(\epsilon_b)|\mathcal{O}||\mathcal{A}|/\delta)).$$

The $\sqrt{T}$ bound on the regret in Theorem 2 suggests that the uniform mixture of the output policies $\widehat{\pi} = \text{Unif}(\{\pi^k\}_{k=1}^T)$ is an $\epsilon$-optimal policy when $T = \widetilde{\mathcal{O}}(1/\epsilon^2)$, leading to Theorem 1 directly. Therefore we only need to prove Theorem 2 now.

Note that we can decompose the total suboptimality into the following terms:

$$\text{Regret}(T) = \sum_{k=1}^T \left(V^* - V^{\pi^k}\right) = \underbrace{\left(\sum_{t=1}^T \left(V^* - V_{f^k}^{\pi^k}\right)\right)}_{(1)} + \underbrace{\left(\sum_{t=1}^T \left(V_{f^k}^{\pi^k} - V^{\pi^k}\right)\right)}_{(2)}. \qquad (16)$$

Our proof bounds these two terms separately and mainly consists of four steps:

1. Prove $V_{f^k}^{\pi^k}$ is an optimistic estimation of $V^*$ for all $k \in [T]$, which implies that term $(1) \leq 0$.

2. Decompose term (2) into the estimation error of the parameter $M_{o,a,h}$ via the system dynamics (3).

3. Bound the cumulative estimation error using the property of MLE.

4. Bound term (2) by connecting the results in the second and third step.

### I.1 STEP 1: PROVE OPTIMISM

First we can show that the constructed set $\mathcal{B}^k$ contains the true model parameter $f^*$ with high probability:

**Lemma 13.** *With probability at least $1 - \delta/2$, we have for all $k \in [T]$, $f^* \in \mathcal{B}^k$.*

*Proof.* See Appendix J.1. $\qquad \square$

Then since $V_{f^k}^{\pi^k} = \max_{f \in \mathcal{B}^k, \pi} V_f^\pi$, we know for all $k \in [T]$,

$$V_{f^k}^{\pi^k} \geq V_{f^*}^{\pi^*} = V^*.$$

Thus, Lemma 13 implies that $V_{f^k}^{\pi^k}$ is an optimistic estimation of $V^*$ for all $k$, and therefore term (1) in (16) is non-positive.

### I.2 STEP 2: DECOMPOSE THE PERFORMANCE DIFFERENCE

Next we aim to handle term (2) in (16) and show the estimation error $\sum_{t=1}^T \left(V_{f^k}^{\pi^k} - V^{\pi^k}\right)$ is small. First we need to decompose the performance difference $V_{f^k}^{\pi^k} - V^{\pi^k}$ into the estimation error of the parameters $M_{o,a,h}$ in order to apply the property of MLE later. Notice that we have,

$$V_{f^k}^{\pi^k} - V^{\pi^k} \leq H \sum_{\tau_H} |P_{f^k}^{\pi^k}(\tau_H) - P_{f^*}^{\pi^k}(\tau_H)|$$

$$= H \sum_{\tau_H} \left|(m_{o_H,H}^k)^\top \cdot \prod_{h=1}^{H-1} M_{o_h,a_h,h}^k \cdot q_0^k - m_{o_H,H}^\top \cdot \prod_{h=1}^{H-1} M_{o_h,a_h,h} \cdot q_0\right| \times \pi^k(\tau_H)$$

$$= H \sum_{\tau_{H-1}} \left\|\prod_{h=1}^{H-1} M_{o_h,a_h,h}^k \cdot q_0^k - \prod_{h=1}^{H-1} M_{o_h,a_h,h} \cdot q_0\right\|_1 \times \pi^k(\tau_{H-1}), \qquad (17)$$

where we use $m_{o_H,H}^k, M_{o_h,a_h,h}^k, q_0^k$ to denote $m_{o_H,H;f^k}, M_{o_h,a_h,h;f^k}, q_{0;f^k}$ and $m_{o_H,H}, M_{o_h,a_h,h}, q_0$ to denote $m_{o_H,H;f^*}, M_{o_h,a_h,h;f^*}, q_{0;f^*}$. The second step is due to Lemma 4 and the last step is because based on Assumption 2 we have set $m_{o_H,H}^k = m_{o_H,H} = e_{o_H,H}$.

The following lemma bridges the term in (17) and the estimation error of $M_{o,a,h}$, whose proof is deferred to Appendix J.2:

**Lemma 14.** *For any $k \in [T]$, $h \in [H]$ and policy $\pi$, we have*

$$\sum_{\tau_h} \left\| \prod_{l=1}^{h} M^k_{o_l,a_l,l} \cdot q_0^k - \prod_{l=1}^{h} M_{o_l,a_l,l} \cdot q_0 \right\|_1 \times \pi(\tau_h)$$

$$\leq \frac{|\mathcal{U}_A|}{\alpha} \left( \sum_{l=1}^{h} \sum_{\tau_l} \|[M^k_{o_l,a_l,l} - M_{o_l,a_l,l}]b_{\tau_{l-1}}\|_1 \times \pi(\tau_l) + \|q_0^k - q_0\|_1 \right), \qquad (18)$$

*where $b_{\tau_l} = \prod_{j=1}^{l} M_{o_j,a_j,j} q_0$.*

**Remark 3.** *From the proof of Lemma 14, we can see that Lemma 14 only utilizes the properties of $\{M^k_{o_l,a_l,l}, q_0^k\}_{l\in[h]}$. Therefore Lemma 14 still holds even if the system dynamics induced by $\{M_{o_l,a_l,l}, q_0\}_{l\in[h]}$ is invalid. We will use this fact in the analysis about the $\epsilon$-bracket number.*

Therefore, substituting Lemma 14 into (17), we can bound the performance difference by the cumulative estimation error:

$$\sum_{k=1}^{T} \left( V_{f^k}^{\pi^k} - V^{\pi^k} \right)$$

$$\leq \frac{|\mathcal{U}_A|H}{\alpha} \sum_{k=1}^{T} \left( \sum_{h=1}^{H-1} \sum_{\tau_h} \|[M^k_{o_h,a_h,h} - M_{o_h,a_h,h}]b_{\tau_{h-1}}\|_1 \times \pi^k(\tau_h) + \|q_0^k - q_0\|_1 \right). \quad (19)$$

### I.3 STEP 3: BOUND THE ESTIMATION ERROR

Now we need to bound the estimation error in (19). First we introduce the following guarantee of MLE from the literature, which connects the log-likelihood ratio $\log(P_{f^*}^\pi(\tau_H)/P_f^\pi(\tau_H))$ and the total variation $\sum_{\tau_H} |P_f^\pi(\tau_H) - P_{f^*}^\pi(\tau_H)|$:

**Lemma 15** ([37, Proposition 14]). *There exists a universal constant $c_1$ such that for any $\delta \in (0,1]$, for all $k \in [T]$ and $f \in \mathcal{F}$, we have with probability at least $1 - \delta/2$ that*

$$\sum_{i=1}^{k} \sum_{h\in[H-1]^+, u_{a,h+1}\in\mathcal{U}_{A,h+1}} \left( \sum_{\tau_H} |\mathbb{P}_f^{\pi^{i,u_{a,h+1},h}}(\tau_H) - \mathbb{P}_{f^*}^{\pi^{i,u_{a,h+1},h}}(\tau_H)| \right)^2$$

$$\leq c_0 \left( \sum_{i=1}^{k} \sum_{h\in[H-1]^+, u_{a,h+1}\in\mathcal{U}_{A,h+1}} \log \left( \frac{\mathbb{P}_{f^*}^{\pi^{i,u_{a,h+1},h}}(\tau_H^{i,u_{a,h+1},h})}{\mathbb{P}_f^{\pi^{i,u_{a,h+1},h}}(\tau_H^{i,u_{a,h+1},h})} \right) + \log(\mathcal{N}_\mathcal{F}(\epsilon_b) T H |\mathcal{U}_A|/\delta) \right).$$

Combining Lemma 15 and the fact that both $f^*$ and $f^k$ belongs to $\mathcal{B}^k$, we have with probability at least $1 - \delta$ that for all $k \in [T]$,

$$\sum_{i=1}^{k-1} \sum_{h\in[H-1]^+, u_{a,h+1}\in\mathcal{U}_{A,h+1}} \left( \sum_{\tau_H} |\mathbb{P}_{f^k}^{\pi^{i,u_{a,h+1},h}}(\tau_H) - \mathbb{P}_{f^*}^{\pi^{i,u_{a,h+1},h}}(\tau_H)| \right)^2 \leq \mathcal{O}(\beta). \qquad (20)$$

The following discussion is conditioned on the event in (20) being true. Then by Cauchy-Schwarz inequality we have for all $k \in [T]$,

$$\sum_{i=1}^{k-1} \sum_{h\in[H-1]^+, u_{a,h+1}\in\mathcal{U}_{A,h+1}} \sum_{\tau_H} |\mathbb{P}_{f^k}^{\pi^{i,u_{a,h+1},h}}(\tau_H) - \mathbb{P}_{f^*}^{\pi^{i,u_{a,h+1},h}}(\tau_H)| \leq \mathcal{O}(\sqrt{kH|\mathcal{U}_A|\beta}). \quad (21)$$

Suppose the length of the longest action sequence in $\mathcal{U}_{A,h}$ is $l_a$. Then since the environment will only generate dummy observations $o_{\text{dummy}}$ after $a_H$, we have for any policy $\pi$ and $f \in \mathcal{F}$,

$$\sum_{\tau_{H+l_a+1}} |\mathbb{P}_f^\pi(\tau_{H+l_a+1}) - \mathbb{P}_{f^*}^\pi(\tau_{H+l_a+1})|$$

$$= \sum_{\tau_{H+l_a+1}} |\mathbb{P}_f^\pi(\tau_H)\mathbf{1}(o_{H+1:H+l_a+1} = o_{\text{dummy}})\pi(a_{H+1:H+l_a+1}|\tau_H)$$

$$- \mathbb{P}^{\pi}_{f*}(\tau_H)\mathbf{1}(o_{H+1:H+l_a+1} = o_{\text{dummy}})\pi(a_{H+1:H+l_a+1}|\tau_H)|$$
$$= \sum_{\tau_H} |\mathbb{P}^{\pi}_f(\tau_H) - \mathbb{P}^{\pi}_{f*}(\tau_H)|.$$

Therefore, we can marginalize the distribution $\mathbb{P}^{\pi}_f(\tau_H)$ and $\mathbb{P}^{\pi}_{f*}(\tau_H)$ in (21) and have for all $k \in [T], i \in [k-1], h \in [H-1]^+$,

$$\sum_{u_{a,h+1}\in\mathcal{U}_{A,h+1}} \sum_{\tau_H} |\mathbb{P}^{\pi^{i,u_{a,h+1},h}}_{f^k}(\tau_H) - \mathbb{P}^{\pi^{i,u_{a,h+1},h}}_{f*}(\tau_H)|$$

$$\geq \sum_{u_{a,h+1}\in\mathcal{U}_{A,h+1}} \sum_{\tau_h,\boldsymbol{o}\in\boldsymbol{O}(u_{a,h+1})} |\mathbb{P}^{\pi^{i,u_{a,h+1},h}}_{f^k}(\tau_h, u_{a,h+1}, \boldsymbol{o}) - \mathbb{P}^{\pi^{i,u_{a,h+1},h}}_{f*}(\tau_h, u_{a,h+1}, \boldsymbol{o})|$$

$$= \sum_{u_{a,h+1}\in\mathcal{U}_{A,h+1}} \sum_{\tau_h,\boldsymbol{o}\in\boldsymbol{O}(u_{a,h+1})} |\mathbb{P}^{\pi^{i,h}}_{f^k}(\tau_h)\mathbb{P}_{f^k}(\boldsymbol{o}|\tau_h; \text{do}(u_{a,h+1}))$$

$$- \mathbb{P}^{\pi^{i,h}}_{f*}(\tau_h)\mathbb{P}_{f*}(\boldsymbol{o}|\tau_h; \text{do}(u_{a,h+1}))| \times \pi^{i,u_{a,h+1},h}(u_{a,h+1}|\tau_h)$$

$$= \sum_{\tau_h} \sum_{u_{a,h+1}\in\mathcal{U}_{A,h+1},\boldsymbol{o}\in\boldsymbol{O}(u_{a,h+1})} |\mathbb{P}^{\pi^{i,h}}_{f^k}(\tau_h)\mathbb{P}_{f^k}(\boldsymbol{o}|\tau_h; \text{do}(u_{a,h+1})) - \mathbb{P}^{\pi^{i,h}}_{f*}(\tau_h)\mathbb{P}_{f*}(\boldsymbol{o}|\tau_h; \text{do}(u_{a,h+1}))|$$

$$= \sum_{\tau_h} \sum_{u\in\mathcal{U}_{h+1}} |\mathbb{P}^{\pi^{i,h}}_{f^k}(\tau_h)\mathbb{P}_{f^k}(u|\tau_h) - \mathbb{P}^{\pi^{i,h}}_{f*}(\tau_h)\mathbb{P}_{f*}(u|\tau_h)|.$$

Here in the first step $\boldsymbol{O}(u_{a,h+1})$ denote the set of observation sequences that occur together with $u_{a,h+1}$ in $\mathcal{U}_{h+1}$ and $\mathbb{P}^{\pi}_f(\tau_h, u_{a,h+1}, \boldsymbol{o})$ denotes the joint probability of observing the trajectory $(\tau_h, \boldsymbol{o}, u_{a,h+1})$. In the third and fourth step we utilize the fact that $\pi^{k,u_{a,h+1},h} = \pi^k_{1:h-1} \circ \text{Unif}(\mathcal{A}) \circ u_{a,h+1}$ and we define $\pi^{i,h} := \pi^i_{1:h-1} \circ \text{Unif}(\mathcal{A})$. Then based on Eq. (4) and Eq. (21), we have for all $k \in [T], h \in [H-1]^+$,

$$\sum_{i=1}^{k-1}\sum_{\tau_h} \pi^{i,h}(\tau_h) \cdot \|b^k_{\tau_h} - b_{\tau_h}\|_1 \leq \mathcal{O}(\sqrt{kH|\mathcal{U}_A|\beta}).$$

Thus via importance weighting, we have for all $k \in [T], h \in [H-1]^+$,

$$\sum_{i=1}^{k-1}\sum_{\tau_h} \pi^i(\tau_h) \cdot \|b^k_{\tau_h} - b_{\tau_h}\|_1 \leq \mathcal{O}(|\mathcal{A}|\sqrt{kH|\mathcal{U}_A|\beta}), \tag{22}$$

$$\sum_{i=1}^{k-1}\sum_{\tau_h} \pi^i(\tau_{h-1}) \cdot \|b^k_{\tau_h} - b_{\tau_h}\|_1 \leq \mathcal{O}(|\mathcal{A}|\sqrt{kH|\mathcal{U}_A|\beta}). \tag{23}$$

In particular, when $h = 0$ we have

$$\|q^k_0 - q_0\|_1 \leq \mathcal{O}(\sqrt{H|\mathcal{U}_A|\beta/k}) \tag{24}$$

Now for all $k \in [T], h \in [H-1]$, we can bound the estimation error as follows:

$$\sum_{i=1}^{k-1}\sum_{\tau_h} \|[M^k_{o_h,a_h,h} - M_{o_h,a_h,h}]b_{\tau_{h-1}}\|_1 \times \pi^i(\tau_{h-1})$$

$$\leq \sum_{i=1}^{k-1}\sum_{\tau_h} \|[M^k_{o_h,a_h,h}b^k_{\tau_{h-1}} - M_{o_h,a_h,h}b_{\tau_{h-1}}]\|_1 \times \pi^i(\tau_{h-1})$$

$$+ \sum_{i=1}^{k-1}\sum_{\tau_h} \|M^k_{o_h,a_h,h}[b^k_{\tau_{h-1}} - b_{\tau_{h-1}}]\|_1 \times \pi^i(\tau_{h-1}). \tag{25}$$

For the first term in (25), we have

$$
\sum_{i=1}^{k-1} \sum_{\tau_h} \| [M^k_{o_h,a_h,h} b^k_{\tau_{h-1}} - M_{o_h,a_h,h} b_{\tau_{h-1}}] \|_1 \times \pi^i(\tau_{h-1})
$$

$$
= \sum_{i=1}^{k-1} \sum_{\tau_h} \pi^i(\tau_{h-1}) \cdot \| b^k_{\tau_h} - b_{\tau_h} \|_1 \le \mathcal{O}(|\mathcal{A}| \sqrt{kH|\mathcal{U}_A|\beta}), \tag{26}
$$

where the second step is due to (23).

To bound the second term, we need to bound $\| M^k_{o,a,h} \|_{1,1}$ first, which is given in the following lemma:

**Lemma 16.** *For any $1 \le j_1 \le j_2 \le H - 1$, trajectory $\tau_{j_1-1}$, policy $\pi$, $f \in \mathcal{F}$ and $x \in \mathbb{R}^{|\mathcal{U}_{j_1}|}$, we have*

$$
\sum_{\tau_{j_1:j_2}} \left\| \prod_{j=j_1}^{j_2} M_{o_j,a_j,j;f} x \right\|_1 \times \pi(\tau_{j_1:j_2}|\tau_{j_1-1}) \le \frac{|\mathcal{U}_A|}{\alpha} \|x\|_1.
$$

The proof of the above lemma uses the regularity condition in Assumption 3 and Lemma 12. Naively, the product $\prod_{j=j_1}^{j_2} M_{o_j,a_j,j;f}$ may indicate that the norm may grow exponentially. However, the condition that $M_{o,a,h;f}$'s row span belongs to the column span of $K_{h-1;f}$ (which is dereived from Lemma 12) and the fact that $K^\dagger_{h-1;f}$ exists, we have:

$$
\prod_{j=j_1}^{j_2} M_{o_j,a_j,j;f} x = \prod_{j=j_1}^{j_2} M_{o_j,a_j,j;f} K_{j_1-1;f} K^\dagger_{j_1-1;f} x
$$

Thus, we can bound $\| \prod_{j=j_1}^{j_2} M_{o_j,a_j,j;f} K_{j_1-1;f} e_l \|_1$ by using the fact that $K_{j_1-1;f} e_l$ is a predictive state $q_{\tau^l_{j_1-1;f};f}$ corresponding to one of the minimum core histories $\tau^l_{j_1-1;f}$, and $\prod_{j=j_1}^{j_2} M_{o_j,a_j,j;f} \cdot q_{\tau^l_{j_1-1;f};f} \times \pi(\tau_{j_1:j_2}|\tau_{j_1-1}) = [\mathbb{P}(u|\tau^l_{j_1-1;f}, \tau_{j_1:j_2}) \mathbb{P}^{\pi_{\tau_{j_1-1}}}(\tau_{j_1:j_2}|\tau^l_{j_1-1;f})]_{u \in \mathcal{U}_{j_2+1}}$ where $\pi_{\tau_{j_1-1}}$ denote the policy $\pi(\cdot|\tau_{j_1-1})$. Note that the proof of the above lemma differs from the one in POMDPs since here we leverage the concept of minimum core histories and the core matrix which are unique to PSRs. The details are deferred to Appendix J.3.

Therefore, using Lemma 16 with $\pi = \pi^i_{1:h-1} \circ \mathrm{Unif}(\mathcal{A})$, we have

$$
\sum_{i=1}^{k-1} \sum_{\tau_h} \| M^k_{o_h,a_h,h} [b^k_{\tau_{h-1}} - b_{\tau_{h-1}}] \|_1 \times \pi^i(\tau_{h-1})
$$

$$
\le \frac{|\mathcal{A}||\mathcal{U}_A|}{\alpha} \sum_{i=1}^{k-1} \sum_{\tau_{h-1}} \pi^i(\tau_{h-1}) \cdot \| b^k_{\tau_{h-1}} - b_{\tau_{h-1}} \|_1 \le \mathcal{O}(|\mathcal{A}|^2 \sqrt{kH|\mathcal{U}_A|^3 \beta}/\alpha), \tag{27}
$$

where the last step comes from (22).

Combining (26) and (27), we have for all $k \in [T], h \in [H-1]$,

$$
\sum_{i=1}^{k-1} \sum_{\tau_h} \| [M^k_{o_h,a_h,h} - M_{o_h,a_h,h}] b_{\tau_{h-1}} \|_1 \times \pi^i(\tau_{h-1}) \le \mathcal{O}(|\mathcal{A}|^2 \sqrt{kH|\mathcal{U}_A|^3 \beta}/\alpha). \tag{28}
$$

## I.4 STEP 4: CONNECT STEP 2 AND STEP 3

Recall that in Step 2 we want to bound

$$
\sum_{k=1}^{T} \left( \sum_{h=1}^{H-1} \sum_{\tau_h} \| [M^k_{o_h,a_h,h} - M_{o_h,a_h,h}] b_{\tau_{h-1}} \|_1 \times \pi^k(\tau_h) + \| q^k_0 - q_0 \|_1 \right).
$$

First, for the second term, we can bound via (24):

$$\sum_{k=1}^{T} \|q_0^k - q_0\|_1 \leq \mathcal{O}(\sqrt{HT|\mathcal{U}_A|\beta}). \tag{29}$$

Now we only need to bound the first term. Notice that in (28) we have bounded this cumulative estimation error weighted by $\pi^i(\tau_{h-1})$ rather than $\pi^k(\tau_{h-1})$. Here we introduce the following lemma from [37] to bridge these two summations with different weights:

**Lemma 17** ([37, Proposition 22]). *Suppose* $\{x_{k,i}\}_{(k,i)\in[T]\times[n_1]}, \{w_{k,j}\}_{(k,j)\in[T]\times[n_2]} \in \mathbb{R}^d$ *satisfy for all* $k \in [T]$

- $\sum_{t=1}^{k-1} \sum_{i=1}^{n_1} \sum_{j=1}^{n_2} |w_{k,j}^\top x_{t,i}| \leq \gamma_k$,

- $\sum_{i=1}^{n_1} \|x_{k,i}\|_2 \leq R_x$,

- $\sum_{i=1}^{n_2} \|w_{k,i}\|_2 \leq R_w$.

*Then we have for all* $k \in [T]$:

$$\sum_{t=1}^{k} \sum_{i=1}^{n_1} \sum_{j=1}^{n_2} |w_{t,j}^\top x_{t,i}| = \mathcal{O}\Big(d\Big(R_w R_x + \max_{t\leq k}\gamma_t\Big)\log^2(Tn_1)\Big).$$

To apply Lemma 17, for any fixed $h \in [H-1]$, we rewrite (28) in the following way:

$$\sum_{t=1}^{k-1} \sum_{u=1}^{|\mathcal{U}_{h+1}|} \sum_{o,a} \sum_{\tau_{h-1}} \Big|\big[(M_{o,a,h}^k - M_{o,a,h})K_{h-1}\big]_u K_{h-1}^\dagger b_{\tau_{h-1}} \times \pi^t(\tau_{h-1})\Big|$$

$$\leq \mathcal{O}(|\mathcal{A}|^2 \sqrt{kH|\mathcal{U}_A|^3\beta}/\alpha), \tag{30}$$

where $X_u$ is the $u$-th row of the matrix $X$. Here we utilize the fact that $b_{\tau_{h-1}} \times \pi^t(\tau_{h-1}) = (\mathbb{P}[u|\tau_{h-1}]\mathbb{P}^{\pi^t}[\tau_{h-1}])_{u\in\mathcal{U}_h}$ belongs to the column space of $K_{h-1}$ due to the definition of core history.

Then for any $t \in [T], u \in \mathcal{U}_{h+1}, o \in \mathcal{O}, a \in \mathcal{A}$, we let $w_{t,u,o,a}$ denote $\big[(M_{o,a,h}^t - M_{o,a,h})K_{h-1}\big]_u$ and $x_{t,\tau_{h-1}}$ denote $K_{h-1}^\dagger b_{\tau_{h-1}} \times \pi^t(\tau_{h-1})$, then (30) can be written as for any $k \in [T]$

$$\sum_{t=1}^{k-1} \sum_{u\in\mathcal{U}_{h+1}, o\in\mathcal{O}, a\in\mathcal{A}} \sum_{\tau_{h-1}} |w_{k,u,o,a}^\top x_{t,\tau_{h-1}}| \leq \mathcal{O}(|\mathcal{A}|^2\sqrt{kH|\mathcal{U}_A|^3\beta}/\alpha). \tag{31}$$

Now we only need to bound $\sum_{\tau_{h-1}} \|x_{k,\tau_{h-1}}\|_2$ and $\sum_{u\in\mathcal{U}_{h+1}, o\in\mathcal{O}, a\in\mathcal{A}} \|w_{k,u,o,a}\|_2$. For $\sum_{\tau_{h-1}} \|x_{k,\tau_{h-1}}\|_2$, we have

$$\sum_{\tau_{h-1}} \|x_{k,\tau_{h-1}}\|_2 = \sum_{\tau_{h-1}} \|K_{h-1}^\dagger b_{\tau_{h-1}} \times \pi^k(\tau_{h-1})\|_2 = \sum_{\tau_{h-1}} \|K_{h-1}^\dagger[\mathbb{P}(u|\tau_{h-1})\mathbb{P}^{\pi^k}(\tau_{h-1})]_{u\in\mathcal{U}_h}\|_2$$

$$= \sum_{\tau_{h-1}} \|v_{\tau_{h-1}}\|_2 \mathbb{P}^{\pi^k}(\tau_{h-1}) \leq \max_{\tau_{h-1}} \|v_{\tau_{h-1}}\|_2,$$

where the third step comes from the definition of core matrix (2).

Notice that we have $K_{h-1}v_{\tau_{h-1}} = [\mathbb{P}(u|\tau_{h-1})]_{u\in\mathcal{U}_h}$ for any $\tau_{h-1}$ and $\|K_{h-1}^\dagger\|_{1\mapsto 1} \leq 1/\alpha$, which implies

$$\|v_{\tau_{h-1}}\|_2 \leq \|v_{\tau_{h-1}}\|_1 \leq \|K_{h-1}^\dagger[\mathbb{P}(u|\tau_{h-1})]_{u\in\mathcal{U}_h}\|_1 \leq \frac{1}{\alpha}\|[\mathbb{P}(u|\tau_{h-1})]_{u\in\mathcal{U}_h}\|_1 \leq \frac{|\mathcal{U}_A|}{\alpha}.$$

Therefore we have for all $k \in [T]$,

$$\sum_{\tau_{h-1}} \|x_{k,\tau_{h-1}}\|_2 \leq \frac{|\mathcal{U}_A|}{\alpha}. \tag{32}$$

For $\sum_{u \in \mathcal{U}_{h+1}, o \in \mathcal{O}, a \in \mathcal{A}} \|w_{k,u,o,a}\|_2$, we have

$$
\begin{aligned}
\sum_{u \in \mathcal{U}_{h+1}, o \in \mathcal{O}, a \in \mathcal{A}} \|w_{k,u,o,a}\|_2 &\leq \sum_{u \in \mathcal{U}_{h+1}, o \in \mathcal{O}, a \in \mathcal{A}} \|w_{k,u,o,a}\|_1 \\
&= \sum_{o \in \mathcal{O}, a \in \mathcal{A}} \sum_{l=1}^{d_{\mathrm{PSR},h-1}} \|((M_{o,a,h}^k - M_{o,a,h}) K_{h-1} e_l\|_1 \\
&\leq \frac{2|\mathcal{A}||\mathcal{U}_A|}{\alpha} \sum_{l=1}^{d_{\mathrm{PSR},h-1}} \|K_{h-1} e_l\|_1 \leq \frac{2|\mathcal{A}||\mathcal{U}_A|^2 d_{\mathrm{PSR}}}{\alpha},
\end{aligned}
\tag{33}
$$

where the third step utilizes Lemma 16 with uniform policy and the last step utilizes the fact $d_{\mathrm{PSR},h-1} \leq d_{\mathrm{PSR}}$ and $\|K_{h-1} e_l\|_1 = \|q_{\tau_{h-1}^l}\|_1 \leq |\mathcal{U}_A|$.

Invoking Lemma 17 with (32),(33),(31), we can obtain for all $k \in [T], h \in [H-1]$,

$$
\begin{aligned}
\sum_{i=1}^k \sum_{\tau_h} &\|[M_{o_h,a_h,h}^k - M_{o_h,a_h,h}] b_{\tau_{h-1}}\|_1 \times \pi^k(\tau_{h-1}) \\
&\leq \mathcal{O}(d_{\mathrm{PSR},h-1} |\mathcal{U}_A|^3 |\mathcal{A}|^2 d_{\mathrm{PSR}} H^{\frac{3}{2}} k^{\frac{1}{2}} / \alpha^2 \cdot \log(TH\mathcal{N}_\mathcal{F}(\epsilon_\mathrm{b}) |\mathcal{O}||\mathcal{A}|/\delta)).
\end{aligned}
\tag{34}
$$

Substituing (29),(34) into (19), we have

$$
\sum_{k=1}^T \left( V_{f^k}^{\pi^k} - V^{\pi^k} \right) \leq \mathcal{O}(d_{\mathrm{PSR}}^2 H^{\frac{7}{2}} |\mathcal{U}_A|^4 |\mathcal{A}|^2 T^{\frac{1}{2}} \alpha^{-3} \cdot \log(TH\mathcal{N}_\mathcal{F}(\epsilon_\mathrm{b}) |\mathcal{O}||\mathcal{A}|/\delta)).
$$

Combining the above result with Step 1, we have

$$
\sum_{k=1}^T \left( V^* - V^{\pi^k} \right) \leq \mathcal{O}(d_{\mathrm{PSR}}^2 H^{\frac{7}{2}} |\mathcal{U}_A|^4 |\mathcal{A}|^2 T^{\frac{1}{2}} \alpha^{-3} \cdot \log(TH\mathcal{N}_\mathcal{F}(\epsilon_\mathrm{b}) |\mathcal{O}||\mathcal{A}|/\delta)).
$$

This concludes our proof.

## J   PROOFS OF LEMMAS IN APPENDIX I

### J.1   PROOF OF LEMMA 13

To prove $f^* \in \mathcal{B}^k$, we need to show that $\sum_{(\pi,\tau_H) \in \mathcal{D}} \log \mathbb{P}_{f^*}^\pi(\tau_H)$ is large. To simplify writing, we denote the $(\pi, \tau_H)$ pairs in $\mathcal{D}$ at the end of $T$-th iteration by $\{(\pi^i, \tau_H^i)\}_{i=1}^{n_T}$, which are indexed by their collection order. Notice that $n_T \leq TH|\mathcal{U}_A|$. To deal with potentially infinite function clas $\mathcal{F}$, we first consider its minimum $\epsilon_\mathrm{b}$-bracket net $\mathcal{G}$ where $\epsilon_\mathrm{b} = 1/(TH|\mathcal{U}_A|)$ and the set of all upper bound functions in $\mathcal{G}$, i.e., $\mathcal{G}_\mathrm{u} := \{f' : \exists f, \text{ such that } [f, f'] \in \mathcal{G}\}$. Then we are able to bound the difference bewteen $\sum_{(\pi,\tau_H) \in \mathcal{D}} \log \mathbb{P}_{f^*}^\pi(\tau_H)$ and $\sum_{(\pi,\tau_H) \in \mathcal{D}} \log \mathbb{P}_f^\pi(\tau_H)$ for any $f \in \mathcal{F}$ via Cramér-Chernoff's method as in [37].

Fix any $f' \in \mathcal{G}_\mathrm{u}, t \in [n_T]$ and let $\mathfrak{F}_t$ denote the filtration induced by $\{(\pi^i, \tau^i)\}_{i=1}^{t-1} \cup \{\pi^t\}$. We have:

$$
\begin{aligned}
\mathbb{E}\Bigg[ \exp\Bigg( &\sum_{i=1}^t \log\Bigg( \frac{\mathbb{P}_{f'}^{\pi^i}(\tau_H^i)}{\mathbb{P}_{f^*}^{\pi^i}(\tau_H^i)} \Bigg) \Bigg) \Bigg] \\
&= \mathbb{E}\Bigg[ \exp\Bigg( \sum_{i=1}^{t-1} \log\Bigg( \frac{\mathbb{P}_{f'}^{\pi^i}(\tau_H^i)}{\mathbb{P}_{f^*}^{\pi^i}(\tau_H^i)} \Bigg) \Bigg) \cdot \mathbb{E}\Bigg[ \exp\Bigg( \log\Bigg( \frac{\mathbb{P}_{f'}^{\pi^t}(\tau_H^t)}{\mathbb{P}_{f^*}^{\pi^t}(\tau_H^t)} \Bigg) \Bigg) \Big| \mathfrak{F}_t \Bigg] \Bigg] \\
&= \mathbb{E}\Bigg[ \exp\Bigg( \sum_{i=1}^{t-1} \log\Bigg( \frac{\mathbb{P}_{f'}^{\pi^i}(\tau_H^i)}{\mathbb{P}_{f^*}^{\pi^i}(\tau_H^i)} \Bigg) \Bigg) \cdot \mathbb{E}\Bigg[ \frac{\mathbb{P}_{f'}^{\pi^t}(\tau_H^t)}{\mathbb{P}_{f^*}^{\pi^t}(\tau_H^t)} \Big| \mathfrak{F}_t \Bigg] \Bigg] \\
&= \mathbb{E}\Bigg[ \exp\Bigg( \sum_{i=1}^{t-1} \log\Bigg( \frac{\mathbb{P}_{f'}^{\pi^i}(\tau_H^i)}{\mathbb{P}_{f^*}^{\pi^i}(\tau_H^i)} \Bigg) \Bigg) \cdot \sum_{\tau_H} P_{f'}^{\pi^t}(\tau_H) \Bigg]
\end{aligned}
$$

$$\leq \mathbb{E}\left[\exp\left(\sum_{i=1}^{t-1}\log\left(\frac{\mathbb{P}_{f'}^{\pi^i}(\tau_H^i)}{\mathbb{P}_{f^*}^{\pi^i}(\tau_H^i)}\right)\right)\cdot\left(1+\frac{1}{TH|\mathcal{U}_A|}\right)\right],$$

where the last step is due to the fact that $\mathcal{G}$ is the minimum $\epsilon_{\mathrm{b}}$-bracket net, which implies that there exists $f \in \mathcal{F}$ such that $\|\mathbb{P}_f^\pi(\cdot) - \mathbb{P}_{f'}^\pi(\cdot)\|_1 \leq \epsilon_{\mathrm{b}}$ for any policy $\pi$ and thus $\|\mathbb{P}_{f'}^\pi(\cdot)\|_1 \leq 1 + \epsilon_{\mathrm{b}}$. Repeat the above arguments and we have

$$\mathbb{E}\left[\exp\left(\sum_{i=1}^{t}\log\left(\frac{\mathbb{P}_{f'}^{\pi^i}(\tau_H^i)}{\mathbb{P}_{f^*}^{\pi^i}(\tau_H^i)}\right)\right)\right] \leq e.$$

Then by Markov's inequality we have for any $\delta \in (0,1]$,

$$\mathbb{P}\left(\sum_{i=1}^{t}\log\left(\frac{\mathbb{P}_{f'}^{\pi^i}(\tau_H^i)}{\mathbb{P}_{f^*}^{\pi^i}(\tau_H^i)}\right) > \log(1/\delta)\right)$$

$$\leq \mathbb{E}\left[\exp\left(\sum_{i=1}^{t}\log\left(\frac{\mathbb{P}_{f'}^{\pi^i}(\tau_H^i)}{\mathbb{P}_{f^*}^{\pi^i}(\tau_H^i)}\right)\right)\right]\cdot\exp[-\log(1/\delta)] \leq e\delta.$$

Therefore by union bound, we have for all $f' \in \mathcal{G}_{\mathrm{u}}, t \in [n_T]$,

$$\mathbb{P}\left(\sum_{i=1}^{t}\log\left(\frac{\mathbb{P}_{f'}^{\pi^i}(\tau_H^i)}{\mathbb{P}_{f^*}^{\pi^i}(\tau_H^i)}\right) > c\log(\mathcal{N}_{\mathcal{F}}(\epsilon_{\mathrm{b}})TH|\mathcal{U}_A|/\delta)\right) \leq \delta/2,$$

where $c$ is a universal constant.

Finally, due to the definition of $\epsilon$-bracket net, we know for all $f \in \mathcal{F}$, there exists $f' \in \mathcal{G}_{\mathrm{u}}$ such that $\mathbb{P}_f^\pi(\tau_H) \leq \mathbb{P}_{f'}^\pi(\tau_H)$ for any trajectory $\tau_H$ and policy $\pi$. Therefore we have for all $f \in \mathcal{F}, t \in [n_T]$,

$$\mathbb{P}\left(\sum_{i=1}^{t}\log\left(\frac{\mathbb{P}_{f}^{\pi^i}(\tau_H^i)}{\mathbb{P}_{f^*}^{\pi^i}(\tau_H^i)}\right) > c\log(\mathcal{N}_{\mathcal{F}}(\epsilon_{\mathrm{b}})TH|\mathcal{U}_A|/\delta)\right) \leq \delta/2,$$

which implies that $f^* \in \mathcal{B}^k$ for all $k \in [T]$ with probability at least $1 - \delta/2$. This concludes our proof.

## J.2 PROOF OF LEMMA 14

First, notice that we can decompose the left hand side of (18) into the following sequence of terms via triangle inequality:

$$\sum_{\tau_h}\left\|\prod_{l=1}^{h}M_{o_l,a_l,l}^k\cdot q_0^k - \prod_{l=1}^{h}M_{o_l,a_l,l}\cdot q_0\right\|_1 \times \pi(\tau_h)$$

$$\leq \sum_{j=1}^{h}\sum_{\tau_h}\left\|\prod_{l=j+1}^{h}M_{o_l,a_l,l}^k\left(M_{o_j,a_j,j}^k - M_{o_j,a_j,j}\right)\cdot b_{\tau_{j-1}}\right\|_1 \times \pi(\tau_h)$$

$$+ \sum_{\tau_h}\left\|\prod_{l=1}^{h}M_{o_l,a_l,l}^k\left(q_0^k - q_0\right)\right\|_1 \times \pi(\tau_h). \tag{35}$$

Then fix $j \in [h]$ and consider the term $\sum_{\tau_h}\left\|\prod_{l=j+1}^{h}M_{o_l,a_l,l}^k\left(M_{o_j,a_j,j}^k - M_{o_j,a_j,j}\right)\cdot b_{\tau_{j-1}}\right\|_1 \times \pi(\tau_h)$ in (35). We have

$$\sum_{\tau_h}\left\|\prod_{l=j+1}^{h}M_{o_l,a_l,l}^k\left(M_{o_j,a_j,j}^k - M_{o_j,a_j,j}\right)\cdot b_{\tau_{j-1}}\right\|_1 \times \pi(\tau_h)$$

$$= \sum_{\tau_j}\pi(\tau_j)\sum_{\tau_{j+1:h}}\left\|\prod_{l=j+1}^{h}M_{o_l,a_l,l}^k\left(M_{o_j,a_j,j}^k - M_{o_j,a_j,j}\right)\cdot b_{\tau_{j-1}}\right\|_1 \times \pi(\tau_{j+1:h}|\tau_j)$$

$$\leq \frac{|\mathcal{U}_A|}{\alpha} \sum_{\tau_j} \left\| \left( M^k_{o_j,a_j,j} - M_{o_j,a_j,j} \right) \cdot b_{\tau_{j-1}} \right\|_1 \cdot \pi(\tau_j), \tag{36}$$

where the last step comes from Lemma 16.

Similarly, apply Lemma 16 to the second part of (35) and we have

$$\sum_{\tau_h} \left\| \prod_{l=1}^{h} M^k_{o_l,a_l,l} \left( q_0^k - q_0 \right) \right\|_1 \times \pi(\tau_h) \leq \frac{|\mathcal{U}_A|}{\alpha} \| q_0^k - q_0 \|_1. \tag{37}$$

Substituting (36) and (37) into (35), we can obtain

$$\sum_{\tau_h} \left\| \prod_{l=1}^{h} M^k_{o_l,a_l,l} \cdot q_0^k - \prod_{l=1}^{h} M_{o_l,a_l,l} \cdot q_0 \right\|_1 \times \pi(\tau_h)$$
$$\leq \frac{|\mathcal{U}_A|}{\alpha} \left( \sum_{l=1}^{h} \sum_{\tau_l} \| [M^k_{o_l,a_l,l} - M_{o_l,a_l,l}] b_{\tau_{l-1}} \|_1 \times \pi(\tau_l) + \| q_0^k - q_0 \|_1 \right).$$

This concludes our proof.

## J.3 PROOF OF LEMMA 16

First, based on Lemma 12, we have chosen $m_{(o,a,u),j_1;f}$ to belong to the column space of $K_{j_1-1;f}$, which implies that

$$\sum_{\tau_{j_1:j_2}} \left\| \prod_{j=j_1}^{j_2} M_{o_j,a_j,j;f} x \right\|_1 \times \pi(\tau_{j_1:j_2}|\tau_{j_1-1})$$
$$= \sum_{\tau_{j_1:j_2}} \left\| \left( \prod_{j=j_1}^{j_2} M_{o_j,a_j,j;f} K_{j_1-1;f} \right) \left( K^\dagger_{j_1-1;f} x \right) \right\|_1 \times \pi(\tau_{j_1:j_2}|\tau_{j_1-1}).$$

Note that since the $l$-th column of $K_{j_1-1;f}$ is $q_{\tau^l_{j_1-1;f};f}$, the $l$-th core history at step $j_1-1$ under the model induced by $f$, we have for any $l \in [d_{\mathrm{PSR},j_1-1;f}]$,

$$\sum_{\tau_{j_1:j_2}} \left\| \left( \prod_{j=j_1}^{j_2} M_{o_j,a_j,j;f} K_{j_1-1;f} e_l \right) \right\|_1 \times \pi(\tau_{j_1:j_2}|\tau_{j_1-1})$$
$$= \sum_{o_{j_1:j_2}} \sum_{a_{j_1:j_2}} \left\| \left( \prod_{j=j_1}^{j_2} M_{o_j,a_j,j;f} K_{j_1-1;f} e_l \right) \right\|_1 \pi((o_{j_1:j_2}, a_{j_1:j_2})|\tau_{j_1-1})$$
$$= \sum_{o_{j_1:j_2}} \sum_{a_{j_1:j_2}} \sum_{u \in \mathcal{U}_{j_2+1}} \mathbb{P}_f(u|(\tau^l_{j_1-1}, o_{j_1:j_2}, a_{j_1:j_2})) \mathbb{P}_f(o_{j_1:j_2}|\tau^l_{j_1-1;f}; \mathrm{do}(a_{j_1:j_2-1})) \pi((o_{j_1:j_2}, a_{j_1:j_2})|\tau_{j_1-1})$$
$$= \sum_{o_{j_1:j_2}} \sum_{a_{j_1:j_2}} \left\{ \sum_{u \in \mathcal{U}_{j_2+1}} \mathbb{P}_f(u|(\tau^l_{j_1-1}, o_{j_1:j_2}, a_{j_1:j_2})) \right\} \mathbb{P}_f(o_{j_1:j_2}|\tau^l_{j_1-1;f}; \mathrm{do}(a_{j_1:j_2-1})) \pi((o_{j_1:j_2}, a_{j_1:j_2})|\tau_{j_1-1})$$
$$\leq |\mathcal{U}_A| \sum_{o_{j_1:j_2}} \sum_{a_{j_1:j_2}} \mathbb{P}_f(o_{j_1:j_2}|\tau^l_{j_1-1;f}; \mathrm{do}(a_{j_1:j_2-1})) \pi((o_{j_1:j_2}, a_{j_1:j_2})|\tau_{j_1-1})$$
$$\leq |\mathcal{U}_A|.$$

Here in the second step $\pi((o_{j_1:j_2}, a_{j_1:j_2})|\tau_{j_1-1})$ denotes $\prod_{j=j_1}^{j_2} \pi(a_j|\tau_{j_1-1}, o_{j_1:j}, a_{j_1:j-1})$ and the third step comes from Lemma 4.

Therefore we have

$$\sum_{\tau_{j_1:j_2}} \left\| \left( \prod_{j=j_1}^{j_2} M_{o_j,a_j,j;f} K_{j-1;f} \right) \left( K_{j_1-1;f}^\dagger x \right) \right\|_1 \times \pi(\tau_{j_1:j_2} | \tau_{j_1-1})$$

$$\leq |\mathcal{U}_A| \left\| K_{j_1-1;f}^\dagger x \right\|_1 \leq \frac{|\mathcal{U}_A|}{\alpha} \|x\|_1,$$

where the third step comes from Assumption 3. This concludes our proof.

## K NECESSITY OF $1/\alpha$ IN THEOREM 1

In this section we show that the polynomial dependence on $1/\alpha$ in Theorem 1 is inevitable in general. More specifically, we have the following theorem:

**Theorem 3.** *For any $0 < \alpha < \frac{1}{2\sqrt{2}}$ and $H, |\mathcal{A}| \in \mathbb{N}^+$, there exists a PSR with core test set $\mathcal{U}_h = \mathcal{O}$ for $h \in [H]$ and $|\mathcal{S}| = |\mathcal{O}| = \mathcal{O}(1)$ which satisfies Assumption 1 so that any algorithm requires at least $\Omega(\min\{\frac{1}{\alpha H}, |\mathcal{A}|^{H-1}\})$ samples to learn a $(1/2)$-optimal policy with probability $1/6$ or higher.*

Theorem 3 indicates that scaling with $1/\alpha$ is unavoidable or else the algorithm will require an exponential number of samples to learn a near optimal policy. The proof is deferred to Appendix K.1.

### K.1 PROOF OF THEOREM 3

We leverage the hard instance constructed in [37] to prove the lower bound, which is based on combinatorial lock. More specifically, we define a POMDP as follows:

- **State space:** There are two states, $\mathcal{S} = \{s_g, s_b\}$.
- **Observation space and emission matrices:** There are three observations, $\mathcal{O} = \{o_g, o_b, o_{\text{dummy}}\}$. For $h \in [H-1]$, we define the emission matrix as follows:

$$\mathbb{O}_h = \begin{pmatrix} \sqrt{2}\alpha & 0 \\ 0 & \sqrt{2}\alpha \\ 1 - \sqrt{2}\alpha & 1 - \sqrt{2}\alpha \end{pmatrix}.$$

For $h = H$, we have

$$\mathbb{O}_H = \begin{pmatrix} 1 & 0 \\ 0 & 1 \\ 0 & 0 \end{pmatrix}.$$

This means that with probability $\alpha$ we can observe the current state and with probability $1 - \alpha$ we only receive a dummy observation at step $h \in [H-1]$. At step $H$, though, we are able to observe the current state.

- **Action space and transition kernels:** There are $|\mathcal{A}|$ actions and the initial state is fixed as $s_g$. For each step $h \in [H-1]$, there exists a good action $a_{g,h} \in \mathcal{A}$ which is chosen uniformly at random from $\mathcal{A}$ such that if the agent is currently in $s_h = s_g$ and takes $a_{g,h}$, it will stay in $s_g$, i.e., $s_{h+1} = s_g$. Otherwise, the agent will always go to $s_{h+1} = s_b$.
- **Reward:** We define $r_h(o) = 0$ for all $h \in [H-1]$ and $o \in \mathcal{O}$. At step $H$, $r_H(o_g) = 1$ while $r_H(o_b) = 0$. This indicates that the agent will receive reward 1 iff the agent takes $a_{g,h}$ along its way.

Since this POMDP satisifes weakly-revealing condition, we know $\mathcal{O}$ is its core test set. Next we show that this POMDP satisfies Assumption 1. First it is can be observed that $K_0 = q_0 = (\sqrt{2}\alpha, 0, 1 - \sqrt{2}\alpha)^\top$ and we can verify that

$$\|K_0^\dagger\|_{1 \mapsto 1} \leq 1/\alpha.$$

Then for any $h \in [H-1]$ and reachable history $\tau_h$, if $a_{1:h} = a_{g,1:h}$, we have

$$\mathbb{P}(s_{h+1} = s_g | \tau_h) = 1, \mathbb{P}(s_{h+1} = s_b | \tau_h) = 0,$$

which implies that

$$\mathbb{P}(o_{h+1} = o_g | \tau_h) = \sqrt{2}\alpha, \mathbb{P}(o_{h+1} = o_b | \tau_h) = 0, \mathbb{P}(o_{h+1} = o_{\text{dummy}} | \tau_h) = 1 - \sqrt{2}\alpha.$$

Otherwise, if there exists $h' \in [h]$ such that $a_{h'} \neq a_{g,h'}$, then we have

$$\mathbb{P}(s_{h+1} = s_b | \tau_h) = 1, \mathbb{P}(s_{h+1} = s_g | \tau_h) = 0,$$

which implies

$$\mathbb{P}(o_{h+1} = o_b | \tau_h) = \sqrt{2}\alpha, \mathbb{P}(o_{h+1} = o_g | \tau_h) = 0, \mathbb{P}(o_{h+1} = o_{\text{dummy}} | \tau_h) = 1 - \sqrt{2}\alpha.$$

This suggests that $K_h = \mathbb{O}_{h+1}$ for $h \in [H-1]$. On the otherhand, since $\sigma_{\min}(K_h) = \sigma_{\min}(\mathbb{O}_{h+1}) \geq \sqrt{2}\alpha$, we have for $h \in [H-1]$,

$$\|K_h^\dagger\|_{1 \mapsto 1} \leq \sqrt{2}\|K_h^\dagger\|_{2 \mapsto 2} \leq \sqrt{2}/(\sqrt{2}\alpha) = 1/\alpha.$$

This shows that the constructed POMDP satisfies Assumption 1.

Now we only need to show that the constructed POMDP attains the lower bound in Theorem 3. This has been proved in [37] and we include the proof here for completeness.

Suppose we can only interact with the POMDP for $T \leq \lfloor \frac{1}{2\sqrt{2}\alpha H} \rfloor$ episodes. Then we know the probability that both $s_g$ and $s_b$ only emit $o_{\text{dummy}}$ in the first $H-1$ steps for all $T$ episodes is lower bounded by $(1 - \sqrt{2}\alpha)^{1/(\sqrt{2}\alpha)}$ since $2 \cdot \lfloor \frac{1}{2\sqrt{2}\alpha H} \rfloor \cdot (H-1) \leq 1/(\sqrt{2}\alpha)$.

Now conditioned on the event that both $s_g$ and $s_b$ only emit $o_{\text{dummy}}$ in the first $H-1$ steps for all $T$ episodes, we can only random guess the optimal action sequence $a_{g,1:H-1}$. Then if $T \leq |\mathcal{A}|^{H-1}/10$, the probability that we fail to guess the optimal action sequence is

$$\binom{|\mathcal{A}|^{H-1} - 1}{T} \bigg/ \binom{|\mathcal{A}|^{H-1}}{T} \geq 0.9,$$

Therefore, with probability $0.9 \times (1 - \sqrt{2}\alpha)^{1/(\sqrt{2}\alpha)} \geq 1/6$, the agent can only learn that the action sequences it chooses in these $T$ episodes is incorrect, which implies that the agent can only random guess from the remained action sequences. Therefore, if $T \leq |\mathcal{A}|^{H-1}/10$, the policy that the agent outputs will be worse than $1/2$-optimal, which concludes our proof.

