# OpenReview forum: "PAC Reinforcement Learning for Predictive State Representations"
_ICLR.cc/2023/Conference — ICLR 2023 poster_

### Official Review · Reviewer_gdmT · 2022-10-24

**Confidence:** 5
**Correctness:** 4
**Technical Novelty And Significance:** 3
**Empirical Novelty And Significance:** Not applicable
**Recommendation:** 8

**Clarity, Quality, Novelty And Reproducibility:**

The paper is quite well-written and studies an important and under-explored problem.

**Strength And Weaknesses:**

*Strength*

- The paper extended a recent breakthrough in POMDPs to a more general framework, which is a good addition to literature.

- The positioning of the paper is good -- well-explained related works, how the more general framework can imply previous results, etc.


*Weakness*

- Given the work by (Liu et al., 2022), it is predictable that the same idea would work in the PSR framework. All POMDP systems are equivalent up to some similarity transformation of belief states (say, as argued in (Boots et al., 2011)), and in a slightly older work by Liu (Liu et al., 2020), the PSR representation of POMDPs was explicitly used. The scope presented in (Liu et al., 2022) is not really because it can only work for POMDPs -- confidence set based on MLE can naturally cover all equivalent systems up to the belief state transformation. So novelty of the paper is limited.

- It would have been helpful how the authors could get around the issue that there is actually no concept of latent states "S". In (Liu et al., 2022), it seemed that anchoring to latent states helps a lot to make analysis easier. I could imagine that such concept might not be necessary, but it would have been nice if the authors could elaborate more kindly and crisply how they could get around the issue.


**Summary Of The Paper:**

The submission considers a sample-efficient learning for PSRs. The nature of the problem is similar to (Liu et al. 2022), and previous algorithmic framework is extended from POMDPs to PSRs. This generalization to PSR seems relatively easy since the core test-set is provided a priori, and the construction of confidence set based on log-likelihood values can remain the same (as long as we know some rough upper bounds of the rank of PSRs). Results for the most general framework PSR can imply several previous results including tabular POMDPs (Liu et al., 2022) and m-step decodable POMDPs (Efroni et al., 2022), and low-rank POMDPs. Proof ideas and techniques are mostly adopted from (Liu et al., 2022), though it is not entirely straight-forward to make things work for PSRs.

**Summary Of The Review:**

Overall, despite some concerns on technical novelty, I think that this paper makes a good contribution, and could be a nice addition to POMDP literature.

---

> ### Author Response · Authors · 2022-11-10
> **Response**
>
> Thank you for the valuable feedback! Here are our responses:
>
> - **Novelty Compared with [37]**:
>
> We would like to point out our novelty and significance given [37] in the following three aspects.
>
> First, the OOM formulation of PSRs in [6] is **not equivalent** to the OOM formulation of POMDPs in [29,37]. The OOM formulation of POMDPs in [29,37] has the special structure that $\mathbb{B} _ {o,a,h} = \mathbb{O} _ {h+1} \mathbb{T} _ {h,a} \text{diag} (\mathbb{O} _ {h}(o|\cdot)) \mathbb{O} _ {h} ^ {\dagger}$ and their analysis all relies on this special structure. However, for PSRs, the predictive operator matrix can have arbitrary structures, so the analysis in [29,37] no longer applies and it is not straightforward to show that MLE also works for PSRs.
>
> Second, OMLE in [37] only considers tabular POMDPs and models the transition and emission kernels directly while CRANE is able to tackle exponentially large observation space with **general function approximation** and model PSR via the **predictive operator matrix** $M$, leading to a potentially more compact representation for general PSRs. Further, with general function approximation, we need to bound the **bracket number** of the function class for several common models including tabular PSRs and linear POMDPs as in Appendix G, which also requires new techniques and is not considered in [37].
>
>
> Third, we want to point out that our general result allows us to **derive PAC bounds for special models that are not considered by [37]**. For example,
> - for **weakly-revealing low-rank POMDPs** (Appendix A.2), our result is state-of-art, compared to [47] which requires additional structural assumptions. Thanks to our general results on PSRs, getting PAC bounds for low-rank POMDP becomes straightforward. One cannot instantiate prior work from [37] on discrete POMDP to low-rank POMDP;
>
> - for **linear POMDPs** (Appendix A.4), our results apply to the overcomplete setting while the literature [7] only considers the undercomplete setting and requires other assumptions such as a linear structure on the state distribution conditioned on future observations apart from linear transition and emission kernels;
>
> - for **$m$-step decodable POMDPs** (Appendix A.5), our result only scales with $poly(O)$ while the model-free algorithms in the literature [13] need $poly(O ^ m)$. This is a big difference when the observation space is large.
>
> We defer these discussions to the Appendix due to the space limit and we apologize for the inconvenience. So in summary, our general results on PSRs allow us to derive state-of-art or improved PAC bounds for various interesting settings in a straightforward manner. These models cannot be directly captured by prior work from [37]. We think this is a significant contribution to the RL literature.
>
> - **Novelty in Analysis Techniques**:
>
> This is an excellent point. Without the assumption of the existence of latent states, the existing analysis techniques in [37] cannot be applied here anymore. We need to first find an efficient way to **characterize the system dynamics**, i.e., Lemma 4 where we discover that the predictive operator matrices are sufficient for this task. Then we need to show MLE still works when the observation matrices and transition matrices may not exist. This is also quite different from [37] because their assumptions and analysis all rely on the **OOM-type representation of POMDPs** which requires the existence of observation matrices in particular. For example, when **bounding norm of the product of predictive operator matrices** (or OOMs in POMDPs), which is the key step of the proof, [37] **directly** inserts $ O _ h O _ h ^{\dagger}$ since each OOM ends with a $O _ h^{\dagger}$. In contrast, we **cannot do** this since the predictive operator matrices can have arbitrary structure. Instead, we identify that the **core matrices** play an essential role in characterizing the system and prove that we can always find a group of predictive operator matrices such that we can insert $ K _ h K _ h ^ {\dagger}$ and bound their norm (Lemma 12 and 16). Finally, when we **bound the norm of the vectors $x _ {k,i}$** to apply Lemma 17, we also have to use a new method that leverages the definition of core matrices while the bound is straightforward in [37] since there $x _ {k,i}$ is a distribution. This also leads to the dependence on $|U _ A| ^ 4$ in our sample complexity as illustrated in Theorem 2 (the sample complexity for POMDPs can be reduced to $ |U_ A| ^ 3$).

---

> > ### Comment · Reviewer_gdmT · 2022-12-03
> > **Update**
> >
> > Thank you for your rebuttal. I overlooked the fact that the paper also considers general function approximation. After carefully reading other reviews and responses, I now agree that the paper is also technically more involved. I still think that the overall ideas and analysis techniques are not drastically different from the O-MLE algorithm (Liu et al., 2022), but it seems fairly non-trivial to carry out the full-analysis in PSR settings (wish such points are more clearly pointed out). I raised my score.

---

> > > ### Author Response · Authors · 2022-12-03
> > > **Thank you**
> > >
> > > Thank you very much!

---

### Official Review · Reviewer_YxZB · 2022-10-24

**Confidence:** 3
**Correctness:** 3
**Technical Novelty And Significance:** 2
**Empirical Novelty And Significance:** Not applicable
**Recommendation:** 5

**Clarity, Quality, Novelty And Reproducibility:**

The problem of partial observability is well motivated and discussed. The paper does a pretty good job of discussing and contrasting itself with related work. However, the clarity could be improved in some parts.
- The notation is very complicated and hard to follow. I’m not sure if there is a good solution for this problem setting, but it should be noted at least. However, there are some obvious things that should be fixed like overloaded $K$.
- Various minor claims throughout seem to have insufficient justifications (see previous section)
- The spacing seems way too compact for the ICLR format.
- There are some typos throughout the paper. Examples: Contribution 2, ‘Theorem 1 indicate’, ‘wealy-revealing’, etc.

**Strength And Weaknesses:**

Strengths:
- Few papers have studied efficient methods for RL in the PSR setting. This seems to be the first to do this comprehensively.
- A new OMLE algorithm is presented to work with PSR models generally, which encompass many other partially observable models.
- The paper does a good job of demonstrating how their methods can be applied to many existing problem settings.
- The beginning discussion is both clear and well-written and contextualizes the work.
- The theory seems good.

Weaknesses:
- The algorithm is pretty similar to existing “OMLE” style algorithms, so the algorithmic novelty is limited.
- To implement the algorithm, it seems a large amount of information must be known about the given problem such as the core test set, the singular value, preconditions in Assumption 3, etc. This begs the question of whether it might just be practically easier to implement existing algorithms for a specific problem, given that one essentially has to know the problem setting anyway.
- The main selling point of this paper is the generality of PSRs. However, it’s unclear whether this generality affords significantly new insights into the problem. All of the examples given are settings for which efficient, simple algorithms already exist. The result would be stronger if there are interesting, new problems settings where this generality produces new, previously unknown results or much better results than the existing literature. But so far this evidence is either not provided or it is not clear.


Other:
- In the beginning it’s motivated that PSRs can be solved via spectral methods in contrast to difficult to optimize EM-style algorithms, but Alg 1 looks like a very difficult MLE problem. Can the authors comment on this?
- It claimed that Liu et al’s algorithm needs the latent state representation known, but I doubt this is actually a fundamental limitation of their algorithm and analysis and it is more so done for simplification. The concurrent work of Liu et al (2022) seems to support my suspicion as well.
- I do not think it is sufficient to state the main results as just poly( problem inputs ). This makes it very hard to compare across papers.
- How large can we expect U_A to be in general?
- “Assumption 3 can be easily satisfied by eliminating those functions which do not satisfy the regularity or validity.” I’m not confident this is an easy task in practice.
- On Page 6 in the justification of the bracket, it says P is lipschitz but with what constant?

Liu et al “Optimistic MLE—A Generic Model-based Algorithm for Partially Observable Sequential Decision Making”


**Summary Of The Paper:**

This paper studies the problem of RL with PSRs. The PSR setting is first motivated by its generality and structural properties. Then, a new algorithm, CRANE, is proposed, which is an optimistic MLE algorithm. It takes as input a model class (to represent a class of PSR) and solves an MLE objective to maintain an uncertainty set. Applications of the results are presented on existing problem settings such as weakly revealing POMPDs and decodable POMDPs.

**Summary Of The Review:**

This paper has a lot of potential, but there are still shortcomings in the significance, algorithmic novelty and overall clarity in its discussions. In particular, it’s unclear what analytic insights can be gleaned from the PSR analysis over existing work.

---

> ### Author Response · Authors · 2022-11-10
> **Response (Part 1)**
>
> Thank you for the valuable feedback! Here are our responses:
>
> - **Comparison with OMLE [37]**:
>
> We agree that our procedure follows OMLE [37]. The difference is that OMLE only considers tabular POMDPs and models the transition and emission kernels directly while CRANE is able to tackle exponentially large observation space with **general function approximation** and model PSR via the **predictive operator matrix** $M$, leading to a potentially more compact representation for general PSRs. Further, with general function approximation, we need to bound the **bracket number** of the function class for several common models including tabular PSRs and linear POMDPs as in Appendix G, which also requires new techniques and is not considered in [37].
>
> We also want to point out that such an optimistic model-based learning framework is common in MDPs and POMDPs and is a standard approach for obtaining PAC bounds for model-based RL. Our novelty is really coming from **analysis and its integration with general function approximation**, i.e., showing such MLE-based optimistic model-based learning with function approximation works for PSRs – a much more general class than POMDPS and MDPs.  More specifically, without the assumption of the existence of latent states, the existing analysis techniques for OMLE cannot be applied here anymore. We need to first find an efficient way to **characterize the system dynamics**, i.e., Lemma 4 where we discover that the predictive operator matrices are sufficient for this task. Then we need to show MLE still works when the observation matrices and transition matrices may not exist. This is also quite different from the OMLE analysis because their assumptions and analysis all rely on the **OOM-type representation of POMDPs** which requires the existence of observation matrices in particular. For example, when **bounding norm of the product of predictive operator matrices** (or OOMs in POMDPs), which is the key step of the proof, they can **directly** insert $ O _ h O _ h ^{\dagger}$ since each OOM ends with a $O _ h^{\dagger}$. In comparison, we **cannot do** this since the predictive operator matrices can have arbitrary structure. Instead, we identify that the **core matrices** play an essential role in characterizing the system and prove that we can always find a group of predictive operator matrices such that we can insert $ K _ h K _ h ^ {\dagger}$ and bound their norm (Lemma 12 and 16). Finally, when we **bound the norm of the vectors $x _ {k,i}$** to apply Lemma 17, we also have to use a new method that leverages the definition of core matrices while the bound is straightforward in OMLE analysis since there $x _ {k,i}$ is a distribution. This also leads to the dependence on $|U _ A| ^ 4$ in our sample complexity as illustrated in Theorem 2 (the sample complexity for OMLE can be reduced to $|U_ A| ^ 3$).
>
>
> - **Information Required to Know**:
>
> We admit that the information mentioned in the review is required. However, the core test set is known in advance for some common models (like weakly-revealing POMDPs and decodable POMDPs). In the literature on PSRs, it is standard to assume the core test set is known as assumed in the landmark paper in PSRs [6] . Note that we do not need to know the “minimum core test sets” in our algorithm and analysis. Assumption 3 can be easily satisfied too if we know an upper bound of the minimal singular value of $K _ h$. In addition, the requirement of knowing the upper bound of the minimal singular value is common in the POMDP literature like OMLE [Assumption 1, 37].

---

> > ### Author Response · Authors · 2022-11-10
> > **Response (Part 2)**
> >
> > - **New Insights Coming from the Generality of PSRs**:
> >
> > We want to clarify that the point that “All of the examples given are settings for which efficient, simple algorithms already exist” is **not true** indeed. As the reviewer pointed out (“The result would be stronger if there are interesting, new problems settings where this generality produces new, previously unknown results or much better results than the existing literature”), we actually **do have interesting new results** when applying our general theorem (Theorem 2) to existing models. We defer these discussions to the Appendix due to the space limit and we apologize for the inconvenience. We will try to include it more in the main text. Some of the new results are listed as below:
> >
> > First, for **weakly-revealing low-rank POMDPs** (Appendix A.2), our results are more general than the existing work [47] since we do not need the assumption of bottleneck variables. We also avoid the past sufficiency assumption in the literature (which implies you can infer the bottleneck variable from short histories) and only require the regularity on core matrices (which is similar to the assumption of bounded future sufficiency in [47]). Note that the algorithm in [47] is not computationally efficient. Our result on low-rank POMDP is state-of-art.
> >
> > Second, for **linear POMDPs** (Appendix A.4), our results apply to the overcomplete setting while the literature [7] only considers the undercomplete setting and requires other assumptions such as a linear structure on the state distribution conditioned on future observations apart from linear transition and emission kernels.  The algorithm in [7] is not computationally efficient.
> >
> > Third, for **$m$-step decodable POMDPs** (Appendix A.5), our result only scales with $poly(O)$ while the model-free algorithms in the literature [13] need $poly(O ^ m)$. This is a big difference when the observation space is large. Note the algorithm in [13] is not computationally efficient either.
> >
> >
> > - **Other Details**:
> >
> > **Spectral methods**: The spectral methods in the literature only apply when there is an exploratory dataset, which is not the setting in our paper – we need to do active exploration.
> >
> > **Comparison with Liu's works**: See our response for comparison with OMLE [37]. For the concurrent work, we want to clarify that they directly assume the norm of the product of predictive operator matrices is bounded. This assumption is more abstract since the predictive operator matrices are not unique (see Lemma 12 in our paper) and this assumption is closely tied with the specific algorithm and the analysis. We think the bound on the core matrix is more straightforward since the core matrix is really a fundamental concept determined by the system dynamics itself.  Also we want to point out to the reviewer that the concurrent work (Liu et al 2022) appeared on arXiv later than ICLR deadline.
> >
> > **More concrete sample complexity**: We apologize for the inconvenience. Our intention is to make our results in the main paper clearer, and we defer the more concrete sample complexity to Theorem 2 in Appendix I (in the format of regret upper bound, but you can easily obtain a more concrete sample complexity from the regret) where we provide the exact dependence of the parameters. As for the comparison between the results in Theorem 2 and POMDP literature, our dependence on $|U _ A|$ is slightly worse because when applying eluder-dimension to bound the regret, we need to bound the norm in Equation 32 while for POMDPs you can get a smaller bound leveraging the existence of latent states.
> >
> > **How large is $|U_A|$**: This depends on the specific problem. For $m$-step weakly-revealing or decodable PODMPs, $|U _ A|$ is typically $A ^ m$. There is also a lower bound in [37] showing that such a polynomial dependence on $A ^ m$ is inevitable in $m$-step weakly-revealing PODMPs.
> >
> > **Assumption 3**: We would like to clarify that our paper is more focused on theoretical analysis and Assumption 3 can be achieved in theory as long as the realizability is satisfied. In addition, for some special cases like POMDPs, we can let the function class model the transition and emission kernels such that the validity and regularity can be achieved more easily.
> >
> > **Lipschitz constant for bracket number**: Please see Lemma 11 in Appendix G.3. We apologize for the inconvenience of deferring some proofs and details to the Appendix.
> >
> > **Notation and Typo**: Thanks for pointing out the typos! We have corrected them in the revised version. For the notations, we indeed have a notation table (Table 3 in Appendix C) to help understanding.

---

> > > ### Comment · Reviewer_YxZB · 2022-11-21
> > > **Thank you**
> > >
> > > Thanks for your response. This has cleared up many concerns, especially regarding the low rank and linear cases. The generality of this paper compared to the extra conditions of [47] and conditions in [7] is significant. I have a few more questions if you could answer.
> > >
> > > I think the important thing to understand given prior work is when $1/\alpha$ might be large or not.
> > > - It is discussed in the appendix section A that the bounded norm of [37] is weaker than assumption 1. Then could assumption 1 be excluding even certain weakly revealing tabular POMDPs if this 1/alpha is not small? Can we be sure that there isn’t a large class of such important problems that cannot be handled under this analysis?
> > > - Similar question for the decodable POMDPs. While the O^m to O is clearly good over [13], can we be sure it is not being absorbed into the $1/\alpha$?
> > >   - Follow up: [13] later avoids O^m assuming realizability and completeness. Could you avoid large $1/\alpha$ for this same case as [13] by choosing better models for PSR?
> > > - Also for [47] the lack of assumption on bottleneck variables is nice for CRANE but again could it be that problems without this explicit bottleneck structure also implicitly have very large $1/\alpha$? Do low rank problems with low $1/\alpha$ contain the class of problems in [47] with low $\nu$ and $\gamma$ for them?
> > >
> > > I think Theorem 3 partly answers these but looks worst case. These questions are more about instances.

---

> > > > ### Author Response · Authors · 2022-11-22
> > > > **Further Response**
> > > >
> > > > Thank you for the reply!
> > > >
> > > > First, for weakly-revealing POMDPs, we admit that $1/\alpha$ can be large for some POMDP models and you are correct that $1/\alpha$ is slightly stronger than the norm assumption in [37]. However, this **does not imply** that our bounds are worse than [37]. This is because with the assumption on the regularity of the core matrix instead of the observation matrix, our sample complexity can scale with $d _ {PSR}$, which is always no larger than $|\mathcal{S}|$ and can be much smaller e.g., when the transition is low-rank (the scaling with $|\mathcal{S}|$ in Corollary 1 comes from the bracket number of tabular POMDPs and can be avoided with more clever function classes). In contrast, the scaling with $|\mathcal{S}|$ is inevitable in the sample complexity of [37] even if they choose a better function class because their **norm bound on the product of OOMs** also incurs dependence on $|\mathcal{S}|$ (and this is why the analysis in [37] is limited to tabular POMDP and cannot deal with low-rank POMDPs). In addition, since our analysis originates from **general PSRs where the observation matrix doesn’t exist**, we cannot assume a bound on the observation matrix and the assumption on the core matrix is inevitable. It is not surprising that the specific analysis in [37] requires a slightly weaker assumption on the norm since their analysis is **tailored to POMDPs** and ours is much **more general**.
> > > >
> > > > Second, for decodable POMDPs, we acknowledge that [13] may be able to avoid $O^m$ with a better function class while in our current general analysis $1/\alpha$ cannot be avoided with a better function class. However, as we mentioned before, this requirement on $1/\alpha$ occurs because we are applying a very general result to a very specific model. Indeed, for $m$-step decodable POMDPs, we can bound the norm of the product of predictive operators by $O(A^m)$ via direct calculation without inserting the core matrix. Then following the same analysis in our paper, we can obtain a sample complexity for $m$-step decodable POMDPs without incurring $1/\alpha$. The key idea we want to convey here is that surprisingly, **$m$-step decodable POMDPs are also PSRs with core test being $m$-step futures**, which is a new discovery and somehow **links the works on $m$-step weakly-revealing and $m$-step decodable POMDPs** for the first time.
> > > >
> > > >
> > > > Third, for low-rank POMDPs, we want to clarify that **the counterpart of $1/\alpha$ in [47] is $1/\nu$ in [47, Assumption 5.1]** since both $\alpha$ and $\nu$ characterize the difficulty of recovering the probability of arbitrary future tests with arbitrary length given the probability of m-step futures. As far as we know, there is no clear relationship between the magnitude of $1/\nu$ in [47, Assumption 5.1] and $1/\alpha$ in our analysis. That said, our analysis has **no counterpart assumption on the existence of bottleneck variables and past sufficiency** ([47, Assumption 5.2], which assumes that one can recover the probability of all future tests given some short histories). Therefore intuitively our analysis still holds when these two types of assumptions do not hold while the analysis in [47] will fail. In addition, note that **past sufficiency is not required to solve tabular POMDPs**. Thus, in fact [47] does not generalize tabular POMDP while ours does.
> > > >
> > > > Please let us know if you have additional concerns! Thank you again for your valuable feedbacks and questions!

---

> > > > ### Author Response · Authors · 2022-11-30
> > > > **Other Opinions?**
> > > >
> > > > Dear Reviewer,
> > > >
> > > > It has been a week since we posted our further response. Does it address all your concerns?

---

> ### Author Response · Authors · 2022-12-06
> **Further Opinions?**
>
> Dear Reviewer,
>
> There is only one week left for the discussion stage. Have we addressed all your concerns?

---

### Official Review · Reviewer_XgNc · 2022-10-27

**Confidence:** 3
**Correctness:** 4
**Technical Novelty And Significance:** 2
**Empirical Novelty And Significance:** 3
**Recommendation:** 6

**Clarity, Quality, Novelty And Reproducibility:**

The work is clearly written, the technical novelty seems limited but provides significant results.

**Strength And Weaknesses:**

The main strength of this paper is that it provides the first sample-efficient algorithm in learning PSRs, although under an additional regularity assumption. While previous work depends on latent states, this work uses  $M_{o, a, h}, q_0$  that is free from latent spaces. Technically, it leverages the properties of the core matrix $K_h$ to control the product of $M_{o, a, h}$ and thus controls the error induced by the estimation error of $M_{o, a, h}$ and $q_0$. This technique allows the CRANE algorithm to work on a more general PSR classes that subsume both weak-revealing settings as well as decodable settings.

There are a few weaknesses and questions:

- The regularity assumption (Assumption 1) is defined on the core matrix $K_h$ and seems to have limited applicability.  Are there any cases belonging to POMDPs that does not satisfy Assumption 1? The authors could better discuss the applicability of Assumption 1 by providing some counterexamples that violates the regularity condition.
- The work focuses on providing a polynomial sample complexity, but omits the specific dependency parameters in the main theorems.  Although providing the first polynomial complexity under a more general setting is significant, it is hard to compare the sample complexity result with previous results when restricted to specific instances. Such as, how to compare CRANE with Liu et al. under $m$-step weak-revealing POMDPs?
- In Liu et al. that proposed OMLE, they uses a generalized eluder-type dimension to bound the regret. What is the relationship of the complexity measure used in this paper ($d_{PSR}$) with the eluded-type dimension? Is it possible to extend complexity measure $d_{PSR}$ to a more general one?
- Since no experiments are provided, is CRANE applicable to real-world problems and computational efficient?
- Is there a specific reason that the authors focuses on the model-based setting? Is it possible to extend the optimism-based algorithm on POMDP to model-free settings?


**Summary Of The Paper:**

The paper considers the problem of provably efficient reinforcement learning on predictive state representations (PSRs) with function approximation. The main contribution of the paper is a PAC algorithm for PSRs with polynomial sample complexity when competing with the globally optimal policy. Although sharing similar design principles with traditional optimism based methods on POMDPs such as the OMLE algorithm, this work extends the scope of OMLE to a larger set of models (PSRs) featured by direct prediction of the future given the past trajectories. The more general model classes includes several examples including the $m$-step weak-revealing POMDPs, the $m$-step decodable POMDPs, etc. Moreover, the result can also be applied to POMDPs equipped with function approximations such as the low-rank structure, the linear structure, etc.

Overall, the proposed algorithm OptimistiC PSR leArniNg with MLE (CRANE) achieves polynomial sample complexity on a subset of PSR problems with a regularity condition. The $\alpha$-regularity condition on PSR is defined upon the core matrix and is a generalization of the weakly revealing condition. When instantiated to POMDPs, the algorithm can not only solve $m$-step weakly revealing POMDPs but also $m$-step decodable POMDPs, etc.

**Summary Of The Review:**

This paper provides the first polynomial sample complexity result for PSRs and is thus a solid contribution. The applicability of the assumption is limited, the sample complexity is not sharp, and the empirical impact is unclear. But it is in general a good submission.

---

> ### Author Response · Authors · 2022-11-10
> **Response (Part 1)**
>
> Thank you for the valuable feedback! Here are our responses:
>
> - **Applicability of Assumption 1**:
>
> First we want to point out that the core matrix $K_h$ is full rank by definition (see Equation (2)), therefore the pseudo-inverse of $K_h$, i.e., $K_h^{\dagger}$ must exist. However, when the system is ill-conditioned, the $\ell _ 1$-norm of $K_h ^ {\dagger}$ might be very large and lead to difficulty of learning the system. In fact, in Theorem 3 (Appendix K) we prove a lower bound showing that the sample complexity of learning the system has **inevitable polynomial scaling** with $\ell _ 1$-norm of $K_h ^ {\dagger}$.
>
> The reviewer asks about in POMDPs whether Assumption 1 will be violated, i.e., whether the $\ell _ 1$-norm of $K_h ^ {\dagger}$ can be **arbitrarily large**. Notably, in the proof of Theorem 3, we construct a POMDP whose $\Vert K_{h} ^ {\dagger} \Vert _1 = 1/\alpha$ where $\alpha>0$ can be **arbitrarily small**. In that special case, $K _ h$ is exactly the observation matrix $ O _ h$ and thus we only need to make the minimal eigenvalue of $O _ h$ small. This counterexample shed light on the applicability of Assumption 1 and the lower bounds in Theorem 3 validates the necessity of Assumption 1.
>
> We understand that the core matrix $K _ h$ is a little more abstract than the observation matrix $O _ h$ in the POMDPs. However, one key idea we want to convey is that the core matrix could be a **more essential quantity** than the observation matrix because $K  _ h$ is a matrix that **characterizes the relationship between future observations and past observations without introducing the existence of latent states**.  In fact, $K _ h$ is related to the **rank** of the system  [41] while $ O _ h $ corresponds to the latent states, i.e., the **non-negative rank** of the system. In Appendix D we give an example where the number of latent states (or the non-negative rank) of the system can be exponentially larger than the rank of the system, and our paper indeed tells that the hardness of learning a partially-observable system depends on its rank rather than non-negative rank.
>
>
> - **More Concrete Sample Complexity**:
>
> We apologize for the inconvenience. Our intention is to make our results in the main paper more clear, and we defer the more concrete sample complexity to Theorem 2 in Appendix I (in the format of regret upper bound, but you can easily obtain a more concrete sample complexity from the regret) where we provide the exact dependence of the parameters. As for the comparison between the results in Theorem 2 and POMDP literature, our dependence on $ U _ A$ is slightly worse because when applying eluder-dimension to bound the regret, we need to bound the norm in Equation (32) while for POMDPs you can get a smaller bound leveraging the existence of latent states.
>
> - **Eluder Dimension and the rank of PSR**:
>
> This is a good point and we understand that these two concepts are a little confusing. In fact, here eluder-dimension is just the name of the analysis technique to bound the generalization error, which basically says that given two groups of $d$-dimensional vectors, you can bound the generalization error by approximately $d$ times training error (see Proposition 22). This technique is utilized both by our work and Liu’s work [37]. In contrast, the rank of PSR $d _ {PSR}$ is a property of the partially observable system itself (i.e., the rank of the system dynamic matrix), similar to the number of latent states $S$ in the POMDPs.
>
> In our work, when we apply Proposition 22, we have the dimension $d$ of the vectors to be $d _ {PSR}$ while in [37] they let it be $ S $. As mentioned before, $S$ can be exponentially larger than $d _ {PSR}$ and $d _ {PSR}$ might be a more essential quantity than $S$.

---

> > ### Author Response · Authors · 2022-11-10
> > **Response (Part 2)**
> >
> > - **Computational Efficiency**:
> >
> > The computation of CRANE mainly requires an MLE oracle (Equation 5) and an optimistic planning oracle to find $f ^ k$ and $\pi ^ k$. The former one is realistic in practice and there are a bunch of MLE oracles. However, the latter one is not computationally efficient. In fact, **even planning without optimism in POMDPs is hard** and there are many computationally infeasible results (e.g., refer to the works listed at the end of this response). As a result, almost all of the current online learning work in POMDPs [3,20,29,34,37] are not computationally efficient other than [18], which requires quasi-polynomial computation time.
> >
> > Results on hardness of planning in POMDPs:
> > 1. Christos H Papadimitriou and John N Tsitsiklis. The complexity of markov decision processes. Mathematics of operations research, 12(3):441–450, 1987.
> >
> > 2. Michael L Littman. Memoryless policies: Theoretical limitations and practical results. From animals to animats, 3:238–245, 1994.
> >
> > 3. Dima Burago, Michel De Rougemont, and Anatol Slissenko. On the complexity of partially observed markov decision processes. Theoretical Computer Science, 157(2):161–183, 1996.
> >
> > 4. Christopher Lusena, Judy Goldsmith, and Martin Mundhenk. Nonapproximability results for partially observable markov decision processes. Journal of artificial intelligence research, 14:83–103, 2001.
> >
> >
> > - **The Reason for Model-Based Learning**:
> >
> > For model-free learning, you first need to define $Q$ function or its analog. However, in partially observable systems, you can not observe the latent states (if they exist) and thus the $Q$ function needs to depend on the whole history, i.e., $Q _ h (\tau _ h, a _ h)$, whose space can scale exponentially with the horizon $H$. This is the reason why in the model-free literature for partially observable systems people can only compete against $M$-step history policies (i.e., the policy only depends on $M$-length history instead of the whole history) and their sample complexity will typically scale with $|O| ^ M$ (in fact, [45] discusses a model-free approach in PSRs and they incur $|O| ^ M$).  In our paper we want to compete against all general policies and thus we believe we need to resort to model-based algorithms.

---

> ### Author Response · Authors · 2022-12-06
> **Further Opinions?**
>
> Dear Reviewer,
>
> There is only one week left for the discussion stage. Have we addressed all your concerns?

---

### Author Response · Authors · 2022-11-10
**Response to All**

We thank all reviewers for reviewing our paper and providing constructive feedback. We have addressed your concerns and please see the detailed responses. Below we give a summary of our rebuttal.

First, we want to emphasize that the novelty of this work is to show that large-scale partially observable systems with **small PSR rank** are PAC learnable using **general function approximation**. Prior most related work ([37]) focuses on tabular POMDPs which are a special case of PSRs. Second, since PSRs are a much more general model (i.e., we show that it can be exponentially more general than POMDPs in Appendix D), our analysis indeed requires **novel insights and techniques**, and we need to rely on **function approximation techniques** to handle potentially exponentially large observation space. Finally, our general result allows us to instantiate our theorem to various special models, including **low-rank POMDPs**, **linear POMDPs**, and **$m$-step decodable POMDPs**. For all these special models, the instantiated corollaries from our result **improve or complement the literature** [7,13,47]. We emphasize that the closely related work [37] does not have such generality, and in fact they only study tabular POMDPs.

---

### Author Response · Authors · 2022-11-20
**Further Discussion**

Dear Reviewers and Area Chair,

We have addressed all concerns, and are happy to discuss more. Please tell us if you have other questions and opinions!

---

### Decision · Program_Chairs · 2023-01-20

**Decision:**

Accept: poster

**Justification For Why Not Higher Score:**

PSRs are a general and powerful model, but the assumptions needed in this paper are difficult to interpret. A cursory reading of Table 1 could leave the reader thinking that the results here subsume all previous work, but even in settings where POMDPs were previously known to be learnable, they do not necessarily satisfy the requirements here.

**Justification For Why Not Lower Score:**

Nevertheless this paper is an important contribution. It shows how to extend the optimistic MLE framework to a richer class of models that extends and generalizes previous work along some interesting directions.

**Metareview: Summary, Strengths And Weaknesses:**

This paper studies the problem of efficiently learning a nearly optimal policy for PSRs with function approximation. In particular they give an algorithm with polynomial sample complexity. Their algorithm, CRANE, is an optimistic MLE algorithm. Furthermore they give applications to weakly revealing POMDPs and decodable POMDPs. The main selling point of this paper is the generality of PSRs. However the quantity $1/\alpha$ which plays a key role in the bounds is difficult to interpret. Indeed even for simple POMDP settings which were already known to be learnable from previous work, the quantity $1/\alpha$ need not be bounded. The paper also assumes the existence of "core histories" which seems like a natural surrogate for the latent states in a POMDP.

**Note From Pc:**

if the above contains the word "oral" or "spotlight" please see: "oral" presentation means -> notable-top-5% and "spotlight" means -> notable-top-25%. As stated in our emails, we are disassociating presentation type from AC recommendations

**Summary Of Ac-Reviewer Meeting:**

N/A